# Learning Provably Improves the Convergence of Gradient Descent

Qingyu Song[*]
**Xiamen University**
simmonssong96@gmail.com

Wei Lin, Hong Xu
**The Chinese University of Hong Kong**
wlin23@cse.cuhk.edu.hk, hongxu@cuhk.edu.hk

## Abstract

Learn to Optimize (L2O) trains deep neural network-based solvers for optimization, achieving success in accelerating convex problems and improving non-convex solutions. However, L2O lacks rigorous theoretical backing for its own training convergence, as existing analyses often use unrealistic assumptions—a gap this work highlights empirically. We bridge this gap by proving the training convergence of L2O models that learn Gradient Descent (GD) hyperparameters for quadratic programming, leveraging the Neural Tangent Kernel (NTK) theory. We propose a deterministic initialization strategy to support our theoretical results and promote stable training over extended optimization horizons by mitigating gradient explosion. Our L2O framework demonstrates over 50% better optimality than GD and superior robustness over state-of-the-art L2O methods on synthetic datasets. The code of our method can be found from https://github.com/NetX-lab/MathL2OProof-Official.

## 1 Introduction

Learn to optimize (L2O) represents an increasingly influential paradigm for tackling optimization problems [6]. Numerous studies have demonstrated the efficacy of employing learning-based models to achieve superior performance across a spectrum of optimization tasks. These encompass convex problems, exemplified by LASSO [7, 8, 22] and logistic regression [23, 34], and non-convex scenarios such as MIMO sum-rate maximization [35] and network resource allocation [33].

Distinct from black-box approaches [5, 36, 41], which directly derive solutions to optimization problems from a neural network (NN), the so-called "white-box" methodologies are garnering increased attention. This heightened interest stems from their inherent advantages, such as enhanced trustworthiness [14] and theoretical guarantees [34]. A key characteristic of these white-box strategies is the integration of mechanisms to ensure the "controllability" of the generated solutions. For instance, Lv et al. [25] employ a NN to predict the step size for the gradient descent (GD) algorithm, where the inherent structure of GD stabilizes the optimization trajectory. Similarly, Heaton et al. [14] integrate a conventional solver within an L2O framework to act as a safeguard, thereby preventing the learning-based model from producing solutions with extreme violations. This principle of guided or constrained learning has also been extended to the training phase of L2O models [39].

Further, "unrolling" has emerged as a prominent technique within L2O [6], characterized by the strategic replacement of components of conventional optimization algorithms with neural network (NN) blocks [12, 15, 20]. For instance, Liu et al. [23] introduce Math-L2O that imposes architectural constraints on unrolled L2O models by deriving necessary conditions for their convergence. Their analysis revealed that for a L2O model to achieve optimality, its embedded NN must effectively perform a linear combination of input feature vectors, weighted by learnable parameter matrices.

---

[*]This work was done at The Chinese University of Hong Kong (CUHK) when Qingyu was a PhD candidate.

39th Conference on Neural Information Processing Systems (NeurIPS 2025).

Empirical validation demonstrates that the proposed methods exhibit strong generalization capabilities when trained using a coordinate-wise input-to-output strategy. Subsequent research by Song et al. [34] further enhance this generalization performance by reducing the magnitude of input features.

Despite these advancements, to the best of our knowledge, a formal demonstration of the convergence for unrolling-based L2O methods in solving general optimization problems remains elusive. While LISTA-CPSS [7] establishes convergence for the well-known LISTA framework [12], its analysis is based on the assumption that neural network (NN) outputs are confined to a specific subspace, a condition that is often not met in practical implementations. Similarly, while Math-L2O [23] derives necessary conditions for convergence, the mechanisms by which the training process itself can guarantee such convergence are not elucidated. Subsequent analysis by Song et al. [34] investigates the inference-time convergence of Math-L2O. However, this work relies on a stringent training assumption, effectively constraining the L2O model to emulate the behavior of a conventional Gradient Descent (GD) algorithm.

This apparent deficiency in comprehensively demonstrating L2O convergence stems from two fundamental, unresolved technical challenges. First, unrolling-based L2O models [8, 12, 22] represent a specialized class of NN architectures. Despite much progress in understanding the training convergence of general neural networks (NNs), notably through the Neural Tangent Kernel (NTK) theory since 2019 [2, 3, 11, 24, 29, 30], a formal proof of training convergence remains conspicuously absent. Such a proof is an essential precursor to establishing the convergence of the L2O model in its primary task of solving optimization problems. Second, the precise relationship between the training convergence achieved during the L2O model's training phase (i.e., optimizing the NN parameters) and the convergence of the L2O model when applied to the target optimization problem (i.e., finding the optimal solution) is not well understood. For instance, Math-L2O [23] is designed to learn the step size for an underlying GD algorithm. While the problem-solving efficacy of Math-L2O is naturally evaluated based on the progression of GD iterations, its training convergence is measured in terms of training steps (e.g., epochs). These two notions of convergence: one on model parameter optimization and the other on problem-solving iterations, are largely decoupled and operate on fundamentally different scales.

In this work, we present the first rigorous demonstration that an unrolling framework can achieve theoretical convergence in solving optimization problems. Our analysis focuses on the state-of-the-art (SOTA) Math-L2O framework, wherein a NN functions as a recurrent block, iteratively generating hyperparameters for an underlying optimization algorithm. The solution obtained at each iteration, which utilizes these generated hyperparameters, is then incorporated as an input feature for the subsequent iteration [23]. This inherent recurrence imparts RNN-like characteristics to Math-L2O, significantly complicating the analysis of its training convergence. Specifically, the recurrent structure causes the NN to manifest as a high-order polynomial function with respect to (w.r.t.) its input features [3]. This characteristic poses challenges for establishing tight analytical bounds, potentially leading to looser convergence rates compared to non-recurrent architectures, as highlighted in related NTK analyses for RNNs [3]. Moreover, the Math-L2O architecture introduces an additional layer of complexity: the emergence of high-order polynomial dependencies not only on the input features but also on the learnable parameters themselves. This distinct feature renders the convergence proof for Math-L2O arguably more intricate than those for conventional RNNs, where such parameter-dependent high-order terms are typically less pronounced.

We address the pivotal connection between the NN's training convergence and the ultimate problem-solving convergence of the L2O model. Within the Math-L2O framework, we establish this critical linkage by explicitly demonstrating an alignment between the convergence dynamics exhibited during the NN's training phase and the convergence characteristics of its underlying backbone optimization algorithm. This alignment provides a novel theoretical bridge, ensuring that a successfully trained L2O model translates to effective convergence when applied to optimization tasks. Our contributions are summarized as follows:

1. We provide a formal proof that the Math-L2O training framework substantially enhances the convergence performance of its underlying backbone algorithms. This is achieved by rigorously establishing an explicit alignment between the convergence rates of the training process and the iterative steps of the backbone algorithm.

2. We establish the first linear convergence rate for Math-L2O training. Inspired by [29], we employ a NN architecture with a single wide layer and utilize NTK to prove the boundedness of NN outputs, gradients, and the training loss function within the Math-L2O framework.

3. We introduce a novel deterministic parameter initialization scheme, coupled with a specific learning rate configuration strategy. This combined approach is proven to guarantee the training convergence of the Math-L2O model across all iterations.

4. We empirically validate our theoretical findings through comprehensive experiments. The results showcase significant performance advantages, including up to a 50% improvement in solution optimality over the standard GD algorithm post-training, and superior robustness compared to SOTA L2O models and the Adam optimizer [10]. Furthermore, ablation studies empirically confirm the practical efficacy and individual contributions of our proposed theorems.

## 2 Preliminary

This section first defines the optimization problem objective and the L2O framework. The L2O training loss is then formulated based on these definitions. Then, the NN's computational graph is employed to detail the forward pass and the derivation of parameter gradients.

### 2.1 Definitions

Let $d > b$, suppose $x \in \mathbb{R}^{d \times 1}$, $y \in \mathbb{R}^{b \times 1}$, and $\mathbf{M} \in \mathbb{R}^{b \times d}$, we define the optimization objective as:

$$\min_{x \in \mathbb{R}^d} f(x) = \frac{1}{2}\|\mathbf{M}x - y\|_2^2. \tag{1}$$

This objective function is commonly selected for convergence analysis [4]. The least-squares problem, a frequent subject in NN convergence studies [2, 3, 11, 21, 29], is a specific instance of the minimization in Equation (1) where $d = b$ and $\mathbf{M} = \mathbf{I}$.

We assume $f$ to be $\beta$-smooth, such that $\|\mathbf{M}^\top \mathbf{M}\|_2 \leq \beta$, and $\mathbf{M}$ to possess full row rank, with $\lambda_{\min}(\mathbf{M}\mathbf{M}^\top) = \beta_0 > 0$. This setting often favors numerical algorithms (e.g., GD) over analytical solutions due to computational complexity. GD's $\mathbf{O}(bd)$ complexity is typically lower than the $\mathbf{O}(b^3)$ of analytical methods involving costly matrix inversions. The loss function is then defined as the sum of $N$ objectives specified in Equation (1):

$$F(X) = \frac{1}{2}\|\mathbf{M}X - Y\|_2^2, \tag{2}$$

where $F$, $\mathbf{M} \in \mathbb{R}^{Nb \times Nd}$, $X \in \mathbb{R}^{Nd \times 1}$, and $Y \in \mathbb{R}^{Nb \times 1}$ represent the concatenated objectives, parameters, variables, and labels, respectively, from $N$ optimization problems (see Appendix A.1 for details). $F$ is also $\beta$-smooth, given that $\|\mathbf{M}^T\mathbf{M}\|_2 \leq \max_{i=1,\ldots,N}\{\|\mathbf{M}_i^T\mathbf{M}_i\|_2\} = \beta$.

**Learn to Optimize (L2O).** Given an initial point $X_0$, L2O takes $X_0$ as the input and generates a solution, denoted as $X_t$, with a machine learning model. Typically, let $g_W$ denote an $L$-layer NN with parameters $W = \{W_1, \ldots, W_L\}$, $W_\ell \in \mathbb{R}^{n_\ell \times n_{\ell-1}}$, $n_1, \ldots, n_L \in \mathbb{R}$. Math-L2O [23] takes an iterative workflow to generate solutions. For each step $t \in [T]$ in solving the problem in Equation (1), the NN model in Math-L2O is defined as $g_W(X_{t-1}, \nabla F(X_{t-1}))$. The NN receives the current state variable $X_{t-1}$ and its gradient $\nabla F(X_{t-1})$ as input. The update rule at step $t$, which employs the Hadamard product (denoted by $\odot$), is formulated as:

$$X_t = X_{t-1} - \frac{1}{\beta}P_t \odot \nabla F(X_{t-1}), \quad P_t = g_W(X_{t-1}, \nabla F(X_{t-1})). \tag{3}$$

$P_t$ represents a vector whose entries are learned step sizes. The NN $g_W$ takes structured layer-wise architecture. It employs a coordinate-wise architecture, processing each input dimension independently, recognized for its robustness in L2O applications [23, 34]. Thus, output dimension of the NN is one, i.e., $n_L = 1$. Denote $[L] := \{1, \ldots, L\}$, for layer $\ell \in [L]$, we denote $G_{\ell,t}$ as the (inner) output of layer $\ell$ at step $t$. Utilizing ReLU (ReLU) [1] and Sigmoid ($\sigma$) [27] activations, $G_{\ell,t}$ is defined as:

$$G_{\ell,t} = \begin{cases} [X_{t-1}, \nabla F(X_{t-1})]^\top & \ell = 0, \\ \text{ReLU}(W_\ell G_{\ell-1,t}) & \ell \in [L-1], \\ P_t = 2\sigma(W_L G_{L-1,t})^\top & \ell = L. \end{cases} \tag{4}$$

The L2O training problem is defined by:

$$F(W) = \tfrac{1}{2}\|\mathbf{M}X_T - Y\|_2^2, \quad X_T = L2O_W(X_0, \nabla F(X_0)). \tag{5}$$

## 2.2 Layer-Wise Derivative of NN's Parameters

Let $k$ denote a training iteration for loss Equation (5) minimization, which is distinct from an optimization step $t$ for solving objective Equation (2). The computational graph in Figure 1 illustrates the Math-L2O forward and backward operations, which parallel those of Recurrent Neural Networks (RNNs) [13]. Figure 1a details the NN block (see Equation (4)). Figure 1b depicts the overall process: the block takes an input solution, performs $T$ internal optimization steps to produce an updated solution (red dashed arrows), and each training iteration $k$ triggers a full backward pass (blue bold lines). As per [23], the gradient flow from the input features to the NN block is detached.

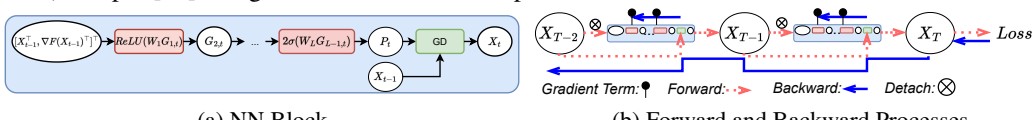

| (a) NN Block | (b) Forward and Backward Processes |

Figure 1: Computational Graph of Math-L2O

The derivative of an objective $F$ w.r.t. the parameters $W_\ell$ of layer $\ell$ is determined via the computational graph, paralleling Back-Propagation-Through-Time (BPTT) for RNNs [26]:

$$\frac{\partial F}{\partial W_\ell} = \frac{\partial F(X_T)}{\partial X_T}\left(\sum_{t=1}^{T}\left(\prod_{j=T}^{t+1}\frac{\partial X_j}{\partial X_{j-1}}\right)\frac{\partial X_t}{\partial P_t}\frac{\partial P_t}{\partial W_\ell}\right). \tag{6}$$

The summation aggregates gradients across $T$ optimization steps. $\prod_{j=T}^{t+1}(\partial X_j/\partial X_{j-1})$ represents the chain rule application from the final output $X_T$ to an intermediate state $X_t$.

Moreover, we derive two key gradients, instrumental for establishing the theoretical results in the ensuing section. Following Definition 2.2 in [2], the gradient of the ReLU is represented by a diagonal matrix $\mathbf{D}_\ell^t$, where its $i$-th diagonal element is $[\mathbf{D}_\ell^t]_{i,i} := \mathbf{1}_{(W_\ell G_{\ell-1,t})_i \geq 0}$ for $i \in [n_\ell]$. Let $\Gamma_t := \mathbf{M}^\top(\mathbf{M}X_t - Y)$ and $\Xi_\ell := (\mathbf{I}_d \otimes W_L)(\prod_{j=L-1}^{\ell+1}\mathbf{D}_{j,t}(\mathbf{I}_d \otimes W_j))\mathbf{I}_{n_\ell}$. Defining $\mathcal{D}(\cdot)$ as the operator that constructs a diagonal matrix from a vector, the gradients for an inner layer $W_\ell$ ($\ell < L$) and the final layer $W_L$ are given by:

$$\frac{\partial F}{\partial W_\ell} = -\tfrac{1}{\beta}\Gamma_T^\top \sum_{t=1}^{T}\left(\prod_{j=T}^{t+1}(\mathbf{I}_d - \tfrac{1}{\beta}\mathbf{M}^\top\mathbf{M}\mathcal{D}(P_j))\right)\mathcal{D}(\Gamma_t)\mathcal{D}\left(P_t \odot (1 - P_t/2)\right)\Xi_\ell \otimes G_{\ell-1,t}^\top, \tag{7}$$

$$\frac{\partial F}{\partial W_L} = -\tfrac{1}{\beta}\Gamma_T^\top \sum_{t=1}^{T}\left(\prod_{j=T}^{t+1}(\mathbf{I}_d - \tfrac{1}{\beta}\mathcal{D}(P_j)\mathbf{M}^\top\mathbf{M})\right)\mathcal{D}(\Gamma_T)\mathcal{D}\left(P_t \odot (1 - P_t/2)\right)G_{L-1,t}^\top, \tag{8}$$

where $\otimes$ denotes the Kronecker product. Equation (8) (for $W_L$) differs from Equation (7) (for $W_\ell$) in its final terms: $G_{L-1,t}^\top$ replaces $\Xi_L \otimes G_{\ell-1,t}^\top$. This simplification arises as $W_L$ is the terminal layer, and $G_{L-1,t}$ is its direct input from layer $L-1$. Thus, its gradient calculation does not involve a subsequent layer propagation factor analogous to $\Xi_L$.

# 3 L2O Convergence Demonstration Framework

This section rigorously substantiates the convergence of the L2O framework, Math-L2O. We first expose theoretical and numerical instabilities prevalent in current SOTA L2O methods. Then, we demonstrate Math-L2O's accelerated training convergence compared to GD and then present a formal methodology to establish its convergence.

## 3.1 Limitations Analysis of Existing SOTA L2O Frameworks

We analyze limitations in the convergence guarantees of two SOTA L2O frameworks: LISTA-CPSS [7] and Math-L2O [23]. LISTA-CPSS [7] constructively proves that its predecessor, LISTA [12], can attain a linear convergence rate. However, this theoretical guarantee is contingent upon several stringent conditions. Math-L2O [23] proposes an L2O framework derived from the GD algorithm, incorporating necessary conditions for convergence. Both frameworks employ sequential solution updates and utilize BPTT for parameter optimization.

Initially, we assess training loss across varying optimization steps. This is pertinent due to the well-documented issue of gradient explosion of BPTT arising from long-term gradient accumulation [19].

Both models are trained on 10 randomly sampled optimization problems for 400 epochs. Figure 2 depicts training losses (y-axis) against optimization steps (x-axis) for several learning rates (distinguished by line color). Data points exhibiting numerical overflow (indicative of gradient explosion at the first training iteration) are excluded, resulting in plot lines terminating before 100 steps for affected configurations. The results demonstrate that both frameworks suffer from poor convergence at low learning rates (LRs) and training instability at high LRs.

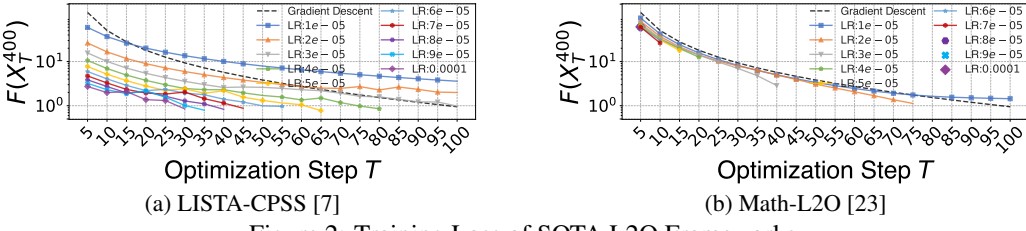

(a) LISTA-CPSS [7]        (b) Math-L2O [23]

Figure 2: Training Loss of SOTA L2O Frameworks

Further, we examine the convergence conditions outlined for LISTA-CPSS [7], illustrating their propensity for violation during typical training procedures. The first condition mandates asymptotic sign consistency between iterates $X_t$ and the solution $X^*$, requiring $\text{sign}(X_t) = \text{sign}(X^*)$ for all $t$. The second condition imposes constraints on the columns of the learned parameter matrix $\mathbf{W}$ relative to the columns of the objective coefficient matrix $\mathbf{M}$. Specifically, denoting column indices by $i$ and $j$, it necessitates that $\mathbf{W}_i^\top \mathbf{M}_i = 1$ and $\mathbf{W}_i^\top \mathbf{M}_j > 1$ for all $j \neq i$.

Following the experimental design in [23], we quantify the violation percentage of the aforementioned conditions during inference. The results are presented in Figure 3. We consider two settings: (i) shared parameters $W$ across iterations (Figure 3a), and (ii) unique parameters $W_t$ per step $t$ (Figure 3b). Both scenarios reveal that the specified conditions are frequently violated post-training. For instance, in the shared $W$ case (Figure 3a), while the conditions hold in later steps, substantial violations occur in early steps. The divergence contradicts the convergence rate analysis presented in [7].

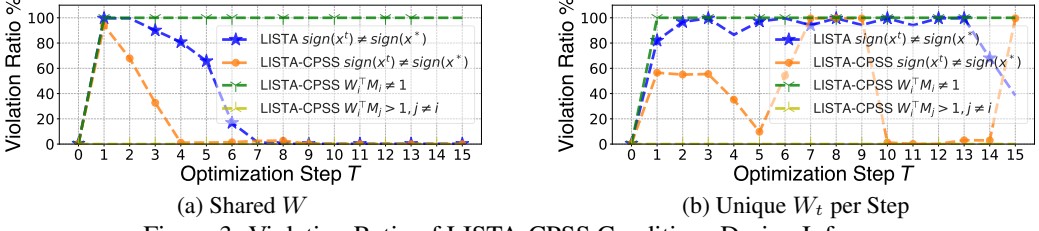

(a) Shared $W$        (b) Unique $W_t$ per Step

Figure 3: Violation Ratio of LISTA-CPSS Conditions During Inference

The preceding observations highlight that training is indispensable for L2O convergence analysis. Three fundamental questions arise in L2O: (i) *What is the impact of training on convergence?* (ii) *How can training be incorporated into the convergence analysis framework?* (iii) *What mechanisms ensure a stable training process?* We propose a concise approach to address these questions, establishing a direct alignment between the training's convergence rate and an existing algorithm's rate.

### 3.2 L2O Convergence Demonstration: Aligning L2O to An Algorithm

First, we introduce a general convergence analysis framework. Let $X^*$ be the optimal solution, $r_t$ represents an iteration-dependent rate term, and $C(X_0)$ be a constant that dependent on the initial point $X_0$ (and $X^*$), the convergence rate of an algorithm (either learned or classical) for minimizing an objective $F(X)$ (e.g., the objective in Equation (1) or the loss in Equation (2)) is often formulated as: $F(X_t) \leq r_t C(X_0)$. For example, standard GD has a rate of $F(X_t) \leq \frac{\beta}{t}\|X_0 - X^*\|_2^2$ [4].

The performance of L2O models, stabilized via training, is typically assessed after $T$ iterations [23, 34]. We formulate the L2O training convergence rate w.r.t. training iteration $k$ as:

$$F(X_T^k) \leq r_k C(X_T^0), \quad \text{where } X_T^0 = \text{L2O}_{W^k}(X_0^0), \tag{9}$$

with $X_T^0$ being the initial solution from the L2O model and a constant mapping $C$. Based on the proof in [38], non-learning GD algorithm's convergence rate corresponding to the initial L2O state is:

$$F(X_T^0) \leq \frac{\beta}{T}\|X_0^0 - X^*\|_2^2. \tag{10}$$

Given the independence of training iteration $k$ and optimization step $T$, we align the LHS of Equation (9) with the RHS of Equation (10) by setting $C(X_T^0) = F(X_T^0)$. Given initial point is constant that $X_T^0 = X_T^k$, this yields the combined training convergence rate:

$$F(X_T^k) \le r_k \frac{\beta}{T} \|X_0^k - X^*\|_2^2. \tag{11}$$

Here, the LHS represents the objective value after $k$ training iterations, while the RHS is a constant term dependent on the initial point $X_0$. W.r.t. $T$, Equation (11) demonstrates a sub-linear convergence rate of at least $\mathcal{O}(1/T)$. The rate indicates that integrating L2O with an existing algorithm via training can enhance its convergence. Such integration is achieved by the Math-L2O framework [23], which utilizes a NN to learn hyperparameters for non-learning algorithms (e.g., step size for GD, step size and momentum for Nesterov Accelerated Gradient [4]).

Further, we construct the Math-L2O training rate $r_k$ (see Equation (9)). Section 4 establishes its linear convergence. Subsequently, Section 5 proposes a deterministic initialization strategy to ensure the alignment ($C(X_T^0) = F(X_T^0)$) and uphold the theoretical conditions for this linear rate.

## 4 Linear Convergence of L2O Training

In this section, we establish the linear convergence rate for training a Math-L2O model employing an over-parameterized NN, w.r.t. the loss defined in Equation (2). By training the NN (Equation (4)) using GD, we establish its linear convergence rate via NTK theory. Classical NTK theory [16] requires infinite NN width to maintain a non-singular kernel matrix, which facilitates a gradient lower bound akin to the Polyak-Lojasiewicz condition [29, 32]. Applying the relaxation from [29] and the rigorous NN formalizations (Section 2), we demonstrate that an NN width of $\mathcal{O}(Nd)$ is sufficient.

To derive the rate, we first introduce a lemma to bound Math-L2O's gradients. We then prove that appropriate initialization leads to deterministic loss minimization in the initial training iteration. After that, we develop a strategy to maintain this property throughout training, thereby ensuring convergence. This approach culminates in a linear convergence rate for an $\mathcal{O}(Nd)$-width NN. The main results are summarized herein, with detailed proofs deferred to Appendix A.5 and Appendix A.6.

### 4.1 Bound Outputs of Math-L2O

We define $\alpha_0 := \sigma_{\min}(G_{L-1,T}^0)$ and let $C_\ell > 0$ for $\ell \in [L]$ be any sequence of positive numbers. Moreover, for $t, j \in [T]$, we define the following quantities:

$$
\begin{aligned}
&\bar{\lambda}_\ell = \|W_\ell^0\|_2 + C_\ell, \quad \Theta_L = \prod_{\ell=1}^L \bar{\lambda}_\ell, \quad \Phi_j = \|X_0\|_2 + \frac{2j-1}{\beta}\|\mathbf{M}^\top Y\|_2, \\
&\Lambda_j = (1+\beta)\|X_0\|_2^2 + \frac{(4j-3)(1+\beta)+\beta}{\beta}\|X_0\|_2\|\mathbf{M}^\top Y\|_2 + \frac{(2j-1)(\beta(2j-1)+(2j-2))}{\beta^2}\|\mathbf{M}^\top Y\|_2^2, \\
&S_{\Lambda,T} = \sum_{t=1}^T \Lambda_t, \qquad\qquad\qquad \delta_1^t = \sum_{s=1}^t \left( \prod_{j=s+1}^t (1 + \frac{1+\beta}{2}\Theta_L \Phi_j) \right) \Lambda_s, \\
&S_{\bar{\lambda},L} = \sum_{\ell=1}^L \bar{\lambda}_\ell^{-2}, \qquad\qquad\quad \delta_2 = \sum_{s=1}^{T-1} \left( \prod_{j=s+1}^{T-1} (1 + \frac{1+\beta}{2}\Theta_L \Phi_j) \right) \Lambda_s, \\
&\zeta_1 = \sqrt{\beta}\|X_0\|_2 + (2T+1)\|Y\|_2, \quad \delta_3 = (1+\beta)\|X_0\|_2 + \left(2T-1+\frac{2T-2}{\beta}\right)\|\mathbf{M}^\top Y\|_2, \\
&\zeta_2 = \|X_0\|_2 + \frac{2T-2}{\beta}\|\mathbf{M}^\top Y\|_2, \quad \delta_4 = \sigma(\delta_3 \Theta_L)(1 - \sigma(\delta_3 \Theta_L)),
\end{aligned}
\tag{12}
$$

where $X_0$ denotes the initial point, and $\mathbf{M}$ (parameter matrix) and $Y$ (labels) are input features from Equation (2). The defined quantities are positive under the conditions $j \ge 1$ and $\bar{\lambda}_\ell > 0$.

First, we derive a bound for the training gradients by considering them as perturbations from initialization. This bound relates the gradient magnitude to the objective function in Equation (2), as detailed in the following lemma. Despite the derivative for inner layers (Equation (7)) containing an additional term compared to that of the last layer (Equation (8)), a uniform bound as stated applies. The proof is provided in Appendix A.5.4.

**Lemma 4.1.** *Assuming* $\max(\|W_\ell^{k+1}\|_2, \|W_\ell^k\|_2) \le \bar{\lambda}_\ell$ *for* $\ell \in [L]$, *for any training iteration* $k$, *the gradient of the $\ell$-th layer parameters* $W_\ell^k$ *is bounded by:* $\left\| \frac{\partial F}{\partial W_\ell^k} \right\|_2 \le \frac{\sqrt{\beta}\Theta_L S_{\Lambda,T}}{2\bar{\lambda}_\ell} \|\mathbf{M}X_T^k - Y\|_2$.

Building upon Lemmas 4.1 and A.6 and auxiliary results (see Appendix A.5), we analyze the dynamics of the final solution $X_T$ w.r.t. parameter updates during training. The subsequent lemma

establishes a rigorous formulation for the fluctuation of $X_T$ in response to changes in parameters between adjacent training iterations. This result demonstrates that Math-L2O, viewed as a function of its learnable parameters, exhibits semi-smoothness, aligning with findings for ReLU-Nets in [29]. The proof is provided in Appendix A.5.3.

The semi-smoothness of the Math-L2O NN is preserved despite its recurrent operations. The coefficient associated with $\|W_\ell^{k+1} - W_\ell^k\|_2$ exhibits $\mathcal{O}(e^{LT})$ scaling, where $e$ is an initialization parameter detailed in Section 5. This represents a looser bound compared to that for ReLU-Nets [29], which is a consequence of Math-L2O's greater architectural complexity, specifically the $T$-fold execution of an $L$-layer NN block (see Equation (8)). However, this scaling behavior is consistent with observations for other deep architectures [2].

**Lemma 4.2.** *For any training iteration $k$, assume there exist constants $\bar\lambda_\ell \in \mathbb{R}^+$ for $\ell \in [L]$ such that $\max_{k' \in \{k,k+1\}} \|W_\ell^{k'}\|_2 \leq \bar\lambda_\ell$. Let $X_t^{k+1}$ and $X_t^k$ be outputs of the Math-L2O (defined in Equations (3) and (4)) corresponding to parameters $W^{k+1} = \{W_\ell^{k+1}\}_{\ell=1}^L$ and $W^k = \{W_\ell^k\}_{\ell=1}^L$, respectively. Then, Math-L2O exhibits the following semi-smoothness property:*

$$\|X_t^{k+1} - X_t^k\|_2 \leq \tfrac{1}{2}\sum_{s=1}^{t-1}\big(\prod_{j=s+1}^t (1 + (1+\beta)/2\Theta_L\Phi_j)\big)\Lambda_s\Theta_L\big(\sum_{\ell=1}^L \bar\lambda_\ell^{-1}\|W_\ell^{k+1} - W_\ell^k\|_2\big).$$

Lemma 4.2 demonstrates that Math-L2O solutions exhibit a bounded response to perturbations in its NN parameters. This finding, in conjunction with Lemma 4.1, facilitates a more nuanced analysis of the loss dynamics. Further, judicious selection of learning rates enables control over the evolution of NN parameters. Such control is instrumental in bounding the constant quantities from these lemmas, thereby establishing the desired convergence rate presented in the subsequent theorem.

## 4.2 Linear Training Convergence Rate of Math-L2O

Leveraging the bounds on Math-L2O's output (Lemma A.6) and its gradient (Lemma 4.1), the following theorem establishes the linear convergence rate for training the Math-L2O model. The proof is provided in Appendix A.6.

**Theorem 4.3.** *Consider the NN defined in Equation (4), using quantities from Equation (12), suppose the following conditions hold at initialization:*

$$\alpha_0 \geq 8(1+\beta)\zeta_2, \quad (13\text{a}) \qquad \alpha_0^2 \geq \frac{\beta^3}{4\beta_0^2}\delta_4^{-2}\big(-\tfrac{1}{2}\Theta_{L-1}^2\Lambda_T S_{\Lambda,T-1} + \Theta_L^2(\Lambda_T + \delta_2)S_{\bar\lambda,L}S_{\Lambda,T}\big). \quad (13\text{b})$$

$$\alpha_0^2 \geq \max_{\ell \in [L]} \frac{\Theta_L}{C_\ell\bar\lambda_\ell}\frac{\beta^2\sqrt{\beta}}{8\beta_0^2}\delta_4^{-2}\zeta_1 S_{\Lambda,T}, \quad (13\text{c}) \qquad \alpha_0^3 \geq \frac{(1+\beta)\beta^2\sqrt{\beta}}{2\beta_0^2}\delta_4^{-2}\Theta_L\Theta_{L-1}\zeta_1\zeta_2 S_{\bar\lambda,L}S_{\Lambda,T}, \quad (13\text{d})$$

*Let the learning rate $\eta$ satisfy:*

$$\eta < \tfrac{8}{\beta}(\delta_2 + \Lambda_T)\big(\delta_2 + \Theta_L S_{\Lambda,T}S_{\bar\lambda,L}\big)^{-1}S_{\Lambda,T}^{-2}, \quad (14\text{a}) \qquad \eta < \tfrac{1}{4}\frac{\beta^2}{\beta_0^2}\delta_4^{-2}\alpha_0^{-2}. \quad (14\text{b})$$

*Then, for weights $W^k = \{W_\ell^k\}_{\ell=1}^L$ at training iteration $k$, the loss function $F(W^k)$ converges linearly to a global minimum:*

$$F(W^k) \leq \big(1 - 4\eta\frac{\beta_0^2}{\beta^2}\delta_4\alpha_0^2\big)^k F(W^0).$$

*Remark* 1. $(1 - 4\eta\frac{\beta_0^2}{\beta^2}\delta_4\alpha_0^2)^k$ is $r_k$ in Equation (11), which is a less than one term since $\delta_4 = \sigma(\delta_3\Theta_L)(1 - \sigma(\delta_3\Theta_L)) > 0$ and $\alpha_0 := \sigma_{\min}(G_{L-1,T}^0) > 0$ ($G_{L-1,T}^0$ is a thin matrix), which ensure that the L2O converges at least as fast as GD.

Equations (14a) and (14b) are based on the quantities defined in Equation (12). Each quantity represents an inner formulation in the demonstration of lemmas and theorems. We use these quantities to simplify the formulations. The conditions specified in Equation (13) impose additional lower bounds on $\alpha_0$, the minimal singular value of the $(L-1)$-th layer's inner output. The bounds stipulated in Equations (13b) to (13d) are influenced by both the network depth $L$ and the number of gradient descent (GD) iterations $T$. In contrast, the constraint in Equation (13a) primarily depends on $T$. An initialization strategy ensuring these conditions are met is proposed in Section 5. We provide a detailed interpretation in Appendix A.2.

## 4.3 Analysis of Learning Rate Magnitude

The bounds in Equations (14a) and (14b) indicate that the learning rate $\eta$ diminishes as $L$ and $T$ increase. We argue that this requirement for a small $\eta$ is not a significant limitation; it is consistent with the NTK framework, which does not rely on large learning rates for convergence. To quantify

this, we examine the scaling of $\eta$ relative to $T, L$, and $\bar{\lambda}_{\max}$. Here, $\bar{\lambda}_{\max} = \max\{\bar{\lambda}_\ell\}, \ell \in [L]$ is the maximum constant upper bound on the singular values of the NN layers (Equation (12)). These bounds are parameters that can be directly influenced by the choice of initialization method.

First, analyzing Equation (14a), we derive the scaling of $\eta$ as $\mathcal{O}(\frac{T\bar{\lambda}_{\max}^{LT}+T^2}{((T\bar{\lambda}_{\max}^{LT})+\bar{\lambda}_{\max}^L T^3 L\bar{\lambda}_{\max}^{-2})T^6})$, where constant factors independent of $T$ and $L$ are omitted. This expression highlights that the magnitude of $\eta$ is strongly dependent on the bound $\bar{\lambda}_{\max}$. This dependence implies that the learning rate can be prevented from becoming extremely small by using a proper initialization method (such as our proposed method in Section 5) to control $\bar{\lambda}_{\max}$.

Moreover, Equation (14b) shows that $\eta$'s magnitude is highly correlated with the lower bound of $\alpha_0$ (the penultimate layer's singular value, per Section 4.1). Given the four distinct lower bounds for $\alpha_0$ derived in Equation (13), we now formulate the magnitude of $\eta$ for each respective case. First, if Equation (13c) holds, $\eta = \mathcal{O}(\exp(2T\bar{\lambda}_{\max}^L)T^{-2})$, which is a non-restrictive bound due to the exponential term. Second, if Equation (13d) holds, $\eta = \mathcal{O}((\bar{\lambda}_{\max}^{2L}T^4 + \bar{\lambda}_{\max}^{2L}(T + T\bar{\lambda}_{\max}^{LT})L\bar{\lambda}_{\max}^{-2}T^3)^{-1})$. This scales inversely with $\bar{\lambda}_{\max}$ and exponentially with $L$ and $T$. Third, if Equation (13b) holds, $\eta = \mathcal{O}(\bar{\lambda}_{\max}^{-L}T^{-3})$, which also scales inversely with $\bar{\lambda}_{\max}$ and exponentially with $L$. Finally, if Equation (13a) holds, $\eta = \mathcal{O}(\exp(\frac{2}{3}T\bar{\lambda}_{\max}^L)(\bar{\lambda}_{\max}^{2L}T^2 L\bar{\lambda}_{\max}^{-2})^{-\frac{2}{3}})$, which, similar to the first case, is a non-restrictive bound due to the exponential term.

The foregoing results indicate that a larger $\bar{\lambda}_{\max}$ correlates with a smaller learning rate $\eta$. Nevertheless, this does not result in a degradation of convergence speed. This conclusion is supported by two observations: *Theoretical Consistency*: The requirement for a small $\eta$ is permissible under NTK theory [16]. The NTK regime assumes infinitely wide networks, where convergence is achieved within a compact space around the initialization, thus obviating the need for large learning steps. *Empirical Insensitivity*: Our experimental results demonstrate that the convergence speed is robust to the learning rate. As depicted in Figure 4a, our method achieves similar convergence rates for $\eta$ across a wide range (e.g., $10^{-3}$ to $10^{-7}$).

Adopting a small learning rate is a pragmatic trade-off to avoid the requirement for an extremely wide NN. Existing analyses [3, 29] that remove the infinite-width assumption often impose a polynomial width dependency (e.g., $\mathcal{O}(N^3)$) on the sample size $N$. In our framework (Section 2), the coordinate-wise L2O treats $d$-dimensional features as independent inputs, leading to an effective sample size of $Nd$. A polynomial dependency on $Nd$ would be impractical. Therefore, we opt for the alternative constraint of a smaller learning rate, which permits a feasible network width.

## 5 Deterministic Initialization

This section introduces an initialization strategy ensuring the alignment between Math-L2O and GD (see Section 3) while also satisfying the conditions presented in Section 4. The proposed initialization strategy first establishes Math-L2O to operate as a standard GD algorithm, and then guarantees the uniform convergence of Math-L2O throughout subsequent training iterations.

### 5.1 Initialization for Alignment

Following methodology in [29], we let $C_\ell = 1$ for $\ell \in [L]$. For parameter matrices initialization $W$ (see Section 2), we randomly initialize parameter matrices of first $L-1$ layers, i.e, $\{W_1^0, \ldots, W_{L-1}^0\}$ from a standard Gaussian distribution and set the last layer's parameter matrix $W_L^0 = \mathbf{0}$. Through the $2\sigma$ activation detailed in Equation (4), it outputs a constant step size, i.e., $P_T = \mathbf{I}$. Consequently, the learning proceeds with a uniform step size of $1/\beta$ after initialization, emulating standard GD and its typical sub-linear convergence rate [38]. Moreover, this zero-initialization of $W_L^0$ ensures that initial gradients for the inner layers are all zero (as shown in Equation (7)), which serves to mitigate gradient explosion.

The condition $\alpha_0 > 0$ (see Theorem 4.3) is fulfilled by randomly sampling the initial weight matrices $\{W_k^0\}_{k=1}^{L-1}$ from a standard Gaussian distribution. This approach generally ensures full row rank for fat matrices (more columns than rows) [37]. Each matrix $W_k^0$ then undergoes QR decomposition. Non-negativity is subsequently enforced upon the elements of the resulting upper triangular factor (e.g., via its element-wise absolute value, achieved in PyTorch using its `sign` function).

## 5.2 Enhancing Singular Values for Linear Convergence of Training

Motivated by properties of minimal singular values in ReLU-Nets identified in [29], we analyze the order-gap for $\alpha_0$ between the left-hand side (LHS) and right-hand side (RHS) of the inequalities in Equation (13). To satisfy these inequalities, we propose increasing $\alpha_0$. This is achieved by applying a constant *expansion coefficient* $e \geq 1$ to the initial NN parameters $\{W_1^0, \ldots, W_{L-1}^0\}$, transforming them to $\{eW_1^0, \ldots, eW_{L-1}^0\}$. This parameter expansion scales the minimal singular value $\alpha_0$ to $e^{L-1}\alpha_0$, reflecting the cumulative impact across $L - 1$ layers. However, other terms on the RHS of Equation (13) also depend on $e$. We then establish four lemmas to demonstrate that the conditions for linear convergence, as specified in Theorem 4.3, are met for an appropriately chosen value of $e$.

First, we set the initial point to the origin, $X_0 = \mathbf{0}$, a choice commonly adopted in L2O literature [23, 34]. Then, with $C_\ell = 1$ for $\ell \in [L]$, we present four lemmas demonstrating that the conditions for linear convergence (see Theorem 4.3) are satisfied for an appropriately chosen constant $e$. The lemmas indicate that a larger $e$ is required as the number of optimization steps ($T$) increases. Specifically, Lemma 5.2 establishes that $e$ scales exponentially with $T$. Conversely, increasing the network depth ($L$) alleviates the need for a large $e$. The proofs are provided in Appendix B.

**Lemma 5.1.** *Assuming $X_0 = \mathbf{0}$, if $e = \Omega(T^{\frac{1}{L-1}})$, then the inequality Equation* (13a) *holds.*

**Lemma 5.2.** *If $e = \Omega(T^{\frac{3T+6}{TL-T-4L+6}})$, then the inequality Equation* (13b) *holds.*

**Lemma 5.3.** *Assuming $X_0 = \mathbf{0}$, if $e = \Omega(T^{\frac{4}{L-1}})$, then the inequality Equation* (13c) *holds.*

**Lemma 5.4.** *Assuming $X_0 = \mathbf{0}$, if $e = \Omega(T^{\frac{5}{L-1}} L^{\frac{1}{L-1}})$, then the inequality Equation* (13d) *holds.*

# 6 Empirical Evaluation

This section presents an empirical evaluation of the framework proposed in Section 3 and the theoretical results from Section 4. Experiments are conducted using Python 3.9 and PyTorch 1.12.0 on an Ubuntu 20.04 system equipped with 128GB of RAM and two NVIDIA RTX 3090 GPUs.

**Data Generation.** Due to GPU memory constraints, vectors $X \in \mathbb{R}^{5120 \times 1}$ and $Y \in \mathbb{R}^{4000 \times 1}$ for Equation (2) are generated by sampling from a standard Gaussian distribution. These represent ten problem instances with respective dimensional components of $512$ (for $X$) and $400$ (for $Y$). Following the coordinate-wise approach in [23], we formed an input feature matrix of $5120 \times 2$. This setup is equivalent to a training batch of $5120$ two-feature samples.

**Math-L2O Model Architecture.** The Math-L2O model is configured with $T = 100$ optimization steps (Equation (2)). Its architecture comprises a $L = 3$-layer DNN, as formulated in Equation (4). The first layer has an output dimension of 2. To ensure over-parameterization, the $(L - 1)$-th (i.e., second) layer's output dimension is set to $512 \times 10 = 5120$. The final layer produces a scalar output (dimension 1). Three specific model configurations are designed for ablation studies, foundational experiments, and robustness evaluations. These are detailed in Appendix C.1.

**Training and Initialization Configurations.** L2O models are trained using the Stochastic Gradient Descent (SGD) optimizer. For the $L = 3$-layer network configuration, parameters for the initial two layers ($l = 1, 2$) are initialized according to the methodology presented in Section 5.1, while parameters for the final layer ($l = 3$) are zero-initialized.

## 6.1 Training Performance

We evaluated the mean training loss in Equation (2) across all samples. Figure 4a illustrates this loss at $T = 100$, benchmarked against the standard GD objective (black dashed line). The results demonstrate that Math-L2O consistently achieves fast training convergence, corroborating the theoretical linear convergence established in Theorem 4.3.

Further, we investigated the robustness of our proposed L2O method to variations in optimization steps and learning rates (LRs). Models corresponding to different step/LR configurations are trained for 400 epochs. Figure 4b presents the training objectives for these configurations, benchmarked against standard GD (black dashed line). In contrast to the instability observed for Math-L2O [23]

and LISTA-CPSS [7] under certain settings (Figure 2), the consistent convergence across all tested configurations in Figure 4b demonstrates the robustness of our proposed L2O approach.

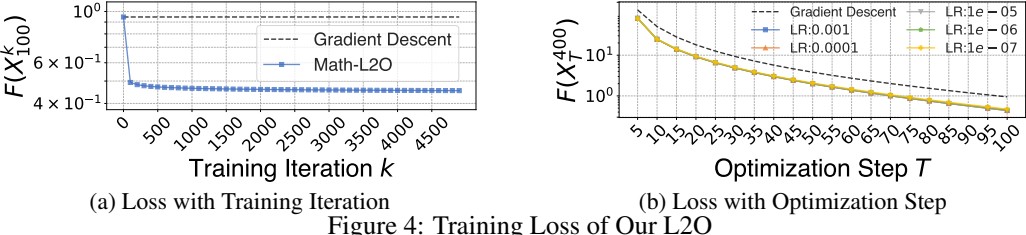

(a) Loss with Training Iteration      (b) Loss with Optimization Step

Figure 4: Training Loss of Our L2O

Moreover, we evaluate the inference performance of our framework against baseline methods. Experimental results (in Appendix C.4) demonstrate the framework's robustness to hyperparameters.

## 6.2    Ablation Studies for Learning Rate $\eta$ and Expansion Coefficient $e$

We conduct ablation studies to assess the impact of the LR $\eta$, theoretically bounded in Equations (14a) and (14b) (Theorem 4.3), and the initialization coefficient $e$, defined in Section 5. The experimental configuration employs $T = 20$, input $X \in \mathbb{R}^{32 \times 32}$, output $Y \in \mathbb{R}^{32 \times 20}$, and a neural network width of 1024. Performance is measured by the relative improvement of the proposed L2O method over standard GD at iteration $T = 20$, calculated as $\frac{\text{obj}_{\text{GD}} - \text{obj}_{\text{L2O}}}{\text{obj}_{\text{GD}}}$. These studies further validate Corollary C.1, which establishes an inverse relationship between the viable LR $\eta$ and the coefficient $e$, implying that a larger $e$ necessitates a smaller $\eta$ to ensure convergence.

With the initialization coefficient fixed at $e = 50$, we evaluate the impact of varying the LR $\eta$ on the relative objective improvement. The results in Figure 5a demonstrate that while LRs such as $10^{-4}$ and smaller achieve convergence, $\eta = 10^{-3}$ leads to unstable behavior or divergence. This finding empirically supports the existence of an operational upper bound on the LR, consistent with the theoretical constraints outlined in Equations (14a) and (14b). Moreover, reducing the LR below this stability threshold results in slower convergence rates. This observation aligns with the implication of Theorem 4.3 that, under the specified conditions, larger permissible LRs yield faster convergence.

Fixing the LR at $\eta = 10^{-7}$, we examine the influence of the initialization coefficient $e$ on performance. The results, presented in Figure 5b, demonstrate that the relative objective improvement consistently increases with larger values of $e$. Additional results exploring different $e$ and LR combinations are deferred to Appendix C owing to space constraints. These findings validate the proposed strategies for selecting the initialization coefficient and learning rate.

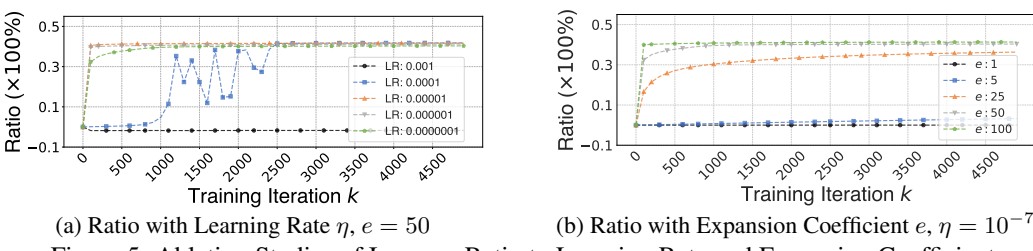

(a) Ratio with Learning Rate $\eta$, $e = 50$      (b) Ratio with Expansion Coefficient $e$, $\eta = 10^{-7}$

Figure 5: Ablation Studies of Improve Ratio to Learning Rate and Expansion Coefficient

## 7   Conclusion

This work analyzes a Learning-to-Optimize (L2O) framework that accelerates Gradient Descent (GD) through adaptive step-size learning. We theoretically prove that the L2O training enhances GD's convergence rate by linking network training bounds to GD's performance. Leveraging Neural Tangent Kernel (NTK) theory and the over-parameterization scheme via wide layers, we establish convergence guarantees for the complete L2O system. A principled initialization strategy is introduced to satisfy the theoretical requirements for these guarantees. Empirical results across various optimization problems validate our theory and demonstrate substantial practical efficacy.

## Acknowledgements

This work is supported in part by funding from CUHK (4937007, 4937008, 5501329, 5501517).

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

# A Appendix

## A.1 Details for Definitions

**General L2O.** Given $X_0$, we have the following L2O update with NN $g$ to generate $X_T$:

$$X_t = X_{t-1} + g(W_1, W_2, \ldots, W_L, X_{t-1}, \nabla F(X_{t-1})), t \in [T]. \tag{15}$$

**Concatenation of $N$ Problems.** For $t \in [T]$, we make the following denotations to represent the concatenation of $N$ samples (each is a unique optimization problem):

$$\mathbf{M} := \begin{bmatrix} \mathbf{M}_1 & & \\ & \cdots & \\ & & \mathbf{M}_N \end{bmatrix}, X_t := [x_{1,t}^\top | x_{2,t}^\top | \ldots | x_{N,t}^\top]^\top, Y := [y_1^\top | y_2^\top | \ldots | y_N^\top]^\top.$$

$X_t$ and $Y$ are still column vectors since we take the coordinate-wise setting from [23].

## A.2 Detailed Interpretation of Quantities in Theorem 4.3

We elaborate on the quantities introduced in Theorem 4.3. Our notational convention is as follows: subscripts $T$ and $k$ identify constant terms that are dependent on the total steps $T$ and the training iteration $k$. Conversely, indices $j$ and $t$ (appearing as superscripts or subscripts) are used to reference scalar-valued functions at a specific step $t$ or index $j$.

- $\bar{\lambda}_\ell$ is a positive constant upper bound for each $\ell$-th layer in NN $g_W$ (see Section 2), which is constructed in the proof of Theorem 4.3.
- $\Theta_L$ is a positive constant w.r.t. $\bar{\lambda}_\ell$.
- Denote $\bar{\lambda}_{\min}, \bar{\lambda}_{\max} := \min\{\bar{\lambda}_\ell\}, \max\{\bar{\lambda}_\ell\}, \ell \in [L]$, $\Theta_L$ is lower and upper bounded by $\Omega(\bar{\lambda}_{\min}^L)$ and $\mathcal{O}(\bar{\lambda}_{\max}^L)$, respectively. Moreover, $\Theta_L^{-1}$ is $\Omega(\bar{\lambda}_{\max}^{-L})$ and $\mathcal{O}(\bar{\lambda}_{\min}^{-L})$.
- $\Phi_j$ is a scalar-valued function w.r.t. step $j$. The constant coefficients are given by initial point $X_0$, coefficient matrix $\mathbf{M}$, and coefficient vector $Y$ from problem defined in Equation (2). $\beta$ is the smoothness extent of objective. We use two denotations, $j$ and $t$, for step, which are used to formulate different computations in formulations. This formulation is derived by the upper bound relaxation of $L_2$-norm of gradient at $X_0$. $\Phi_j$ is $\mathcal{O}(j)$ and $\Omega(j)$.
- $\Lambda_j$ is a scalar-valued function w.r.t. step $j$, which is identical to those in $\Phi_j$. $\Lambda_j$ is $\mathcal{O}(j^2)$ and $\Omega(j^2)$.
- $S_{\Lambda,T}$ and $S_{\bar{\lambda},L}$ are positive constants, which represents the summation of $\Lambda$ of $T$ steps and summation of $\bar{\lambda}$ of $L$-th NN layers, respectively.
- $S_{\Lambda,T}$ is used in the demonstration for Lemma 4.1 (bound of gradient of NN training), line 625, page 22. The proof is achieved by upper bound relaxation of $L_2$-norm. $S_{\bar{\lambda},L}$ is used in Theorem 4.3 and related auxiliary lemmas. $S_{\Lambda,T}$ is $\mathcal{O}(T^3)$ and $\Omega(T^3)$. $S_{\Lambda,T}^{-1}$ is $\mathcal{O}(T^{-3})$ and $\Omega(T^{-3})$. Denote $\bar{\lambda}_{\min}, \bar{\lambda}_{\max} = \min\{\bar{\lambda}_\ell\}, \max\{\bar{\lambda}_\ell\}, \ell \in [L]$, $S_{\bar{\lambda},L}$ is $\Omega(L\bar{\lambda}_{\max}^{-2})$ and $\mathcal{O}(L\bar{\lambda}_{\min}^{-2})$. Moreover, $S_{\bar{\lambda},L}^{-1}$ is $\Omega(L^{-1}\bar{\lambda}_{\max}^2)$ and $\mathcal{O}(L^{-1}\bar{\lambda}_{\min}^2)$.
- $\zeta_1$ and $\zeta_2$ are two positive constants scale linearly w.r.t., $X_0$, $\mathbf{M}$, $Y$, $T$, and $\beta$. $\zeta_1$ and $\zeta_2$ are both $\Omega(T)$ and $\mathcal{O}(T)$.
- $\zeta_1$ and $\zeta_2$: Two positive constants scale linearly w.r.t., $X_0$, $\mathbf{M}$, $Y$, $T$, and $\beta$. $\zeta_1$ and $\zeta_2$ are both $\Omega(T)$ and $\mathcal{O}(T)$.
- $\delta_1^t$: A scalar-valued function w.r.t. step $t$. The constant coefficients are $\Theta_L$, $\Phi_j$, and $\Lambda_s$, where $s$ denotes an step. Denote $\bar{\lambda}_{\min}, \bar{\lambda}_{\max} = \min\{\bar{\lambda}_\ell\}, \max\{\bar{\lambda}_\ell\}, \ell \in [L]$, $\delta_1^t$ is $\Omega(t\bar{\lambda}_{\min}^{Lt})$ and $\mathcal{O}(t\bar{\lambda}_{\max}^{Lt})$.
- $\delta_2$: Positive constant scales with $T$. Denote $\bar{\lambda}_{\min}, \bar{\lambda}_{\max} = \min\{\bar{\lambda}_\ell\}, \max\{\bar{\lambda}_\ell\}, \ell \in [L]$, $\delta_2$ is $\Omega(T\bar{\lambda}_{\min}^{LT})$ and $\mathcal{O}(T\bar{\lambda}_{\max}^{LT})$. Moreover, $\delta_2^{-1}$ is $\Omega(T\bar{\lambda}_{\max}^{-LT})$ and $\mathcal{O}(T\bar{\lambda}_{\min}^{-LT})$.
- $\delta_3$: Positive constant scales linearly w.r.t., $X_0$, $\mathbf{M}$, $Y$, $T$, and $\beta$. $\delta_3$ is both $\Omega(T)$ and $\mathcal{O}(T)$.
- $\delta_4$: Denote $\bar{\lambda}_{\min}, \bar{\lambda}_{\max} = \min\{\bar{\lambda}_\ell\}, \max\{\bar{\lambda}_\ell\}, \ell \in [L]$, $\delta_4$ is $\Omega(\exp(-T\bar{\lambda}_{\max}^L))$ and $\mathcal{O}(\exp(-T\bar{\lambda}_{\min}^L))$. Moreover, $\delta_4^{-1}$ is $\mathcal{O}(\exp(T\bar{\lambda}_{\max}^L))$ and $\Omega(\exp(T\bar{\lambda}_{\min}^L))$.

## A.3 Derivative of General L2O

In this section, we derive a general framework for any L2O models by the chain rule, which gives us a complete workflow of each component in the derivatives within the chain. Then, we apply it to the Math-L2O framework [23] to get the formulation for the L2O model defined in Equation (4).

Due to the chain rule, we derive the following general formulation of the derivative in L2O model:

$$\frac{\partial F(X_T)}{\partial W_\ell} = \frac{\partial F(X_T)}{\partial X_T}\left(\frac{\partial X_T}{\partial X_{T-1}}\frac{\partial X_{T-1}}{\partial W_\ell} + \frac{\partial X_T}{\partial G_{L,t}}\frac{\partial G_{L,t}}{\partial W_\ell}\right).$$

We then calculate each term in the right-hand side (RHS) in the above formulation. First, we calculate $\frac{\partial X_{T-1}}{\partial W_\ell}$ as:

$$\frac{\partial X_{T-1}}{\partial W_\ell} = \frac{\partial X_{T-1}}{\partial X_{T-2}}\frac{\partial X_{T-2}}{\partial W_\ell} + \frac{\partial X_{T-1}}{\partial G_{L,T-1}}\frac{\partial G_{L,T-1}}{\partial W_\ell}.$$

Thus, we can iteratively derive the gradient until $X_1$. After rearranging terms, we have the following complete formulation of $\frac{\partial F}{\partial W_\ell}$:

$$\frac{\partial F(X_T)}{\partial W_\ell} = \frac{\partial F(X_T)}{\partial X_T}\Big(\sum_{t=1}^{T}(\prod_{j=T}^{t+1}\frac{\partial X_j}{\partial X_{j-1}})\frac{\partial X_t}{\partial G_{L,t}}\frac{\partial G_{L,t}}{\partial W_\ell}\Big). \tag{16}$$

We note that $\frac{\partial X_j}{\partial X_{j-1}}$ relies on different implementations. For example, for general L2O model that the update in each step is directly the output of neural networks (NNs), we have $\frac{\partial X_j}{\partial X_{j-1}} := \mathbf{I} + \frac{\partial G_{L,j}}{\partial X_{j-1}}$. Then, Equation (16) is derived by:

$$\frac{\partial F}{\partial W_\ell} = \frac{\partial F(X_T)}{\partial X_T}\Big(\sum_{t=1}^{T}(\prod_{j=T}^{t+1}(\mathbf{I} + \frac{\partial G_{L,j}}{\partial X_{j-1}}))\frac{\partial X_T}{\partial G_{L,t}}\frac{\partial G_{L,t}}{\partial W_\ell}\Big). \tag{17}$$

$\frac{\partial G_{L,j}}{\partial X_{j-1}}$ depends on specific implementation of NNs. Liu et al. [23] simplify $\frac{\partial G_{L,j}}{\partial X_{j-1}}$ by detaching input tensor from the back-propagation process, which truncate the branches in the chain from $F(X_T)$ to $W_\ell$. The detaching operation yields simpler $\frac{\partial X_j}{\partial X_{j-1}}$. As will be introduced in the following sections, $\frac{\partial X_j}{\partial X_{j-1}}$ depends only on NN's output.

Further, the definition of $\frac{\partial X_T}{\partial G_{L,t}}$ is framework-dependent. In the general L2O model, $\frac{\partial X_T}{\partial G_{L,t}} := \mathbf{I}$, whereas in Math-L2O [23], it is defined based on the FISTA algorithm [4]. Subsequently, we perform a layer-by-layer computation for each derivative $\frac{\partial G_{L,j}}{\partial X_{t-1}}$ and $\frac{\partial G_{L,t}}{\partial W_\ell}$.

First, we derive $\frac{\partial G_{L,t}}{\partial G_{L-1,t}}$ by:

$$\frac{\partial G_{L,t}}{\partial G_{L-1,t}} = \begin{cases} \nabla\,\mathrm{ReLU}(G_{L-1,t})W_\ell & \ell \in [L-1], \\ \nabla 2\sigma(G_{\ell,t})W_\ell & \ell = L. \end{cases}$$

For simplification, we use $\nabla\,\mathrm{ReLU}$ and $\nabla 2\sigma$ to represent derivatives $\nabla\,\mathrm{ReLU}(G_{L-1,t})$ and $\nabla 2\sigma(G_{\ell,t})$, respectively, which are corresponding diagonal matrices of coordinate-wise activation function's derivatives. Next, $\frac{\partial G_{L,t}}{\partial X_{t-1}}$ is given by:

$$\frac{\partial G_{L,j}}{\partial X_{T-1}} = (\prod_{\ell=L}^{2}\frac{\partial G_{l,j}}{\partial G_{l,j-1}})\frac{\partial G_{1,j-1}}{\partial X_{T-1}} = \nabla 2\sigma w_L(\prod_{\ell=L-1}^{2}\nabla\,\mathrm{ReLU}W_\ell)[\mathbf{I}, \mathbf{H}^\top], \tag{18}$$

where $\mathbf{H} := \mathbf{M}^\top\mathbf{M}$ denotes the Hessian matrix of the loss function in Equation (2).

Second, $\frac{\partial G_{L,t}}{\partial W_\ell}$ is given by:

$$\begin{aligned}
\frac{\partial G_{l,t}}{\partial W_\ell} &= (\prod_{j=L}^{\ell+1}\frac{\partial G_{j,t}}{\partial G_{j-1,t}})\frac{\partial G_{l,t}}{\partial W_\ell} \\
&= \begin{cases} \nabla 2\sigma w_L(\prod_{j=L-1}^{\ell+1}\nabla\,\mathrm{ReLU}W_j)\nabla\,\mathrm{ReLU}(\mathbf{I}_{n_\ell} \otimes G_{\ell-1,t}{}^\top) & \ell \in [L-1], \\ \nabla 2\sigma(\mathbf{I}_{n_\ell} \otimes G_{L-1,t}{}^\top) & \ell = L, \end{cases}
\end{aligned} \tag{19}$$

where $\mathbf{I}_{n_\ell} \in \mathbb{R}^{n_\ell \times n_\ell}$, $\otimes$ denotes Kronecker Product, and $\mathbf{I}_{n_\ell} \otimes G_{\ell-1,t}{}^\top \in \mathbb{R}^{n_\ell \times n_\ell n_{\ell-1}}$.

Substituting Equation (18) and Equation (19) into Equation (17) yields following final derivative formulation of general L2O model:

$$
\begin{aligned}
&\frac{\partial F}{\partial W_\ell} \\
&= \frac{\partial F(X_T)}{\partial X_T}\left(\sum_{t=1}^{T}\left(\prod_{j=T}^{t+1}(\mathbf{I}+\frac{\partial G_{L,j}}{\partial X_{j-1}})\right)\frac{\partial X_T}{\partial G_{L,t}}\frac{\partial G_{L,t}}{\partial W_\ell}\right),
\end{aligned}
$$

$$
=\begin{cases}
\mathbf{K}_{n_\ell,n_{\ell-1}}\Bigg((X_T^{k\top}\mathbf{M}^\top-Y^\top)\mathbf{M} \\
\qquad\Big(\sum_{t=1}^{T}\big(\mathbf{I}+\nabla 2\sigma w_L(\prod_{\ell=L-1}^{2}\nabla\mathrm{ReLU}W_\ell)[\mathbf{I},\mathbf{H}^\top]\big)^{T-t} \\
\qquad\quad \nabla 2\sigma w_L^\top(\prod_{j=L-1}^{\ell+1}\nabla\mathrm{ReLU}W_j)\nabla\mathrm{ReLU}(\mathbf{I}_{n_\ell}\otimes G_{\ell-1,t}{}^\top)\Big)\Bigg)^\top & \ell\in[L-1], \\[1em]
\mathbf{K}_{n_\ell,n_{\ell-1}}\Bigg((X_T^{k\top}\mathbf{M}^\top-Y^\top)\mathbf{M} \\
\qquad\Big(\sum_{t=1}^{T}\big(\mathbf{I}+\nabla 2\sigma w_L(\prod_{\ell=L-1}^{2}\nabla\mathrm{ReLU}W_\ell)[\mathbf{I},\mathbf{H}^\top]\big)^{T-t} \\
\qquad\quad \nabla 2\sigma(\mathbf{I}_{n_\ell}\otimes G_{L-1,t}{}^\top)\Big)\Bigg)^\top & l=L,
\end{cases}
$$

(20)

where $\mathbf{K}_{n_\ell,n_{\ell-1}}$ denotes a commutation matrix, which is a $n_\ell*n_{\ell-1}\times n_\ell*n_{\ell-1}$ permutation matrix that swaps rows and columns in the vectorization process.

### A.4 Derivative of Coordinate-Wise Math-L2O

Based on the results in Appendix A.3, in this section, we construct the gradient formulations for Math-L2O model. We present the results in Equation (7) and Equation (8).

As defined in Equations (3) and (4), Math-L2O [23] learns to choose hyperparameters of existing non-learning algorithms [23, 34]. Suppose $P_i\in\mathbb{R}^{N*d}, i\in[0,\ldots,T]$ is the hyperparameter vector generated by NNs. Suppose $X_{-1}:=X_0$, based on Equation 3, the solution update process from the initial step is defined by:

$$
\begin{aligned}
X_1 &= X_0 - \tfrac{1}{\beta}P_1\odot\nabla F(X_0), \\
X_2 &= X_1 - \tfrac{1}{\beta}P_2\odot\nabla F(X_1), \\
&\ldots, \\
X_T &= X_{T-1} - \tfrac{1}{\beta}P_T\odot\nabla F(X_{T-1}),
\end{aligned}
\tag{21}
$$

We re-use the definition in Section 2 that defines $\mathcal{D}(\cdot)$ as the operator that constructs a diagonal matrix from a vector, we calculate the following one-line and linear-like formulation of $X_T$ with $X_0$:

$$
X_T = \prod_{t=T}^{1}(\mathbf{I}-\tfrac{1}{\beta}\mathcal{D}(P_t)\mathbf{M}^\top\mathbf{M})X_0 + \tfrac{1}{\beta}\sum_{t=1}^{T}\prod_{s=T}^{t+1}(\mathbf{I}-\tfrac{1}{\beta}\mathcal{D}(P_s)\mathbf{M}^\top\mathbf{M})\mathcal{D}(P_t)\mathbf{M}^\top Y. \quad (22)
$$

Given that $P_t$ is generated by a non-linear neural network with $X_{t-1}$ as input, the resulting system dynamics are inherently non-linear. Consequently, this system cannot be formulated as the aforementioned linear dynamic system. Moreover, we note that for non-smooth problems, the uncertain sub-gradient can be replaced by the gradient map to obtain analogous formulations [34].

Due to the above computational graph in Figure 1, the gradient of $X_t$ comes from $X_{t-1}$ and $P_t$, which yields the following framework of each layer's derivative (Equation (6)):

$$
\frac{\partial F}{\partial W_\ell} = \frac{\partial F(X_T)}{\partial X_T}\left(\sum_{t=1}^{T}\left(\prod_{j=T}^{t+1}\frac{\partial X_j}{\partial X_{j-1}}\right)\frac{\partial X_t}{\partial P_t}\frac{\partial P_t}{\partial W_\ell}\right). \tag{23}
$$

We obtain the above equation by counting the number of formulations from $F$ to $W_\ell$. From the Figure 1, we conclude that each timestamp $t$ leads to the gradient of $\frac{\partial X_T}{\partial X_{T-1}}$. Thus, there are $\prod_{j=T}^{t+1}\frac{\partial X_j}{\partial X_{j-1}}$ blocks of formulation in total.

We start with deriving the formulation of gradient w.r.t. the GD algorithm, which yields the gradient of $\frac{\partial X_T}{\partial P_T}$. Due to the GD formulation in Equation (21), we derive $\frac{\partial X_t}{\partial X_{t-1}}$ as:

$$
\begin{aligned}
\frac{\partial X_t}{\partial X_{t-1}} =& \mathbf{I}_d - \frac{1}{\beta}\frac{\partial\Big(P_t\odot\nabla F(X_{t-1})\Big)}{\partial X_{t-1}} \\
=& \mathbf{I}_d - \frac{1}{\beta}\frac{\partial P_t\odot\big(\mathbf{M}^\top(\mathbf{M}X_{t-1}-Y)\big)}{\partial X_{t-1}}, \\
=& \mathbf{I}_d - \frac{1}{\beta}\mathcal{D}(P_t)\mathbf{M}^\top\mathbf{M} - \frac{1}{\beta}\frac{\partial P_t\odot\big(\mathbf{M}^\top(\mathbf{M}X_{t-1}-Y)\big)}{\partial P_t}\frac{\partial P_t}{\partial X_{t-1}}, \\
=& \mathbf{I}_d - \frac{1}{\beta}\mathcal{D}(P_t)\mathbf{M}^\top\mathbf{M} - \frac{1}{\beta}\mathcal{D}\big(\mathbf{M}^\top(\mathbf{M}X_{t-1}-Y)\big)\frac{\partial P_t}{\partial X_{t-1}}.
\end{aligned}
\tag{24}
$$

Next, we calculate $\frac{\partial P_t}{\partial X_{t-1}}$. Similarly, we derive $\frac{\partial\mathrm{vec}(G_{L,t})}{\partial W_\ell}$ and each $\frac{\partial\mathrm{vec}(G_{L,j})}{\partial X_{j-1}}$ of Math-L2O layer-by-layer. $\frac{\partial\mathrm{vec}(G_{L,t})}{\partial\mathrm{vec}(G_{L-1,t})}$ in Math-L2O is similar to Equation (19). We calculate:

$$
\begin{cases}
\frac{\partial P_t}{\partial W_\ell} = \mathcal{D}\big(P_t\odot(1-P_t/2)\big)(\mathbf{I}_d\otimes W_L)\prod_{j=L-1}^{\ell+1}\mathbf{D}_{j,t}\mathbf{I}_d\otimes W_j\mathbf{I}_{n_\ell}\otimes G_{\ell-1,t}{}^\top & \ell\in[L-1], \\
\frac{\partial P_t}{\partial W_L} = \mathcal{D}\big(P_t\odot(1-P_t/2)\big)G_{L-1,t}{}^\top & \ell=L.
\end{cases}
\tag{25}
$$

Similarly, we calculate the following derivative of output of Math-L2O w.r.t. it input at step $t$:

$$
\frac{\partial P_t}{\partial X_{t-1}} = \mathcal{D}\big(P_t\odot(1-P_t/2)\big)W_L(\prod_{\ell=L-1}^{2}\mathbf{D}_{\ell,t}W_\ell)[\mathbf{I},\mathbf{H}^\top]^\top.
\tag{26}
$$

Substituting Equation (26) into Equation (24) yields $\frac{\partial X_t}{\partial X_{t-1}}$:

$$
\begin{aligned}
\frac{\partial X_t}{\partial X_{t-1}} =& \mathbf{I}_d - \frac{1}{\beta}\mathcal{D}(P_t)\mathbf{M}^\top\mathbf{M} \\
& - \frac{1}{\beta}\mathcal{D}\big(\mathbf{M}^\top(\mathbf{M}X_{t-1}-Y)\big)\mathcal{D}\big(P_t\odot(1-P_t/2)\big)W_L(\prod_{\ell=L-1}^{2}\mathbf{D}_{\ell,t}W_\ell)[\mathbf{I},\mathbf{H}^\top]^\top.
\end{aligned}
\tag{27}
$$

We note that in [23], the gradient formulations are simplified in the implementation by detaching the input feature from the computational graph. Thus, we can eliminate the complicated last term in the above formulation, which leads to the following compact version:

$$
\frac{\partial X_t}{\partial X_{t-1}} = \mathbf{I}_d - \frac{1}{\beta}\mathcal{D}(P_t)\mathbf{M}^\top\mathbf{M}.
\tag{28}
$$

In this paper, we take the gradient formulation in Equation (28).

Next, we calculate the $\frac{\partial X_t}{\partial P_t}$ component in Equation (23). We calculate the derivative of GD's output w.r.t. its input hyperparameter $P$ (generated by NNs) as:

$$
\frac{\partial X_t}{\partial P_t} = -\frac{1}{\beta}\mathcal{D}(\nabla F(X_{t-1})) = -\frac{1}{\beta}\mathcal{D}\big(\mathbf{M}^\top(\mathbf{M}X_{t-1}-Y)\big),
\tag{29}
$$

where $\nabla F(X_{t-1}) := \mathbf{M}^\top(\mathbf{M}X_{t-1}-Y)$ is the first-order derivative of the objective in Equation 1.

Substituting Equation (25), Equation (28), and Equation (29) into Equation (23) yields the final derivative of all layers' parameters.

First, for $\ell=L$, since there is no cumulative gradients of later layers, Equation 8 is directly calculated by:

$$
\begin{aligned}
\frac{\partial F}{\partial W_L} = & -\frac{1}{\beta}\sum_{t=1}^{T}\big(\mathbf{M}^\top(\mathbf{M}X_T-Y)\big)^\top\big(\prod_{j=T}^{t+1}\mathbf{I} - \frac{1}{\beta}\mathcal{D}(P_j)\mathbf{M}^\top\mathbf{M}\big) \\
& \mathcal{D}\big((\mathbf{M}^\top(\mathbf{M}X_{t-1}-Y))\big)\mathcal{D}\big(P_t\odot(1-P_t/2)\big)G_{L-1,t}{}^\top.
\end{aligned}
$$

And its transpose is given by:

$$
\begin{aligned}
\frac{\partial F}{\partial W_L}{}^\top = & -\frac{1}{\beta}\sum_{t=1}^{T}G_{L-1,t}\mathcal{D}\big(P_t\odot(1-P_t/2)\big)\mathcal{D}\big((\mathbf{M}^\top(\mathbf{M}X_{t-1}-Y))\big) \\
& \big(\prod_{j=t+1}^{T}\mathbf{I} - \frac{1}{\beta}\mathbf{M}^\top\mathbf{M}\mathcal{D}(P_j)\big)\mathbf{M}^\top(\mathbf{M}X_T-Y).
\end{aligned}
\tag{30}
$$

When $\ell \in [L-1]$, the derivative is calculated by:

$$\begin{aligned}
\frac{\partial F}{\partial W_\ell} &= \frac{\partial F(X_T)}{\partial X_T}\left(\sum_{t=1}^{T}\left(\prod_{j=T}^{t+1}\frac{\partial X_j}{\partial X_{j-1}}\right)\frac{\partial X_t}{\partial P_t}\frac{\partial P_t}{\partial W_\ell}\right),\\
&= -\tfrac{1}{\beta}\sum_{t=1}^{T}(\mathbf{M}^\top(\mathbf{M}X_T - Y))^\top\left(\prod_{j=T}^{t+1}\mathbf{I}_d - \tfrac{1}{\beta}\mathbf{M}^\top\mathbf{M}\mathcal{D}(P_j)\right)\\
&\qquad \mathcal{D}\big((\mathbf{M}^\top(\mathbf{M}X_{t-1} - Y))\big)\mathcal{D}\big(P_t \odot (1 - P_t/2)\big)\\
&\qquad (\mathbf{I}_d \otimes W_L)\prod_{j=L-1}^{\ell+1}\mathbf{D}_{j,t}\mathbf{I}_d \otimes W_j\mathbf{I}_{n_\ell} \otimes G_{\ell-1,t}^{\top}.
\end{aligned}$$

*Remark* 2. The only difference between Equation (8) and Equation (7) lies in the last term, where Equation (7) is more complicated due to the accumulated gradients from later layers.

The above two formulations are used in the next section to derive the gradient bound for each layer.

### A.5 Tools

In this section, prior to constructing the convergence bounds, we first derive several analytical tools. These tools are foundational for the convergence rate analysis and also establish key properties of the L2O models. We use superscript $k$ to denote parameters and variables at training iteration $k$, and subscript $t$ to denote the optimization step.

#### A.5.1 NN's Outputs are Bounded

First, we demonstrate that the outputs and inner outputs of NN layers within the L2O model are bounded.

**Bound** $\left\|\mathbf{I} - \tfrac{1}{\beta}\mathcal{D}(P_t^k)\mathbf{M}^\top\mathbf{M}\right\|_2, \forall k, t.$

**Lemma A.1.** *Suppose* $\|\mathbf{M}^\top\mathbf{M}\|_2 \leq \beta$ *and* $0 < P_t^k < 2$*, we have the following bound:*

$$\left\|\mathbf{I} - \tfrac{1}{\beta}\mathcal{D}(P_t^k)\mathbf{M}^\top\mathbf{M}\right\|_2 < 1. \tag{31}$$

*Proof.* Suppose eigenvalues and eigenvectors of $\mathbf{M}^\top\mathbf{M}$ are $\sigma_i$ and $v_i$, $i \in [1, \ldots, N*d]$ respectively, we calculate:

$$\tfrac{1}{\beta}\mathcal{D}(P_t^k)\mathbf{M}^\top\mathbf{M}v_i = \tfrac{\sigma_i}{\beta}\mathcal{D}(P_t^k)v_i.$$

Due to $0 < P_t^k < 2$, we have following spectral norm definition:

$$\left\|\mathbf{I} - \tfrac{1}{\beta}\mathcal{D}(P_t^k)\mathbf{M}^\top\mathbf{M}\right\|_2 = \max_{x \in \mathbb{R}^d}\frac{x^\top(\mathbf{I} - \tfrac{1}{\beta}\mathcal{D}(P_t^k)\mathbf{M}^\top\mathbf{M})x}{x^\top x}$$

Then, by taking $x = v_i$, we calculate:

$$v_i^\top(\mathbf{I} - \tfrac{1}{\beta}\mathcal{D}(P_t^k)\mathbf{M}^\top\mathbf{M})v_i = 1 - \tfrac{1}{\beta}v_i^\top\mathcal{D}(P_t^k)\mathbf{M}^\top\mathbf{M}v_i = 1 - \tfrac{\sigma_i}{\beta}v_i^\top\mathcal{D}(P_t^k)v_i \overset{①}{\leq} 1,$$

where ① is due to $0 < P_t^k < 2$. □

*Remark* 3. In our design, we ensure $0 < P_t^k < 2$ by an activation function $2\sigma$ at the output layer.

**Bound** $\|\mathcal{D}(P_t^k)\|_2, \forall k, t.$ Similar to the bound of $\left\|\mathbf{I} - \tfrac{1}{\beta}\mathcal{D}(P_t^k)\mathbf{M}^\top\mathbf{M}\right\|_2, \forall k, t$, due to the Sigmoid function, we directly have:

**Lemma A.2.** *Suppose* $0 < P_t^k < 2$*, we have the following bound:*

$$\|\mathcal{D}(P_t^k)\|_2 < 2. \tag{32}$$

*Proof.* Since $\mathcal{D}$ is the diagonalization operation and $0 < P_t^k < 2$, we directly have $\|\mathcal{D}(P_t^k)\|_2 < 2$. □

Besides, we can derive another bound from the Lipschitz property for the Sigmoid activation function:

$$
\begin{aligned}
\|\mathcal{D}(P_t^k)\|_2 =& \|2\sigma(\text{ReLU}(\text{ReLU}([X_{t-1}^k, \mathbf{M}^\top(\mathbf{M}X_{t-1}^k - Y)]W_1^{k^\top}) \cdots W_{L-1}^{k}{}^\top)W_L^k{}^\top)\|_\infty, \\
&\overset{\text{\textcircled{1}}}{\leq} \tfrac{1}{2}\|[X_{t-1}^k, \mathbf{M}^\top(\mathbf{M}X_{t-1}^k - Y)]\|_2 \textstyle\prod_{s=1}^{L-1}\|W_s^k\|_2 + 1, \\
&\overset{\text{\textcircled{2}}}{\leq} \tfrac{1}{2}(\|X_t^k\|_2 + \|\mathbf{M}^\top(\mathbf{M}X_t^k - Y)\|_2)\textstyle\prod_{s=1}^{L-1}\|W_s^k\|_2 + 1.
\end{aligned}
\tag{33}
$$

\textcircled{1} is from equation (17), Lemma 4.2 of [30]. \textcircled{2} is from triangle inequality.

*Remark* 4. In contrast to the Lipschitz continuous property of ReLU, the aforementioned bound associated with the Sigmoid function prevents the derivation of meaningful numerical results. To analyze the convergence rate of Gradient Descent (GD), a tighter bound on the neural network's output is required. One potential alternative is the convex cone defined by $W_L^k$ for the last hidden layer. However, such a cone spans an unbounded space for the set of learnable parameters.

**Bound Semi-Smoothness of NN's Output, i.e., $\|\mathcal{D}(P_t^{k+1}) - \mathcal{D}(P_t^k)\|_2$, $\forall k, t$.** Since our L2O model is a coordinate-wise model [23], suppose $P_i = \alpha_i(P_t^{k+1})_i + (1-\alpha_i)(P_t^k)_i$, $\alpha_p \in [0,1]$, based on Mean Value Theorem, we have $(\mathcal{D}(P_t^{k+1}) - \mathcal{D}(P_t^k))_i = \frac{\partial F}{P_i}((P_t^{k+1})_i - (P_t^k)_i)$. Thus, we bound $\|\mathcal{D}(P_t^{k+1}) - \mathcal{D}(P_t^k)\|_2$ by the following lemma:

**Lemma A.3.** *Denote $j \in [L]$, for some $\bar{\lambda}_j \in \mathbb{R}$, we assume $\|W_j^{k+1}\|_2 \leq \bar{\lambda}_j$. Using quantities from Equation* (12)*, we have:*

$$
\begin{aligned}
&\|\mathcal{D}(P_t^{k+1}) - \mathcal{D}(P_t^k)\|_2 \\
\leq& \tfrac{1}{2}(1+\beta)\|X_{t-1}^{k+1} - X_{t-1}^k\|_2 \Theta_L \\
&+ \tfrac{1}{2}(\|X_{t-1}^k\|_2 + \|\mathbf{M}^\top(\mathbf{M}X_{t-1}^k - Y)\|_2)\Theta_L \textstyle\sum_{\ell=1}^L \bar{\lambda}_\ell^{-1}\|W_\ell^{k+1} - W_\ell^k\|_2.
\end{aligned}
\tag{34}
$$

*Remark* 5. The above lemma shows the output of NN is a "mixed" Lipschitz continuous on input feature and learnable parameters. The first term illustrates the Lipschitz property on input feature. The second term can be regarded as a Lipschitz property on learnable parameters with a stable input feature.

*Proof.* Due to Mean Value Theorem, we have:

$$\|\mathcal{D}(P_t^{k+1}) - \mathcal{D}(P_t^k)\|_2$$

$$=\|\mathcal{D}(2\sigma(\mathrm{ReLU}(\cdots\mathrm{ReLU}([X_{t-1}^{k+1}, \mathbf{M}^\top(\mathbf{M}X_{t-1}^{k+1} - Y)]W_1^{k+1\top})\cdots W_{L-1}^{k+1\top})W_L^{k+1}))$$

$$\quad - \mathcal{D}(2\sigma(\mathrm{ReLU}(\cdots\mathrm{ReLU}([X_{t-1}^{k}, \mathbf{M}^\top(\mathbf{M}X_{t-1}^{k} - Y)]W_1^{k\top})\cdots W_{L-1}^{k\top})W_L^{k}))\|_2,$$

$$\leq (2\sigma(P_i)(1-\sigma(P_i)))_{\max}$$

$$\quad \| \mathrm{ReLU}(\cdots\mathrm{ReLU}([X_{t-1}^{k+1}, \mathbf{M}^\top(\mathbf{M}X_{t-1}^{k+1} - Y)]W_1^{k+1\top})\cdots W_{L-1}^{k+1\top})W_L^{k+1}$$

$$\quad - \mathrm{ReLU}(\cdots\mathrm{ReLU}([X_{t-1}^{k}, \mathbf{M}^\top(\mathbf{M}X_{t-1}^{k} - Y)]W_1^{k\top})\cdots W_{L-1}^{k\top})W_L^{k}\|_\infty,$$

$$\leq \tfrac{1}{2}\| \mathrm{ReLU}(\mathrm{ReLU}([X_{t-1}^{k+1}, \mathbf{M}^\top(\mathbf{M}X_{t-1}^{k+1} - Y)]W_1^{k+1\top})\cdots W_{L-1}^{k+1\top})W_L^{k+1}$$

$$\quad - \mathrm{ReLU}(\mathrm{ReLU}([X_{t-1}^{k}, \mathbf{M}^\top(\mathbf{M}X_{t-1}^{k} - Y)]W_1^{k\top})\cdots W_{L-1}^{k\top})W_L^{k}\|_\infty,$$

$$\overset{①}{\leq} \tfrac{1}{2}\| \mathrm{ReLU}(\cdots\mathrm{ReLU}([X_{t-1}^{k+1}, \mathbf{M}^\top(\mathbf{M}X_{t-1}^{k+1} - Y)]W_1^{k+1\top})\cdots W_{L-1}^{k+1\top})$$

$$\quad - \mathrm{ReLU}(\cdots\mathrm{ReLU}([X_{t-1}^{k}, \mathbf{M}^\top(\mathbf{M}X_{t-1}^{k} - Y)]W_1^{k\top})\cdots W_{L-1}^{k\top})\|_\infty\|W_L^{k+1}\|_2$$

$$\quad + \tfrac{1}{2}\| \mathrm{ReLU}(\cdots\mathrm{ReLU}([X_{t-1}^{k}, \mathbf{M}^\top(\mathbf{M}X_{t-1}^{k} - Y)])W_{L-1}^{k\top})\|_2\|W_L^{k+1} - W_L^{k}\|_2,$$

$$\overset{②}{\leq} \tfrac{1}{2}\| \mathrm{ReLU}(\cdots\mathrm{ReLU}([X_{t-1}^{k+1}, \mathbf{M}^\top(\mathbf{M}X_{t-1}^{k+1} - Y)]W_1^{k+1\top})\cdots W_{L-2}^{k+1\top})W_{L-1}^{k+1\top}$$

$$\quad - \mathrm{ReLU}(\cdots\mathrm{ReLU}([X_{t-1}^{k}, \mathbf{M}^\top(\mathbf{M}X_{t-1}^{k} - Y)]W_1^{k\top})\cdots W_{L-2}^{k\top})W_{L-1}^{k\top}\|_\infty\bar{\lambda}_L$$

$$\quad + \tfrac{1}{2}\|[X_{t-1}^{k}, \mathbf{M}^\top(\mathbf{M}X_{t-1}^{k} - Y)]\|_2\prod_{j=1}^{L-1}\bar{\lambda}_j\|W_L^{k+1} - W_L^{k}\|_2,$$

$$\overset{③}{\leq} \tfrac{1}{2}\| \mathrm{ReLU}(\cdots\mathrm{ReLU}([X_{t-1}^{k+1}, \mathbf{M}^\top(\mathbf{M}X_{t-1}^{k+1} - Y)]W_1^{k+1\top})\cdots W_{L-2}^{k+1\top})$$

$$\quad - \mathrm{ReLU}(\cdots\mathrm{ReLU}([X_{t-1}^{k}, \mathbf{M}^\top(\mathbf{M}X_{t-1}^{k} - Y)]W_1^{k\top})\cdots W_{L-2}^{k\top})\|_\infty\bar{\lambda}_{L-1}\bar{\lambda}_L$$

$$\quad + \tfrac{1}{2}\|[X_{t-1}^{k}, \mathbf{M}^\top(\mathbf{M}X_{t-1}^{k} - Y)]\|_2\prod_{j=1}^{L-1}\bar{\lambda}_j\|W_L^{k+1} - W_L^{k}\|_2,$$

$$\quad + \tfrac{1}{2}\|[X_{t-1}^{k}, \mathbf{M}^\top(\mathbf{M}X_{t-1}^{k} - Y)]\|_2\prod_{j=1}^{L-2}\bar{\lambda}_j\bar{\lambda}_L\|W_{L-1}^{k+1} - W_{L-1}^{k}\|_2,$$

$$\overset{④}{=} \tfrac{1}{2}\| \mathrm{ReLU}(\cdots\mathrm{ReLU}([X_{t-1}^{k+1}, \mathbf{M}^\top(\mathbf{M}X_{t-1}^{k+1} - Y)]W_1^{k+1\top})\cdots W_{L-2}^{k+1\top})$$

$$\quad - \mathrm{ReLU}(\cdots\mathrm{ReLU}([X_{t-1}^{k}, \mathbf{M}^\top(\mathbf{M}X_{t-1}^{k} - Y)]W_1^{k\top})\cdots W_{L-2}^{k\top})\|_\infty\bar{\lambda}_{L-1}\bar{\lambda}_L$$

$$\quad + \tfrac{1}{2}\|[X_{t-1}^{k}, \mathbf{M}^\top(\mathbf{M}X_{t-1}^{k} - Y)]\|_2\Theta_L(\bar{\lambda}_L^{-1}\|W_L^{k+1} - W_L^{k}\|_2 + \bar{\lambda}_{L-1}^{-1}\|W_{L-1}^{k+1} - W_{L-1}^{k}\|_2),$$

$$\cdots,$$

$$\overset{⑤}{\leq} \tfrac{1}{2}\|[X_{t-1}^{k+1}, \mathbf{M}^\top(\mathbf{M}X_{t-1}^{k+1} - Y)] - [X_{t-1}^{k}, \mathbf{M}^\top(\mathbf{M}X_{t-1}^{k} - Y)]\|_2\Theta_L$$

$$\quad + \tfrac{1}{2}\|[X_{t-1}^{k}, \mathbf{M}^\top(\mathbf{M}X_{t-1}^{k} - Y)]\|_2\Theta_L\left(\sum_{\ell=1}^{L}\bar{\lambda}_\ell^{-1}\|W_\ell^{k+1} - W_\ell^{k}\|_2\right),$$

$$\overset{⑥}{\leq} \tfrac{1}{2}(1 + \beta)\|X_{t-1}^{k+1} - X_{t-1}^{k}\|_2\Theta_L$$

$$\quad + \tfrac{1}{2}(\|X_{t-1}^{k}\|_2 + \|\mathbf{M}^\top(\mathbf{M}X_{t-1}^{k} - Y)\|_2)\Theta_L\left(\sum_{\ell=1}^{L}\bar{\lambda}_\ell^{-1}\|W_\ell^{k+1} - W_\ell^{k}\|_2\right).$$

① is due to triangle and Cauchy Schwarz inequalities, where we make a upper bound relaxation from $\infty$-norm to 2-norm. ② is due to 1-Lipschitz property of ReLU and $\max(\|W_L^{k+1}\|_2, \|W_L^k\|_2) \leq \bar{\lambda}_L$ in the definition. It is note-worthy that any activations with constant-Lipchitz properties can be applied. ③ is due to triangle and Cauchy Schwarz inequalities as well. We make a arrangement in ④ and eliminate inductions in $\cdots$. In ⑤. we make another upper bound relaxation from $\infty$-norm to 2-norm. ⑥ is due to triangle inequality, the definition of Frobenius norm, and $\|\mathbf{M}^\top\mathbf{M}\|_2 \leq L$ of objective's L-smooth property. $\square$

**Semi-Smoothness of Inner Output of NN, i.e., Bound** $\|G_{\ell,t}^a - G_{\ell,t}^b\|_2, \ell \in [L-1], \forall a, b, t$**.**

**Lemma A.4.** *Denote $\ell \in [L-1]$, for some $\bar{\lambda}_\ell \in \mathbb{R}$, we assume $\max(\|W_\ell^a\|_2, \|W_\ell^b\|_2) \leq \bar{\lambda}_\ell$. Using quantities from Equation (12), we have:*

$$\|G_{\ell,t}^a - G_{\ell,t}^b\|_2 \leq (1+\beta)\|X_{t-1}^a - X_{t-1}^b\|_2 \prod_{j=1}^{\ell} \bar{\lambda}_j$$
$$+ (\|X_{t-1}^b\|_2 + \|\mathbf{M}^\top(\mathbf{M}X_{t-1}^b - Y)\|_2) \prod_{j=1}^{\ell} \bar{\lambda}_j \sum_{s=1}^{\ell} \bar{\lambda}_s^{-1} \|W_s^a - W_s^b\|_2.$$

*Proof.* Since the bounding target in Lemma A.4 is a degenerated version of that in Lemma A.3. Similar to the proof of Lemma A.3, we calculate:

$$\|G_{\ell,t}^a - G_{\ell,t}^b\|_2$$
$$= \| \text{ReLU}(\text{ReLU}([X_{t-1}^a, \mathbf{M}^\top(\mathbf{M}X_{t-1}^a - Y)]W_1^{a\top}) \cdots W_\ell^{a\top})$$
$$- \text{ReLU}(\text{ReLU}([X_{t-1}^b, \mathbf{M}^\top(\mathbf{M}X_{t-1}^b - Y)]W_1^{b\top}) \cdots W_\ell^{b\top})\|_2,$$
$$\leq \|[X_{t-1}^a, \mathbf{M}^\top(\mathbf{M}X_{t-1}^a - Y)] - [X_{t-1}^b, \mathbf{M}^\top(\mathbf{M}X_{t-1}^b - Y)]\|_2 \prod_{j=1}^{\ell} \bar{\lambda}_j$$
$$+ \|[X_{t-1}^b, \mathbf{M}^\top(\mathbf{M}X_{t-1}^b - Y)]\|_2 \prod_{j=1}^{\ell} \bar{\lambda}_j \sum_{s=1}^{\ell} \bar{\lambda}_s^{-1} \|W_s^a - W_s^b\|_2,$$
$$\leq (1+\beta)\|X_{t-1}^a - X_{t-1}^b\|_2 \prod_{j=1}^{\ell} \bar{\lambda}_j$$
$$+ (\|X_{t-1}^b\|_2 + \|\mathbf{M}^\top(\mathbf{M}X_{t-1}^b - Y)\|_2) \prod_{j=1}^{\ell} \bar{\lambda}_j \sum_{s=1}^{\ell} \bar{\lambda}_s^{-1} \|W_s^a - W_s^b\|_2.$$

$\square$

**Bound NN's Inner Output $G_{l,t}^k$, $l = [L-1]$, $\forall k, t$.**

**Lemma A.5.** *Denote $\ell \in [L-1]$, for some $\bar{\lambda}_\ell \in \mathbb{R}$, we assume $\|W_\ell^k\|_2 \leq \bar{\lambda}_\ell$. Using quantities from Equation (12), we have:*

$$\|G_{\ell,t}^k\|_2 \leq \left((1+\beta)\|X_0\|_2 + \left(2t - 1 + \tfrac{2t-2}{\beta}\right)\|\mathbf{M}^\top Y\|_2\right) \prod_{s=1}^{\ell} \bar{\lambda}_s.$$

*Proof.*

$$\|G_{\ell,t}^k\|_2 = \| \text{ReLU}(\text{ReLU}([X_{t-1}^k, \mathbf{M}^\top(\mathbf{M}X_{t-1}^k - Y)]W_1^{k\top}) \cdots W_\ell^{k\top})\|_2,$$
$$\overset{①}{\leq} \|[X_{t-1}^k, \mathbf{M}^\top(\mathbf{M}X_{t-1}^k - Y)]\|_2 \prod_{s=1}^{\ell} \|W_s^k\|_2,$$
$$\overset{②}{\leq} (\|X_{t-1}^k\|_2 + \|\mathbf{M}^\top(\mathbf{M}X_{t-1}^k - Y)\|_2) \prod_{s=1}^{\ell} \|W_s^k\|_2,$$
$$\overset{③}{\leq} \left((1+\beta)\|X_0\|_2 + \left(\tfrac{(1+\beta)2(t-1)}{\beta} + 1\right)\|\mathbf{M}^\top Y\|_2\right) \prod_{s=1}^{\ell} \|W_s^k\|_2,$$
$$\leq \left((1+\beta)\|X_0\|_2 + \left(2t - 1 + \tfrac{2t-2}{\beta}\right)\|\mathbf{M}^\top Y\|_2\right) \prod_{s=1}^{\ell} \bar{\lambda}_s.$$

① is from equation (17), Lemma 4.2 of [30]. ② is from triangle inequality. ③ is due to definition of $\beta$-smoothness of objective and upper bound of $\|X_t\|_2$ in Lemma A.6. $\square$

### A.5.2 Outputs of L2O are Bounded

Next, we establish bounds for the Math-L2O's outputs. Leveraging the momentum-free setting, we formulate the dynamics from $X_0$ to $X_t$ as a *semi-linear* system, where parameters are non-linearly generated by the NN block (see Figure 1a). Application of the Cauchy-Schwarz and triangle inequalities to this system yields the following explicit bound.

**Lemma A.6** (Bound on Math-L2O Output). *For any training iteration $k$, the $t$-th output $X_t^k$ of Math-L2O (as per Equation (3)) is bounded by: $\|X_t^k\|_2 \leq \|X_0\|_2 + \tfrac{2t}{\beta}\|\mathbf{M}^\top Y\|_2$.*

*Proof.* We calculate the upper bound based on the one-line formulation from $X_0$ in Equation (22).

$$\|X_t^k\|_2$$
$$=\left\|\prod_{s=t}^{1}(\mathbf{I}-\tfrac{1}{\beta}\mathcal{D}(P_s^k)\mathbf{M}^\top\mathbf{M})X_0+\tfrac{1}{\beta}\sum_{s=1}^{t}\prod_{j=t}^{s+1}(\mathbf{I}-\tfrac{1}{\beta}\mathcal{D}(P_s^k)\mathbf{M}^\top\mathbf{M})\mathcal{D}(P_s^k)\mathbf{M}^\top Y\right\|_2$$
$$\overset{①}{\le}\left\|\prod_{s=1}^{t}(\mathbf{I}-\tfrac{1}{\beta}\mathcal{D}(P_s^k)\mathbf{M}^\top\mathbf{M})X_0\right\|_2+\left\|\tfrac{1}{\beta}\sum_{s=1}^{t}\prod_{j=t}^{s+1}(\mathbf{I}-\tfrac{1}{\beta}\mathcal{D}(P_s^k)\mathbf{M}^\top\mathbf{M})\mathcal{D}(P_s^k)\mathbf{M}^\top Y\right\|_2$$
$$\overset{②}{\le}\prod_{s=1}^{t}\left\|\mathbf{I}-\tfrac{1}{\beta}\mathcal{D}(P_s^k)\mathbf{M}^\top\mathbf{M}\right\|_2\|X_0\|_2$$
$$+\tfrac{1}{\beta}\sum_{s=1}^{t}\prod_{j=t}^{s+1}\left\|\mathbf{I}-\tfrac{1}{\beta}\mathcal{D}(P_s^k)\mathbf{M}^\top\mathbf{M}\right\|_2\|\mathcal{D}(P_s^k)\|_2\|\mathbf{M}^\top Y\|_2,$$
$$\overset{③}{\le}\|X_0\|_2+\tfrac{2}{\beta}\sum_{s=1}^{t}\|\mathbf{M}^\top Y\|_2=\|X_0\|_2+\tfrac{2t}{\beta}\|\mathbf{M}^\top Y\|_2,$$

where ① is from the triangle inequality, ② is due to Cauchy Schwarz inequalities, and ③ is due to Lemma A.1 and Lemma A.2. $\qquad\square$

This lemma demonstrates that Math-L2O outputs remain bounded independently of the training iteration $k$ and the specific learnable parameters.

### A.5.3  L2O is Semi-Smooth to Its Parameters

In this section, we treat the L2O model defined in Equation (21) and its corresponding neural network as functions of their learnable parameters. We then prove that these functions are semi-smooth with respect to these parameters. This property is foundational for establishing the convergence of the gradient descent algorithm, as its analysis inherently involves the relationship between parameters at adjacent iterations.

First, we give the following explicit formulation of $P$:

$$P_t^k = 2\sigma(W_L^k\,\text{ReLU}(W_{L-1}^k(\cdots\text{ReLU}(W_1^k[X_{t-1}^k,\mathbf{M}^\top(\mathbf{M}X_{t-1}^k-Y)]^\top)\cdots)))^\top,$$
$$= 2\sigma(\text{ReLU}(\cdots\text{ReLU}([X_{t-1}^k,\mathbf{M}^\top(\mathbf{M}X_{t-1}^k-Y)]W_1^\top)\cdots W_{L-1}^{k}{}^\top)W_L^k).$$

Moreover, we present ReLU activation function with signal matrices defined in Section 2. We denote $\cdot_K$ as the entry-wise product to the matrices, which is also equivalent to reshape a matrix to a vector then product a diagonal signal matrix and reshape back afterward.

$$P_t^k = 2\sigma(W_L^k\mathbf{D}_{L-1}\cdot_K W_{L-1}^k(\cdots\mathbf{D}_1\cdot_K(W_1^k[X_{t-1}^k,\mathbf{M}^\top(\mathbf{M}X_{t-1}^k-Y)]^\top)\cdots))^\top,$$
$$= 2\sigma((\cdots\cdots([X_{t-1}^k,\mathbf{M}^\top(\mathbf{M}X_{t-1}^k-Y)]W_1^\top)\cdot_K\mathbf{D}_1\cdots)W_{L-1}^{k}{}^\top\cdot_K\mathbf{D}_{L-1}W_L^k).$$

**Proof for Lemma 4.2.**  We demonstrate the semi-smoothness of Math-L2O's output, i.e., bound $\|X_t^{k+1}-X_t^k\|_2,\forall k,t$

*Proof.* Diverging from the approach in [30], $X_T^{k+1}$ and $X_T^k$ are the outputs of a non-linear neural network corresponding to different inputs. A direct subtraction between these terms, as would be feasible in a linear-like system, is therefore intractable. Consequently, we must construct an upper bound for this difference. By applying a norm-based relaxation and utilizing the quantities defined in

Equation (12), we proceed with the following calculation:

$$
\|X_t^{k+1} - X_t^k\|_2
$$
$$
= \| X_{t-1}^{k+1} - \tfrac{1}{\beta}\mathcal{D}(P_t^{k+1})\big(\mathbf{M}^\top(\mathbf{M}X_{t-1}^{k+1} - Y)\big) - \big(X_{t-1}^k - \tfrac{1}{\beta}\mathcal{D}(P_t^k)(\mathbf{M}^\top(\mathbf{M}X_{t-1}^k - Y))\big)\|_2,
$$
$$
= \Big\| \big(\mathbf{I} - \tfrac{1}{\beta}\mathcal{D}(P_t^{k+1})\mathbf{M}^\top\mathbf{M}\big)X_{t-1}^{k+1} - \big(\mathbf{I} - \tfrac{1}{\beta}\mathcal{D}(P_t^k)\mathbf{M}^\top\mathbf{M}\big)X_{t-1}^k
$$
$$
\qquad + \tfrac{1}{\beta}(\mathcal{D}(P_t^{k+1}) - \mathcal{D}(P_t^k))\mathbf{M}^\top Y \Big\|_2
$$
$$
\overset{①}{\leq} \Big\| \big(\mathbf{I} - \tfrac{1}{\beta}\mathcal{D}(P_t^{k+1})\mathbf{M}^\top\mathbf{M}\big) - \big(\mathbf{I} - \tfrac{1}{\beta}\mathcal{D}(P_t^k)\mathbf{M}^\top\mathbf{M}\big) \Big\|_2 \|X_{t-1}^{k+1}\|_2
$$
$$
\qquad + \Big\|\mathbf{I} - \tfrac{1}{\beta}\mathcal{D}(P_t^k)\mathbf{M}^\top\mathbf{M}\Big\|_2 \|X_{t-1}^{k+1} - X_{t-1}^k\|_2 + \tfrac{1}{\beta}\|\mathbf{M}^\top Y\|_2\|\mathcal{D}(P_t^{k+1}) - \mathcal{D}(P_t^k)\|_2,
$$
$$
\overset{②}{\leq} \|\mathcal{D}(P_t^{k+1}) - \mathcal{D}(P_t^k)\|_2 \|X_{t-1}^{k+1}\|_2 + \|X_{t-1}^{k+1} - X_{t-1}^k\|_2 + \tfrac{1}{\beta}\|\mathbf{M}^\top Y\|_2\|\mathcal{D}(P_t^{k+1}) - \mathcal{D}(P_t^k)\|_2,
$$
$$
\overset{③}{\leq} \|\mathcal{D}(P_t^{k+1}) - \mathcal{D}(P_t^k)\|_2(\|X_0\|_2 + \tfrac{2t-2}{\beta}\|\mathbf{M}^\top Y\|_2) + \|X_{t-1}^{k+1} - X_{t-1}^k\|_2
$$
$$
\qquad + \tfrac{1}{\beta}\|\mathbf{M}^\top Y\|_2\|\mathcal{D}(P_t^{k+1}) - \mathcal{D}(P_t^k)\|_2,
$$
$$
= (\|X_0\|_2 + \tfrac{2t-1}{\beta}\|\mathbf{M}^\top Y\|_2)\|\mathcal{D}(P_t^{k+1}) - \mathcal{D}(P_t^k)\|_2 + \|X_{t-1}^{k+1} - X_{t-1}^k\|_2,
$$
$$
\overset{④}{\leq} (\|X_0\|_2 + \tfrac{2t-1}{\beta}\|\mathbf{M}^\top Y\|_2)
$$
$$
\qquad \Big(\tfrac{1}{2}(1+\beta)\|X_{t-1}^{k+1} - X_{t-1}^k\|_2\Theta_L
$$
$$
\qquad\quad + \tfrac{1}{2}(\|X_{t-1}^k\|_2 + \|\mathbf{M}^\top(\mathbf{M}X_{t-1}^k - Y)\|_2)\Theta_L\sum_{\ell=1}^L \bar{\lambda}_\ell^{-1}\|W_\ell^{k+1} - W_\ell^k\|_2\Big)
$$
$$
\qquad + \|X_{t-1}^{k+1} - X_{t-1}^k\|_2,
$$
$$
= \Big(1 + (\|X_0\|_2 + \tfrac{2t-1}{\beta}\|\mathbf{M}^\top Y\|_2)\tfrac{1+\beta}{2}\Theta_L\Big)\|X_{t-1}^{k+1} - X_{t-1}^k\|_2,
$$
$$
\qquad + \tfrac{1}{2}(\|X_0\|_2 + \tfrac{2t-1}{\beta}\|\mathbf{M}^\top Y\|_2)
$$
$$
\qquad\quad (\|X_{t-1}^k\|_2 + \|\mathbf{M}^\top(\mathbf{M}X_{t-1}^k - Y)\|_2)\Theta_L\sum_{\ell=1}^L \bar{\lambda}_\ell^{-1}\|W_\ell^{k+1} - W_\ell^k\|_2,
$$
$$
\overset{⑤}{\leq} \tfrac{1}{2}\sum_{s=1}^t \Big(\prod_{j=s+1}^t \big(1 + (\|X_0\|_2 + \tfrac{2j-1}{\beta}\|\mathbf{M}^\top Y\|_2)\tfrac{1+\beta}{2}\Theta_L\big)\Big)
$$
$$
\qquad \underbrace{(\|X_0\|_2 + \tfrac{2s-1}{\beta}\|\mathbf{M}^\top Y\|_2)\big((1+\beta)\|X_0\|_2 + (2s-1+\tfrac{2s-2}{\beta})\|\mathbf{M}^\top Y\|_2\big)}_{\Lambda_s}
$$
$$
\qquad \Theta_L\sum_{\ell=1}^L \bar{\lambda}_\ell^{-1}\|W_\ell^{k+1} - W_\ell^k\|_2,
$$

where ① is from triangle inequality. ② is from Lemma A.6. ③ is due to inductive summation to $t = 1$. ④ is due to the semi-smoothness of NN's output in Lemma A.3. ⑤ is from induction.

*Remark* 6. We note that the above upper bound relaxation is non-loose. Current existing approaches derive semi-smoothness in terms of NN functions, where parameters matrices are linearly applied and activation functions are Lipschitz continuous. However, in our setting under [23], the sigmoid activation is not Lipschitz continuous. Moreover, the input that is utilized to generate $X_t^{k+1}$ is from $X_{t-1}^{k+1}$, which is not identical to the $X_{t-1}^k$ for generating $X_{t-1}^k$.

$\square$

### A.5.4 Gradients are Bounded

In this section, we derive bound for the gradient of each layer's parameter at the given iteration $k$.

**Proof for Lemma 4.1** We demonstrate that the gradients of Math-L2O's each layer are bounded.

*Proof.* For $\ell = L$, we calculate the gradient on $W_L^k$ (Equation (8)):

$$\left\| \frac{\partial F}{\partial W_L^k} \right\|_2$$

$$= \frac{1}{\beta} \left\| \sum_{t=1}^{T} \left( \mathbf{M}^\top (\mathbf{M} X_T^k - Y) \right)^\top \right.$$

$$\left. \left( \prod_{j=T}^{t+1} \mathbf{I} - \frac{1}{\beta} \mathcal{D}(P_j^k) \mathbf{M}^\top \mathbf{M} \right) \mathcal{D}\left( \mathbf{M}^\top (\mathbf{M} X_{t-1}^k - Y) \right) \mathcal{D}\left( P_t^k \odot (1 - P_t^k/2) \right) G_{L-1,t}^{k}{}^\top \right\|_2,$$

$$\overset{\text{①}}{\leq} \frac{1}{\beta} \sum_{t=1}^{T} \| \mathbf{M}^\top (\mathbf{M} X_T^k - Y) \|_2 \prod_{j=T}^{t+1} \left\| \left( \mathbf{I}_d - \frac{1}{\beta} \mathcal{D}(P_j^k) \mathbf{M}^\top \mathbf{M} \right) \right\|_2$$

$$\| \mathcal{D}\left( \mathbf{M}^\top (\mathbf{M} X_{t-1}^k - Y) \right) \|_2 \| \mathcal{D}\left( P_t^k \odot (1 - P_t^k/2) \right) \|_2 \| G_{L-1,t}^k \|_2,$$

$$\overset{\text{②}}{\leq} \frac{1}{2\sqrt{\beta}} \| \mathbf{M} X_T^k - Y \|_2 \sum_{t=1}^{T} (\| \mathbf{M}^\top \mathbf{M} X_{t-1}^k \|_2 + \| \mathbf{M}^\top Y \|_2) \| G_{L-1,t}^k \|_2,$$

$$\overset{\text{③}}{\leq} \frac{\sqrt{\beta}}{2} \| \mathbf{M} X_T^k - Y \|_2 \prod_{\ell=1}^{L-1} \bar{\lambda}_\ell \sum_{t=1}^{T} \left( (1+\beta) \| X_0 \|_2 + \left( 2t - 1 + \frac{2t-2}{\beta} \right) \| \mathbf{M}^\top Y \|_2 \right)$$

$$\left( \| X_0 \|_2 + \frac{2t-1}{\beta} \| \mathbf{M}^\top Y \|_2 \right),$$

$$= \frac{\sqrt{\beta}}{2} \| \mathbf{M} X_T^k - Y \|_2 \prod_{\ell=1}^{L-1} \bar{\lambda}_\ell \sum_{t=1}^{T}$$

$$\underbrace{(1+\beta)\| X_0 \|_2^2 + \left( (4t-3)(1 + \frac{1}{\beta}) + 1 \right) \| X_0 \|_2 \| \mathbf{M}^\top Y \|_2 + \frac{(2T-1)(\beta(2T-1)+(2T-2))}{\beta^2} \| \mathbf{M}^\top Y \|_2^2}_{\Lambda_t},$$

$$= \frac{\sqrt{\beta} \Theta_L S_{\Lambda,T}}{2 \bar{\lambda}_L} \| \mathbf{M} X_T^k - Y \|_2,$$

where ① is from triangle and Cauchy-Schwarz inequalities. ② is from the bound of "$p$" in Lemma A.1. ③ is from the bound of L2O model's output in Lemma A.6 and inner outputs in Lemma A.5.

For $\ell \in [L-1]$, we calculate gradient on $W_\ell^k$ (Equation (7)) at iteration $k$ by:

$$\left\| \frac{\partial F}{\partial W_\ell^k} \right\|_2$$

$$= \left\| -\frac{1}{\beta} \sum_{t=1}^{T} (\mathbf{M}^\top (\mathbf{M} X_T^k - Y))^\top \left( \prod_{j=T}^{t+1} \mathbf{I}_d - \frac{1}{\beta} \mathbf{M}^\top \mathbf{M} \mathcal{D}(P_j^k) \right) \right.$$

$$\mathcal{D}\left( \mathbf{M}^\top (\mathbf{M} X_{t-1}^k - Y) \right) \mathcal{D}\left( P_t^k \odot (1 - P_t^k/2) \right) (\mathbf{I}_d \otimes W_L^k)$$

$$\left. \prod_{j=L-1}^{\ell+1} \mathbf{D}_{j,t}^k \mathbf{I}_d \otimes W_j^k \mathbf{I}_{n_\ell} \otimes G_{\ell-1,t}^{k}{}^\top \right\|_2,$$

$$\overset{\text{①}}{\leq} \frac{1}{\beta} \sum_{t=1}^{T} \| \mathbf{M}^\top (\mathbf{M} X_T^k - Y) \|_2 \prod_{j=T}^{t+1} \| \mathbf{I}_d - \frac{1}{\beta} \mathbf{M}^\top \mathbf{M} \mathcal{D}(P_j^k) \|_2 \| \mathcal{D}\left( \mathbf{M}^\top (\mathbf{M} X_{t-1}^k - Y) \right) \|_2$$

$$\| \mathcal{D}\left( P_t^k \odot (1 - P_t^k/2) \right) (\mathbf{I}_d \otimes W_L^k) \|_2 \left\| \prod_{j=L-1}^{\ell+1} \mathbf{D}_{j,t}^k \mathbf{I}_d \otimes W_j^k \mathbf{I}_{n_\ell} \otimes G_{\ell-1,t}^{k}{}^\top \right\|_2,$$

$$\overset{\text{②}}{\leq} \frac{\sqrt{\beta}}{2} \| \mathbf{M} X_T^k - Y \|_2 \prod_{j=\ell+1}^{L} \| W_j^k \|_2 \sum_{t=1}^{T} (\| \mathbf{M}^\top \mathbf{M} X_{t-1}^k \|_2 + \| \mathbf{M}^\top Y \|_2) \| G_{\ell-1,t}^k \|_2,$$

$$\overset{\text{②̲}}{=} \frac{\sqrt{\beta}}{2} \| \mathbf{M} X_T^k - Y \|_2 \prod_{j=1, j \neq \ell}^{L} \bar{\lambda}_j \sum_{t=1}^{T}$$

$$\underbrace{(1+\beta)\| X_0 \|_2^2 + \left( (4t-3)(1 + \frac{1}{\beta}) + 1 \right) \| X_0 \|_2 \| \mathbf{M}^\top Y \|_2 + \frac{(2T-1)(\beta(2T-1)+(2T-2))}{\beta^2} \| \mathbf{M}^\top Y \|_2^2}_{\Lambda_t},$$

$$= \frac{\sqrt{\beta} \Theta_L}{2 \bar{\lambda}_\ell} S_{\Lambda,T} \| \mathbf{M} X_T^k - Y \|_2,$$

① is from triangle and Cauchy-Schwarz inequalities. Inequality ② is from bounds of "$p$" in Lemma A.1 and we make a rearrangement in it. In inequality ②, we use norm's triangle inequality of dot product and Kronecker product, bounds of NN's inner output in Lemma A.5, and we calculate $\prod_{j=1, j \neq \ell}^{L} \| W_j^k \|_2 = \prod_{j=\ell+1}^{L} \| W_j^k \|_2 * \prod_{s=1}^{\ell-1} \| W_j^k \|_2$. We reuse the result in the proof for the last layer's gradient upper bound for case $\ell = L$ in equality ③ to get the final result. $\qquad \square$

## A.6 Bound Linear Convergence Rate

Now we are able to substitute the above formulation into three bounding targets in Equation (44) and bound them one-by-one by the NTK theorem. We summarize the main idea of NTK theory before

the proof. The main technique of NTK theory is the establishment of non-singularity of the kernel matrix by a wide-NN layer, where kernel matrix is for the gradient of loss to learnable parameters. This invokes the Polyak-Lojasiewicz condition (a more relaxed condition than strongly convex) for linear convergence. Due to the page limit, we eliminate the explicit formulation of kernel matrix in main page. Following the methodology in [29], the non-singularity of kernel matrix is established by $\sigma_{\min}(G_{L-1,T}^0) > 0$. It is guaranteed by the conditions in Theorem 4.3 and implemented by the initialization strategy in Section 5.

*Proof.* We start to prove the Theorem 4.3 by proving the following lemma.

**Lemma A.7.**

$$
\begin{cases}
\|W_\ell^r\|_2 \leq \bar{\lambda}_\ell, & \ell \in [L], \quad r \in [0, k], \\
\sigma_{\min}(G_{L-1,T}^r) \geq \frac{1}{2}\alpha_0, & r \in [0, k], \\
F([W]^r) \leq (1 - \eta 4\eta \frac{\beta_0^2}{\beta^2}\delta_4)^r F([W]^0), & r \in [0, k].
\end{cases}
\tag{35}
$$

*Remark* 7. The first inequality means that there exists a scalar $\bar{\lambda}_\ell$ that bounds each layer's learnable parameter. The second inequality means that the last inner output is lower bounded. The last inequality is the linear rate of training.

### A.6.1 Induction Part 1: NN's Parameter and the Last Inner Output are Bounded

For $k = 0$, Equation (35) degenerates and holds trivially. Assume Equation (35) holds up to iteration $k$, we aim to prove it still holds for iteration $k + 1$. First, we calculate the following term:

$$
\begin{aligned}
\|W_\ell^{k+1} - W_\ell^0\|_2 &\overset{①}{\leq} \sum_{s=0}^k \|W_\ell^{s+1} - W_\ell^s\|_2 \\
&\overset{②}{=} \eta \sum_{s=0}^k \left\|\frac{\partial F}{W_\ell^s}\right\|_2 \\
&\overset{③}{\leq} \eta \sum_{s=0}^k \frac{\sqrt{\beta}\Theta_L}{2\bar{\lambda}_\ell} S_{\Lambda,T} \|\mathbf{M}X_T^s - Y\|_2, \\
&\overset{④}{\leq} \eta \frac{\sqrt{\beta}\Theta_L}{2\bar{\lambda}_\ell} S_{\Lambda,T} \sum_{s=0}^k (1 - \eta 4\eta \frac{\beta_0^2}{\beta^2}\delta_4)^{s/2} \|\mathbf{M}X_T^0 - Y\|_2,
\end{aligned}
$$

where ① is due to triangle inequality. ② is due to the definition of gradient descent. ③ is due the gradient is being upper-bounded in Lemma 4.1 and our assumption that $\|W_\ell^r\|_2 \leq \bar{\lambda}_\ell, \quad \ell \in [L], \forall r \in [0, k]$. ④ is due to the linear rate in our induction assumption.

Define $u := \sqrt{1 - \eta 4\eta \frac{\beta_0^2}{\beta^2}\delta_4}$, we calculate the summation of geometric sequence by:

$$
\begin{aligned}
\eta \frac{\sqrt{\beta}\Theta_L}{2\bar{\lambda}_\ell} S_{\Lambda,T} \sum_{s=0}^k u^s \|\mathbf{M}X_T^0 - Y\|_2 &= \eta \frac{\sqrt{\beta}\Theta_L}{2\bar{\lambda}_\ell} S_{\Lambda,T} \frac{1-u^{k+1}}{1-u} \|\mathbf{M}X_T^0 - Y\|_2, \\
&\overset{①}{=} \frac{1}{4\eta \frac{\beta_0^2}{\beta^2}\delta_4} \frac{\sqrt{\beta}\Theta_L}{2\bar{\lambda}_\ell} S_{\Lambda,T} (1 - u^2) \frac{1-u^{k+1}}{1-u} \|\mathbf{M}X_T^0 - Y\|_2, \\
&\overset{②}{\leq} \frac{1}{4\eta \frac{\beta_0^2}{\beta^2}\delta_4} \frac{\sqrt{\beta}\Theta_L}{2\bar{\lambda}_\ell} S_{\Lambda,T} \|\mathbf{M}X_T^0 - Y\|_2, \\
&\overset{③}{\leq} \frac{1}{4\eta \frac{\beta_0^2}{\beta^2}\delta_4} \frac{\sqrt{\beta}\Theta_L}{2\bar{\lambda}_\ell} S_{\Lambda,T} \left(\sqrt{\beta}\|X_0\|_2 + (2T+1)\|Y\|_2\right), \\
&\overset{④}{\leq} C_\ell,
\end{aligned}
$$

where ① is due to $1 - u^2 = \eta 4\eta \frac{\beta_0^2}{\beta^2}\delta_4$. ② is due to $0 \leq u \leq 1$. ③ is due to NN's output's bound in Lemma A.6. ④ is due to the lower bound on the singular value of last inner output layer in Equation (13c).

Thus, we have:

$$
\|W_\ell^{k+1} - W_\ell^0\|_2 \leq C_\ell.
\tag{36}
$$

Denote $\sigma_1(\cdot)$ as calculating the smallest singular value of any matrices, due to Weyl's inequality [28], we have:

$$\left|\|W_\ell^{k+1}\|_2 - \|W_\ell^0\|_2\right| \le \sigma_1(W_\ell^{k+1} - W_\ell^0),$$
$$\le \|W_\ell^{k+1} - W_\ell^0\|_2,$$
$$\le C_\ell.$$

where the first inequality is from Weyl's inequality and the last inequality is due to Equation (36). Then, we directly have $\|W_\ell^{k+1}\|_2 - \|W_\ell^0\|_2 \le C_\ell$ and $\|W_\ell^{k+1}\|_2 \le \|W_\ell^0\|_2 + C_\ell = \bar\lambda_\ell$.

Next, we bound $G_{L-1,T}^{k+1}$ by calculating:

$$\|G_{L-1,T}^{k+1} - G_{L-1,T}^0\|_2$$
$$\overset{①}{\le}(1+\beta)\|X_{T-1}^{k+1} - X_{T-1}^0\|_2\prod_{j=1}^{L-1}\bar\lambda_j$$
$$+ (\|X_{T-1}^0\|_2 + \|\mathbf{M}^\top(\mathbf{M}X_{T-1}^0 - Y)\|_2)\prod_{j=1}^{L-1}\bar\lambda_j\sum_{\ell=1}^{L-1}\bar\lambda_\ell^{-1}\|W_\ell^{k+1} - W_\ell^0\|_2,$$
$$\overset{②}{\le}(1+\beta)2(\|X_0\|_2 + \tfrac{2T-2}{\beta}\|\mathbf{M}^\top Y\|_2)\prod_{j=1}^{L-1}\bar\lambda_j$$
$$+ (\|X_{T-1}^0\|_2 + \|\mathbf{M}^\top(\mathbf{M}X_{T-1}^0 - Y)\|_2)\prod_{j=1}^{L-1}\bar\lambda_j\sum_{\ell=1}^{L-1}\bar\lambda_\ell^{-1}\|W_\ell^{k+1} - W_\ell^0\|_2,$$
$$\overset{③}{\le}(1+\beta)\sum_{i=0}^k\tfrac{1}{2}\Theta_L\underbrace{\sum_{s=1}^{T-1}\left(\prod_{j=s+1}^{T-1}\left(1+\tfrac{1+\beta}{2}\Theta_L\Phi_j\right)\right)\Lambda_s}_{\delta_1^{T-1}}\sum_{\ell=1}^L\bar\lambda_\ell^{-1}\|W_\ell^{i+1} - W_\ell^i\|_2\prod_{j=1}^{L-1}\bar\lambda_j$$
$$+ (\|X_{T-1}^0\|_2 + \|\mathbf{M}^\top(\mathbf{M}X_{T-1}^0 - Y)\|_2)\prod_{j=1}^{L-1}\bar\lambda_j\sum_{\ell=1}^L\bar\lambda_\ell^{-1}\|W_\ell^{k+1} - W_\ell^0\|_2,$$

(37)

where ① is due to the semi-smoothness of NN's inner output in Lemma A.4. ② is due to the triangle inequality. ③ is due to semi-smoothness of L2O in Lemma 4.2.

Further, based on the inner results in the former demonstration for $\|W_\ell^{k+1} - W_\ell^0\|_2$, we have:

$$\sum_{i=0}^k\|W_\ell^{i+1} - W_\ell^i\|_2 \le \frac{1}{4\eta\frac{\beta_0^2}{\beta^2}\delta_4}\frac{\sqrt{\beta}\Theta_L}{2\bar\lambda_\ell}S_{\Lambda,T}\|\mathbf{M}X_T^0 - Y\|_2.$$

Substituting above result back into Equation (37) yields:

$$\|G_{L-1,T}^{k+1} - G_{L-1,T}^0\|_2$$
$$\le(1+\beta)2(\|X_0\|_2 + \tfrac{2T-2}{\beta}\|\mathbf{M}^\top Y\|_2)\prod_{j=1}^{L-1}\bar\lambda_j$$
$$+ (\|X_{T-1}^0\|_2 + \|\mathbf{M}^\top(\mathbf{M}X_{T-1}^0 - Y)\|_2)\prod_{j=1}^{L-1}\bar\lambda_j\sum_{\ell=1}^{L-1}\bar\lambda_\ell^{-1}\frac{1}{4\eta\frac{\beta_0^2}{\beta^2}\delta_4}\frac{\sqrt{\beta}\Theta_L}{2\bar\lambda_\ell}S_{\Lambda,T}\|\mathbf{M}X_T^0 - Y\|_2,$$
$$\overset{①}{\le}\frac{1}{4\eta\frac{\beta_0^2}{\beta^2}\delta_4}(1+\beta)\zeta_2\left(\sqrt{\beta}\|X_0\|_2 + (2T+1)\|Y\|_2\right)S_{\Lambda,T}\prod_{j=1}^{L-1}\bar\lambda_j\sum_{\ell=1}^L\bar\lambda_\ell^{-1}\frac{\sqrt{\beta}\Theta_L}{2\bar\lambda_\ell}$$
$$+ 2(1+\beta)(\|X_0\|_2 + \tfrac{2T-2}{\beta}\|\mathbf{M}^\top Y\|_2)\prod_{j=1}^{L-1}\bar\lambda_j,$$
$$\overset{②}{\le}\frac{1}{4\eta\frac{\beta_0^2}{\beta^2}\delta_4}(1+\beta)\zeta_2\left(\sqrt{\beta}\|X_0\|_2 + (2T+1)\|Y\|_2\right)S_{\Lambda,T}\prod_{j=1}^{L-1}\bar\lambda_j\sum_{\ell=1}^L\bar\lambda_\ell^{-1}\frac{\sqrt{\beta}\Theta_L}{2\bar\lambda_\ell}$$
$$+ \tfrac{1}{4}\alpha_0,$$
$$\overset{③}{\le}\tfrac{1}{2}\alpha_0,$$

(38)

where ① is due to NN's output's bound in Lemma A.6 and ② and ③ are due to the other lower bound for minimal singular value of NN's inner output in Equation (13a) and Equation (13d). The inequality in Equation (38) implies $\sigma_{\min}(G_{L-1}^{k+1}) \ge \tfrac{1}{2}\alpha_0$ since $\sigma_{\min}(G_{L-1}^0) = \alpha_0$.

Based on the above two inequalities, we prove the linear rate in Theorem 4.3 step-by-step in the following sub-section.

### A.6.2 Induction Part 2: Linear Convergence

In this section, we aim to prove that $F([W]^{k+1}) \leq (1 - \eta 4\eta \frac{\beta_0^2}{\beta^2} \delta_4)^{k+1} F([W]^0)$.

**Step 1: Split Perfect Square**  By leveraging term $\mathbf{M} X_T^k$, we can split the perfect square in objective $F([W]^{k+1})$ as:

$$F([W]^{k+1}) = F([W]^k) + \tfrac{1}{2}\|\mathbf{M} X_T^{k+1} - \mathbf{M} X_T^k\|_2^2 + (\mathbf{M} X_T^{k+1} - \mathbf{M} X_T^k)^\top (\mathbf{M} X_T^k - Y). \quad (39)$$

Based on [29], we aim to demonstrate that $F([W]^{k+1})$ can be upper-bounded by $c_k F([W]^k)$, where $c_k < 1$ is a coefficient related to training iteration $k$.

**Step 2: Bound Term-by-Term**  We aim to upperly bound all terms in Equation (39) by $F([W]^k)$.

**Bound the first term** $\tfrac{1}{2}\|\mathbf{M} X_T^{k+1} - \mathbf{M} X_T^k\|_2^2$. First, based on the $\beta$-smoothness of objective $F$, we calculate

$$\begin{aligned}
\tfrac{1}{2}\|\mathbf{M} X_T^{k+1} - \mathbf{M} X_T^k\|_2^2 &= \tfrac{1}{2}(X_T^{k+1} - X_T^k)^\top \mathbf{M}^\top \mathbf{M}(X_T^{k+1} - X_T^k), \\
&\leq \tfrac{1}{2}\|X_T^{k+1} - X_T^k\|_2^2 \|\mathbf{M}^\top \mathbf{M}\|_2, \\
&\leq \tfrac{\beta}{2}\|X_T^{k+1} - X_T^k\|_2^2.
\end{aligned}$$

The above inequality shows that we need to bound the distance between outputs of two iterations. Moreover, since our target is to construct linear convergence rate, we need to find the upper bound of above inequality w.r.t. the objective $F([W]^k)$, i.e., $\tfrac{1}{2}\|\mathbf{M} X_T^k - Y\|_2^2$. We apply Lemma 4.2 to derive the following lemma.

**Lemma A.8.** *Denote $\ell \in [L]$, for some $\bar{\lambda}_\ell \in \mathbb{R}$, we assume $\max(\|W_\ell^{k+1}\|_2, \|W_\ell^k\|_2) \leq \bar{\lambda}_\ell, \forall k$. Using quantities from Equation (12), we further define the following quantities with $i, j \in [T]$:*

$$\begin{aligned}
\Lambda_i &= (1+\beta)\|X_0\|_2^2 + \big((4i-3)(1+\tfrac{1}{\beta})+1\big)\|X_0\|_2\|\mathbf{M}^\top Y\|_2 \\
&\quad + \tfrac{(2i-1)(\beta(2i-1)+(2i-2))}{\beta^2}\|\mathbf{M}^\top Y\|_2^2, \\
\Phi_j &= \|X_0\|_2 + \tfrac{2j-1}{\beta}\|\mathbf{M}^\top Y\|_2, \\
\delta_1{}^T &= \Big(\sum_{s=1}^T (\prod_{j=s+1}^T (1 + \tfrac{1+\beta}{2}\Theta_L \Phi_j))\Big(\sum_{j=1}^s \Lambda_j\Big)\Big).
\end{aligned}$$

*We have the following upperly bounding property:*

$$\tfrac{1}{2}\|\mathbf{M} X_T^{k+1} - \mathbf{M} X_T^k\|_2^2 \leq \tfrac{\beta^2 \eta^2}{16}(\delta_1{}^T)^2 \Big(S_{\Lambda,T}\Big)^2 \Big(\Theta_L^2 \sum_{\ell=1}^L \bar{\lambda}_\ell^{-2}\Big)^2 \tfrac{1}{2}\|\mathbf{M} X_T^k - Y\|_2. \quad (40)$$

*Proof.* We calculate:

$$\begin{aligned}
&\tfrac{1}{2}\|\mathbf{M} X_T^{k+1} - \mathbf{M} X_T^k\|_2^2 \leq \tfrac{\beta}{2}\|X_T^{k+1} - X_T^k\|_2^2, \\
&\overset{①}{\leq} \tfrac{\beta}{2}\Big(\sum_{s=1}^T \big(\prod_{j=s+1}^T (1 + \tfrac{1+\beta}{2}\Theta_L \Phi_j)\big)\tfrac{1}{2}\Lambda_s \Theta_L \sum_{\ell=1}^L \bar{\lambda}_\ell^{-1}\|W_\ell^{k+1} - W_\ell^k\|_2\Big)^2, \\
&\overset{②}{=} \tfrac{\beta\eta^2}{2}\Big(\sum_{s=1}^T \big(\prod_{j=s+1}^T (1 + \tfrac{1+\beta}{2}\Theta_L \Phi_j)\big)\tfrac{1}{2}\Lambda_s \Theta_L \sum_{\ell=1}^L \bar{\lambda}_\ell^{-1}\big\|\tfrac{\partial F}{\partial W_\ell^k}\big\|_2\Big)^2, \\
&\overset{③}{\leq} \tfrac{\beta\eta^2}{2}\Big(\sum_{s=1}^T \big(\prod_{j=s+1}^T (1 + \tfrac{1+\beta}{2}\Theta_L \Phi_j)\big)\tfrac{1}{2}\Lambda_s \Theta_L \sum_{\ell=1}^L \bar{\lambda}_\ell^{-1}\tfrac{\sqrt{\beta}\Theta_L}{2\bar{\lambda}_\ell}\big(S_{\Lambda,T}\big)\|\mathbf{M} X_T^k - Y\|_2\Big)^2, \\
&= \tfrac{\beta^2 \eta^2}{32}\Big(\underbrace{\big(\sum_{s=1}^T (\prod_{j=s+1}^T (1 + \tfrac{1+\beta}{2}\Theta_L \Phi_j))\Lambda_s\big)}_{\delta_1{}^T}\big(S_{\Lambda,T}\big)\Theta_L^2 \sum_{\ell=1}^L \bar{\lambda}_\ell^{-2}\|\mathbf{M} X_T^k - Y\|_2\Big)^2, \\
&= \tfrac{\beta^2 \eta^2}{16}(\delta_1{}^T)^2 \Big(S_{\Lambda,T}\Big)^2 \Big(\Theta_L^2 \sum_{\ell=1}^L \bar{\lambda}_\ell^{-2}\Big)^2 \tfrac{1}{2}\|\mathbf{M} X_T^k - Y\|_2,
\end{aligned}$$

$$(41)$$

① is from semi-smoothness of L2O's output in Lemma 4.2, Appendix A.5.3. ② is due to gradient descent with learning rate $\eta$. ③ is from gradient bounds in Lemma 4.1. $\qquad \square$

**Bound the second term** $(\mathbf{M}X_T^{k+1} - \mathbf{M}X_T^k)^\top(\mathbf{M}X_T^k - Y)$. We calculate:

$$
\begin{aligned}
&(\mathbf{M}X_T^{k+1} - \mathbf{M}X_T^k)^\top(\mathbf{M}X_T^k - Y)\\
=&(X_T^{k+1} - X_T^k)^\top\mathbf{M}^\top(\mathbf{M}X_T^k - Y),\\
=&(X_T^{k+1} - X_T^k)^\top\mathbf{M}^\top(\mathbf{M}X_T^k - Y).
\end{aligned}
\tag{42}
$$

Following the methodology in [29], we hold all other learnable parameters fixed and focus the analysis on the gradient with respect to the last layer, $W_L$. This approach facilitates the construction of a non-singular NTK, which in turn establishes the PL condition, thereby guaranteeing a linear convergence rate.

Given last NN layer's learnable parameter $W_L^{k+1}$ at iteration $k + 1$, due to the GD formulation in Equation (21), we define the following quantity:

$$
Z = X_{T-1}^k - \tfrac{1}{\beta}\mathcal{D}(2\sigma(W_L^{k+1}G_{L-1,T}^k)^\top)\mathbf{M}^\top(\mathbf{M}X_{T-1}^k - Y),
\tag{43}
$$

where $G_{L-1,T}^k$ represents inner output of layer $L - 1$ at training iteration $k$.

With $Z$, we reformulate Equation (42) as:

$$
\begin{aligned}
&(X_T^{k+1} - X_T^k)^\top\mathbf{M}^\top(\mathbf{M}X_T^k - Y),\\
=&(X_T^{k+1} - Z + Z - X_T^k)^\top\mathbf{M}^\top(\mathbf{M}X_T^k - Y),\\
=&(X_T^{k+1} - Z)^\top\mathbf{M}^\top(\mathbf{M}X_T^k - Y) + (Z - X_T^k)^\top\mathbf{M}^\top(\mathbf{M}X_T^k - Y),
\end{aligned}
\tag{44}
$$

where $X_T^{k+1}$ at training iteration $k + 1$ with $W_L^{k+1}$ and solution $X_T^k$ at training iteration $k$ with $W_L^k$ are defined as:

$$
X_T^{k+1} = X_{T-1}^{k+1} - \tfrac{1}{\beta}\mathcal{D}(2\sigma(W_L^{k+1}G_{L-1,T}^{k+1})^\top)\mathbf{M}^\top(\mathbf{M}X_{T-1}^{k+1} - Y).
$$

$$
X_T^k = X_{T-1}^k - \tfrac{1}{\beta}\mathcal{D}(2\sigma(W_L^kG_{L-1,T}^k)^\top)\mathbf{M}^\top(\mathbf{M}X_{T-1}^k - Y).
$$

Then, we have the following lemmas to bound the two terms, respectively:

**Lemma A.9.** *Denote $\ell \in [L]$, for some $\bar{\lambda}_\ell, \in \mathbb{R}$ with $j \in [T]$, we assume $\max(\|W_\ell^{k+1}\|_2, \|W_\ell^k\|_2) \le \bar{\lambda}_\ell$. Define the following quantities with $t \in [T]$:*

$$
\begin{aligned}
\Lambda_t =&(1 + \beta)\|X_0\|_2^2 + \big((4t - 3)(1 + \tfrac{1}{\beta}) + 1\big)\|X_0\|_2\|\mathbf{M}^\top Y\|_2\\
&+ \tfrac{(2T-1)(\beta(2T-1)+(2T-2))}{\beta^2}\|\mathbf{M}^\top Y\|_2^2,\\
\Phi_j =&\|X_0\|_2 + \tfrac{2j-1}{\beta}\mathbf{M}^\top Y\|_2,\\
\Theta_L =&\Theta_L,\\
\delta_2 =&\textstyle\sum_{s=1}^{T-1}\Big(\prod_{j=s+1}^T\big(1 + \tfrac{1+\beta}{2}\Theta_L\Phi_j\big)\Big)\Lambda_s.
\end{aligned}
$$

*We have the following upperly bounding property:*

$$
(X_T^{k+1} - Z)^\top\mathbf{M}^\top(\mathbf{M}X_T^k - Y) \le \tfrac{\beta\eta}{2}(\Lambda_T + \delta_2)\Theta_L^2 S_{\bar{\lambda},L}S_{\Lambda,T}\tfrac{1}{2}\|\mathbf{M}X_T^k - Y\|_2^2.
$$

*Proof.* We straightforwardly apply upper bound relaxation in this part, where we reuse the results of the first term $\tfrac{1}{2}\|\mathbf{M}X_T^{k+1} - \mathbf{M}X_T^k\|_2^2$'s upper bound in Lemma A.8.

To reuse the results, we would like to construct the $X_{T-1}^{k+1} - X_{T-1}^k$ term. We substitute Equation (46) into above equation and use the Cauchy-Schwarz inequality for vectors to split our bounding targets

into two parts and relax the $L_2$-norm of vector summations into each element by triangle inequalities:

$$(X_T^{k+1} - Z)^\top \mathbf{M}^\top (\mathbf{M}X_T^k - Y)$$

$$= \Big( X_{T-1}^{k+1} - \tfrac{1}{\beta}\mathcal{D}(2\sigma(W_L^{k+1}G_{L-1,T}^{k+1})^\top)\mathbf{M}^\top (\mathbf{M}X_{T-1}^{k+1} - Y)$$

$$- \big( X_{T-1}^k - \tfrac{1}{\beta}\mathcal{D}(2\sigma(W_L^{k+1}G_{L-1,T}^k)^\top)\mathbf{M}^\top (\mathbf{M}X_{T-1}^k - Y)\big)\Big)^\top \mathbf{M}^\top (\mathbf{M}X_T^k - Y),$$

$$\overset{①}{\leq} \Big( \Big\| \big(\mathbf{I}_d - \tfrac{1}{\beta}\mathcal{D}(2\sigma(W_L^{k+1}G_{L-1,T}^{k+1})^\top)\mathbf{M}^\top\mathbf{M}\big)X_{T-1}^{k+1}$$

$$- \big(\mathbf{I}_d - \tfrac{1}{\beta}\mathcal{D}(2\sigma(W_L^{k+1}G_{L-1,T}^k)^\top)\mathbf{M}^\top\mathbf{M}\big)X_{T-1}^k \Big\|_2$$

$$+ \tfrac{1}{\beta}\Big\| \Big(\underbrace{\mathcal{D}(2\sigma(W_L^{k+1}G_{L-1,T}^{k+1})^\top) - \mathcal{D}(2\sigma(W_L^{k+1}G_{L-1,T}^k)^\top)}_{C_{k+1}}\Big)\mathbf{M}^\top Y \Big\|_2\Big)$$

$$\|\mathbf{M}^\top(\mathbf{M}X_T^k - Y)\|_2,$$

$$\overset{②}{\leq} \Big( \Big\| \big(\mathbf{I}_d - \tfrac{1}{\beta}\mathcal{D}(2\sigma(W_L^{k+1}G_{L-1,T}^{k+1})^\top)\mathbf{M}^\top\mathbf{M}\big)(X_{T-1}^{k+1} - X_{T-1}^k) \Big\|_2 \qquad (45)$$

$$+ \Big\| \Big( \big(\mathbf{I}_d - \tfrac{1}{\beta}\mathcal{D}(2\sigma(W_L^{k+1}G_{L-1,T}^{k+1})^\top)\mathbf{M}^\top\mathbf{M}\big)$$

$$- \big(\mathbf{I}_d - \tfrac{1}{\beta}\mathcal{D}(2\sigma(W_L^{k+1}G_{L-1,T}^k)^\top)\mathbf{M}^\top\mathbf{M}\big)\Big)X_{T-1}^k \Big\|_2$$

$$+ \tfrac{1}{\beta}\|C_{k+1}\mathbf{M}^\top Y\|_2\Big)\|\mathbf{M}^\top(\mathbf{M}X_T^k - Y)\|_2,$$

$$\overset{③}{\leq} \Big( \Big\| \big(\mathbf{I}_d - \tfrac{1}{\beta}\mathcal{D}(2\sigma(W_L^{k+1}G_{L-1,T}^{k+1})^\top)\mathbf{M}^\top\mathbf{M}\big)\Big\|_2 \|X_{T-1}^{k+1} - X_{T-1}^k\|_2$$

$$+ \|\tfrac{1}{\beta}C_{k+1}\mathbf{M}^\top\mathbf{M}\|_2\|X_{T-1}^k\|_2 + \tfrac{1}{\beta}\|C_{k+1}\|_2\|\mathbf{M}^\top Y\|_2\Big)\|\mathbf{M}^\top(\mathbf{M}X_T^k - Y)\|_2,$$

$$\overset{④}{\leq} \Big( \|X_{T-1}^{k+1} - X_{T-1}^k\|_2 + \|X_{T-1}^k\|_2\|C_{k+1}\|_2 + \tfrac{1}{\beta}\|\mathbf{M}^\top Y\|_2\|C_{k+1}\|_2\Big)\|\mathbf{M}^\top(\mathbf{M}X_T^k - Y)\|_2,$$

$$\overset{⑤}{\leq} \Big( \|X_{T-1}^{k+1} - X_{T-1}^k\|_2 + \big(\|X_0\|_2 + \tfrac{2T-1}{\beta}\|\mathbf{M}^\top Y\|_2\big)\|C_{k+1}\|_2\Big)\|\mathbf{M}^\top(\mathbf{M}X_T^k - Y)\|_2,$$

where ① is due to triangle and Cauchy-Schwarz inequalities. ② is due to triangle inequality. ③ is due to Cauchy-Schwarz inequality. ④ is due to $\beta$-smooth definition that $\mathbf{M}^\top\mathbf{M} \leq \beta$ and $\|\mathbf{I}_d - \tfrac{1}{\beta}\mathcal{D}(2\sigma(W_L^{k+1}G_{L-1,T}^{k+1})^\top)\mathbf{M}^\top\mathbf{M}\|_2 \leq 1$ in Lemma A.1. ④ is due to the upper bound of $X_{T-1}$ in Lemma A.6.

Further, we bound $C_{k+1} := \mathcal{D}(2\sigma(W_L^{k+1}G_{L-1,T}^{k+1})^\top) - \mathcal{D}(2\sigma(W_L^{k+1}G_{L-1,T}^k)^\top)$. We apply the Mean Value Theorem and assume a point $v_1^k$. For $v_1^k$'s each entry $(v_1^k)_i$, for some $\alpha_{1i}^k \in [0, 1]$, we calculate $(v_1^k)_i$ as:

$$(v_1^k)_i = \alpha_{1i}^k((W_L^{k+1}G_{L-1,T}^{k+1})^\top)_i + (1 - \alpha_{1i}^k)((W_L^{k+1}G_{L-1,T}^k)^\top)_i.$$

Then, we can represent quantity $\|C_{k+1}\|_2$ by:

$$\|\mathcal{D}(2\sigma(W_L^{k+1}G_{L-1,T}^{k+1})^\top) - \mathcal{D}(2\sigma(W_L^{k+1}G_{L-1,T}^k)^\top)\|_2$$

$$\overset{①}{\leq} \Big\| \tfrac{\partial 2\sigma}{\partial v_1^k} \odot (W_L^{k+1}G_{L-1,T}^{k+1} - W_L^{k+1}G_{L-1,T}^k)^\top \Big\|_\infty,$$

$$\overset{②}{\leq} \tfrac{1}{2}\Big\| (W_L^{k+1}G_{L-1,T}^{k+1} - W_L^{k+1}G_{L-1,T}^k)^\top \Big\|_\infty,$$

$$\overset{③}{\leq} \tfrac{1}{2}\|W_L^{k+1}\|_2\|G_{L-1,T}^{k+1} - G_{L-1,T}^k\|_2 \leq \tfrac{1}{2}\bar{\lambda}_L\|G_{L-1,T}^{k+1} - G_{L-1,T}^k\|_2,$$

where ① is from the Mean Value Theorem. ② is from the gradient upper bound of Sigmoid function. ③ is from triangle inequality and definition of learnable parameter $W_L$.

We further substitute the upper bound of $\|G_{L-1,T}^{k+1} - G_{L-1,T}^k\|_2$ in Lemma A.4 and calculate:

$$\frac{1}{2}\bar{\lambda}_L\|G_{L-1,T}^{k+1} - G_{L-1,T}^k\|_2$$
$$\leq \frac{1}{2}\bar{\lambda}_L\Big((1+\beta)\|X_{T-1}^{k+1} - X_{T-1}^k\|_2\prod_{j=1}^{L-1}\bar{\lambda}_j$$
$$+ (\|X_{T-1}^k\|_2 + \|\mathbf{M}^\top(\mathbf{M}X_{T-1}^k - Y)\|_2)\prod_{j=1}^{L-1}\bar{\lambda}_j\sum_{\ell=1}^{L-1}\bar{\lambda}_\ell^{-1}\|W_\ell^{k+1} - W_\ell^k\|_2\Big)$$
$$\overset{①}{\leq}\frac{1}{2}(1+\beta)\Theta_L\|X_{T-1}^{k+1} - X_{T-1}^k\|_2$$
$$+ \frac{1}{2}\Big((1+\beta)\|X_0\|_2 + (2T-1+\tfrac{2T-2}{\beta})\|\mathbf{M}^\top Y\|_2\Big)\Theta_L\sum_{\ell=1}^{L-1}\bar{\lambda}_\ell^{-1}\|W_\ell^{k+1} - W_\ell^k\|_2.$$

where ① is due to upper bound of $X_{T-1}$ in Lemma A.6.

Substituting the above inequality back into Equation (45) yields:

$$(X_T^{k+1} - Z)^\top\mathbf{M}^\top(\mathbf{M}X_T^k - Y)$$
$$\leq\Big(\|X_{T-1}^{k+1} - X_{T-1}^k\|_2 + (\|X_0\|_2 + \tfrac{2T-1}{\beta}\|\mathbf{M}^\top Y\|_2)\|C_{k+1}\|_2\Big)\|\mathbf{M}^\top(\mathbf{M}X_T^k - Y)\|_2,$$
$$\leq\Big(\|X_{T-1}^{k+1} - X_{T-1}^k\|_2$$
$$+ (\|X_0\|_2 + \tfrac{2T-1}{\beta}\|\mathbf{M}^\top Y\|_2)$$
$$\Big(\tfrac{1}{2}(1+\beta)\Theta_L\|X_{T-1}^{k+1} - X_{T-1}^k\|_2$$
$$+ \tfrac{1}{2}\big((1+\beta)\|X_0\|_2 + (2T-1+\tfrac{2T-2}{\beta})\|\mathbf{M}^\top Y\|_2\big)\Theta_L\sum_{\ell=1}^{L-1}\bar{\lambda}_\ell^{-1}\|W_\ell^{k+1} - W_\ell^k\|_2\Big)\Big)$$
$$\|\mathbf{M}^\top(\mathbf{M}X_T^k - Y)\|_2,$$
$$=\Big(\big(1 + \tfrac{1+\beta}{2}\Theta_L(\|X_0\|_2 + \tfrac{2T-1}{\beta}\|\mathbf{M}^\top Y\|_2)\big)\|X_{T-1}^{k+1} - X_{T-1}^k\|_2$$
$$+ \Big(\tfrac{1}{2}\big((1+\beta)\|X_0\|_2 + (2T-1+\tfrac{2T-2}{\beta})\|\mathbf{M}^\top Y\|_2\big)$$
$$(\|X_0\|_2 + \tfrac{2T-1}{\beta}\|\mathbf{M}^\top Y\|_2)\Theta_L\sum_{\ell=1}^{L-1}\bar{\lambda}_\ell^{-1}\|W_\ell^{k+1} - W_\ell^k\|_2\Big)\Big)$$
$$\|\mathbf{M}^\top(\mathbf{M}X_T^k - Y)\|_2,$$
$$=\Big(\big(1 + \tfrac{1+\beta}{2}\Theta_L\underbrace{(\|X_0\|_2 + \tfrac{2T-1}{\beta}\|\mathbf{M}^\top Y\|_2)}_{\Phi_T}\big)\|X_{T-1}^{k+1} - X_{T-1}^k\|_2 +$$
$$\Big(\tfrac{1}{2}\underbrace{(1+\beta)\|X_0\|_2^2 + ((4T-3)(1+\tfrac{1}{\beta})+1)\|X_0\|_2\|\mathbf{M}^\top Y\|_2 + \tfrac{(2T-1)(\beta(2T-1)+(2T-2))}{\beta^2}\|\mathbf{M}^\top Y\|_2^2}_{\Lambda_T}\Big)$$
$$\Theta_L\sum_{\ell=1}^{L-1}\bar{\lambda}_\ell^{-1}\|W_\ell^{k+1} - W_\ell^k\|_2\Big)\|\mathbf{M}^\top(\mathbf{M}X_T^k - Y)\|_2,$$
$$=\Big(\big(1 + \tfrac{1+\beta}{2}\Theta_L\Phi_T\big)\|X_{T-1}^{k+1} - X_{T-1}^k\|_2 + \tfrac{1}{2}\Lambda_T\Theta_L\sum_{\ell=1}^{L-1}\bar{\lambda}_\ell^{-1}\|W_\ell^{k+1} - W_\ell^k\|_2\Big)$$
$$\|\mathbf{M}^\top(\mathbf{M}X_T^k - Y)\|_2,$$

Further, we apply semi-smoothness of L2O model in Lemma 4.2 and upper bound of gradient in Lemma 4.1 to derive the upper bound. We calculate:

$$(X_T^{k+1} - Z)^\top \mathbf{M}^\top (\mathbf{M}X_T^k - Y)$$

$$\leq \Big( \big(1 + \tfrac{1+\beta}{2}\Theta_L\Phi_T\big)\|X_{T-1}^{k+1} - X_{T-1}^k\|_2 + \tfrac{1}{2}\Lambda_T\Theta_L\sum_{\ell=1}^{L-1}\bar{\lambda}_\ell^{-1}\|W_\ell^{k+1} - W_\ell^k\|_2 \Big)$$

$$\quad \|\mathbf{M}^\top (\mathbf{M}X_T^k - Y)\|_2,$$

$$\overset{①}{\leq} \Big( \big(1 + \tfrac{1+\beta}{2}\Theta_L\Phi_T\big)\tfrac{1}{2}\Theta_L\sum_{s=1}^{T-1}\Big(\prod_{j=s+1}^{T-1}\big(1 + \tfrac{1+\beta}{2}\Theta_L\Phi_j\big)\Big)\Lambda_s\sum_{\ell=1}^{L}\bar{\lambda}_\ell^{-1}\|W_\ell^{k+1} - W_\ell^k\|_2$$

$$\quad + \tfrac{1}{2}\Lambda_T\Theta_L\sum_{\ell=1}^{L-1}\bar{\lambda}_\ell^{-1}\|W_\ell^{k+1} - W_\ell^k\|_2\Big)\|\mathbf{M}^\top (\mathbf{M}X_T^k - Y)\|_2,$$

$$\leq \Big( \tfrac{1}{2}\Theta_L \underbrace{\sum_{s=1}^{T-1}\Big(\prod_{j=s+1}^{T}\big(1 + \tfrac{1+\beta}{2}\Theta_L\Phi_j\big)\Big)\Lambda_s}_{\delta_2} \sum_{\ell=1}^{L}\bar{\lambda}_\ell^{-1}\|W_\ell^{k+1} - W_\ell^k\|_2$$

$$\quad + \tfrac{1}{2}\Lambda_T\Theta_L\sum_{\ell=1}^{L-1}\bar{\lambda}_\ell^{-1}\|W_\ell^{k+1} - W_\ell^k\|_2\Big)\|\mathbf{M}^\top (\mathbf{M}X_T^k - Y)\|_2,$$

$$= \tfrac{1}{2}\Theta_L\Big(\delta_2\bar{\lambda}_L^{-1}\|W_L^{k+1} - W_L^k\|_2 + (\Lambda_T + \delta_2)\sum_{\ell=1}^{L-1}\bar{\lambda}_\ell^{-1}\|W_\ell^{k+1} - W_\ell^k\|_2\Big)\|\mathbf{M}^\top (\mathbf{M}X_T^k - Y)\|_2,$$

$$\overset{②}{\leq} \tfrac{1}{2}\Theta_L(\Lambda_T + \delta_2)\sum_{\ell=1}^{L}\bar{\lambda}_\ell^{-1}\|W_\ell^{k+1} - W_\ell^k\|_2\|\mathbf{M}^\top (\mathbf{M}X_T^k - Y)\|_2,$$

where ① is due to Lemma 4.2. ② is due to $\Lambda_T \geq 0$.

Further, based on the gradient descent, i.e., $W_\ell^{k+1} = W_\ell^k - \eta\frac{\partial F}{\partial W_\ell^k}$, we substitute the bound of gradient in Lemma 4.1 and calculate:

$$(X_T^{k+1} - Z)^\top \mathbf{M}^\top (\mathbf{M}X_T^k - Y)$$

$$\leq \tfrac{1}{2}\Theta_L(\Lambda_T + \delta_2)\sum_{\ell=1}^{L}\bar{\lambda}_\ell^{-1}\|W_\ell^{k+1} - W_\ell^k\|_2\|\mathbf{M}^\top (\mathbf{M}X_T^k - Y)\|_2,$$

$$\leq \tfrac{\eta}{2}\Theta_L(\Lambda_T + \delta_2)\sum_{\ell=1}^{L}\bar{\lambda}_\ell^{-1}\Big\|\tfrac{\partial F}{\partial W_\ell^k}\Big\|_2\|\mathbf{M}^\top (\mathbf{M}X_T^k - Y)\|_2,$$

$$\overset{①}{\leq} \tfrac{\eta}{2}\Theta_L(\Lambda_T + \delta_2)\sum_{\ell=1}^{L}\bar{\lambda}_\ell^{-1}\tfrac{\sqrt{\beta}\Theta_L}{2\bar{\lambda}_\ell}S_{\Lambda,T}\|\mathbf{M}X_T^k - Y\|_2\|\mathbf{M}^\top (\mathbf{M}X_T^k - Y)\|_2,$$

$$\overset{②}{\leq} \tfrac{\beta\eta}{2}(\Lambda_T + \delta_2)\Theta_L^2 S_{\bar{\lambda},L}S_{\Lambda,T}\tfrac{1}{2}\|\mathbf{M}X_T^k - Y\|_2^2,$$

where ① is due to Lemma 4.1 and ② is due to $\|M\|_2 \leq \sqrt{\beta}$. $\qquad\square$

**Lemma A.10.** *Define the following quantities with $t \in [T]$:*

$$\Lambda_t = (1 + \beta)\|X_0\|_2^2 + \big((4t - 3)(1 + \tfrac{1}{\beta}) + 1\big)\|X_0\|_2\|\mathbf{M}^\top Y\|_2$$

$$\quad + \tfrac{(2T-1)(\beta(2T-1)+(2T-2))}{\beta^2}\|\mathbf{M}^\top Y\|_2^2,$$

$$\Phi_j = \|X_0\|_2 + \tfrac{2j-1}{\beta}\mathbf{M}^\top Y\|_2,$$

$$\Theta_L = \Theta_L,$$

$$\delta_3 = \big((1 + \beta)\|X_0\|_2 + \big(2T - 1 + \tfrac{2T-2}{\beta}\big)\|\mathbf{M}^\top Y\|_2\big).$$

*We have the following upperly bounding property:*

$$(Z - X_T^k)^\top \mathbf{M}^\top (\mathbf{M}X_T^k - Y)$$

$$\leq \Big( -\eta 8\sigma(\delta_3\Theta_L)^2(1 - \sigma(\delta_3\Theta_L))^2\tfrac{\beta_0^2}{\beta^2}\alpha_0^2 + \tfrac{\eta\beta}{2}\Theta_{L-1}^2\Lambda_T\sum_{t=1}^{T-1}\Lambda_t\Big)\tfrac{1}{2}\|\mathbf{M}X_T^k - Y\|_2^2.$$

*Proof.* In our above demonstrations, we have constructed a non-negative coefficient of the upper bound w.r.t. the objective $\tfrac{1}{2}\|\mathbf{M}X_T^k - Y\|_2^2$. To achieve the requirement of the linear convergence

rate, we would like a negative one from our remaining bounding target. We calculate:

$$(Z - X_T^k)^\top \mathbf{M}^\top (\mathbf{M}X_T^k - Y)$$

$$= \left( X_{T-1}^k - \tfrac{1}{\beta}\mathcal{D}(2\sigma(W_L^{k+1}G_{L-1,T}^k)^\top)(\mathbf{M}^\top(\mathbf{M}X_{T-1}^k - Y)) \right.$$

$$\left. - \left( X_{T-1}^k - \tfrac{1}{\beta}\mathcal{D}(2\sigma(W_L^k G_{L-1,T}^k)^\top)(\mathbf{M}^\top(\mathbf{M}X_{T-1}^k - Y)) \right) \right)^\top \mathbf{M}^\top(\mathbf{M}X_T^k - Y), \quad (46)$$

$$= -\tfrac{1}{\beta}\left(\mathbf{M}^\top(\mathbf{M}X_{T-1}^k - Y)\right)^\top \mathcal{D}\left(2\sigma(W_L^{k+1}G_{L-1,T}^k)^\top - 2\sigma(W_L^k G_{L-1,T}^k)^\top\right)$$

$$\left(\mathbf{M}^\top(\mathbf{M}X_{T-1}^k - Y)\right).$$

Similarly, due to Mean Value Theorem, suppose $v_{2,i}^k = \alpha_i(W_L^{k+1}G_{L-1,T}^k)_i + (1-\alpha_i)(W_L^k G_{L-1,T}^k)_i$, $v_{2,i}^k \in [0,1]$, based on Mean Value Theorem, we calculate:

$$2\sigma(W_L^{k+1}G_{L-1,T}^k)_i^\top - 2\sigma(W_L^k G_{L-1,T}^k)_i^\top = \tfrac{\partial(2\sigma(v_{2,i}^k))}{\partial(v_{2,i}^k)_i}(W_L^{k+1}G_{L-1,T}^k)_i - (W_L^k G_{L-1,T}^k)_i.$$

Denote $v_{2,i}^k := [\tfrac{\partial(2\sigma(v_{2,i}^k))}{\partial(v_{2,i}^k)_i}]$, we calculate:

$$\mathcal{D}\left(2\sigma(W_L^{k+1}G_{L-1,T}^k)^\top - 2\sigma(W_L^k G_{L-1,T}^k)^\top\right)$$

$$= \mathcal{D}\left(\left[\tfrac{\partial 2\sigma(v_{2,i}^k)}{\partial v_{2,i}^k}((W_L^{k+1}G_{L-1,T}^k)_i - (W_L^k G_{L-1,T}^k)_i)\right]^\top\right),$$

$$= \mathcal{D}\left(\left[2\sigma(v_{2,i}^k)(1 - \sigma(v_{2,i}^k))((W_L^{k+1} - W_L^k)G_{L-1,T}^k)_i\right]^\top\right),$$

$$= \mathcal{D}\left([2\sigma(v_{2,i}^k)(1 - \sigma(v_{2,i}^k))]^\top\right)\mathcal{D}\left(((W_L^{k+1} - W_L^k)G_{L-1,T}^k)^\top\right),$$

$$\overset{①}{=} -\eta\mathcal{D}\left([2\sigma(v_{2,i}^k)(1 - \sigma(v_{2,i}^k))]^\top\right)\mathcal{D}\left(\tfrac{\partial F}{\partial W_L^k}G_{L-1,T}^k{}^\top\right),$$

where $v_{2\,i}^k := \alpha_i(W_L^{k+1}G_{L-1,T}^k)_i + (1-\alpha)_i(W_L^k G_{L-1,T}^k)_i$ is an interior point between the corresponding entries of $W_L^{k+1}G_{L-1,T}^k$ and $W_L^k G_{L-1,T}^k$. ① is from gradient descent formulation of $W_L^k$ in Equation (8).

Substituting above into Equation (46) yields:

$$(Z - X_T^k)^\top \mathbf{M}^\top(\mathbf{M}X_T^k - Y)$$

$$= \tfrac{\eta}{\beta}\left(\mathbf{M}^\top(\mathbf{M}X_{T-1}^k - Y)\right)^\top \mathcal{D}\left([2\sigma(v_{2,i}^k)(1 - \sigma(v_{2,i}^k))]^\top\right)\mathcal{D}(\tfrac{\partial F}{\partial W_L^k}G_{L-1,T}^k)^\top\left(\mathbf{M}^\top(\mathbf{M}X_T^k - Y)\right),$$

$$= \tfrac{\eta}{\beta}\tfrac{\partial F}{\partial W_L^k}G_{L-1,T}^k\mathcal{D}\left([2\sigma(v_{2,i}^k)(1 - \sigma(v_{2,i}^k))]^\top\right)\mathcal{D}\left(\mathbf{M}^\top(\mathbf{M}X_{T-1}^k - Y)\right)\left(\mathbf{M}^\top(\mathbf{M}X_T^k - Y)\right),$$

Further, we substitute the gradient formulation in Equation (8) and calculate:

$$(Z - X_T^k)^\top \mathbf{M}^\top(\mathbf{M}X_T^k - Y)$$

$$= -\tfrac{\eta^2}{\beta^2}\sum_{t=1}^T\left(\mathbf{M}^\top(\mathbf{M}X_T - Y)\right)^\top\left(\prod_{j=T}^{t+1}\mathbf{I} - \tfrac{1}{\beta}\mathcal{D}(P_j)\mathbf{M}^\top\mathbf{M}\right)$$

$$\mathcal{D}\left((\mathbf{M}^\top(\mathbf{M}X_{t-1} - Y))\right)\mathcal{D}\left(P_t \odot (1 - P_t/2)\right)G_{L-1,t}{}^\top G_{L-1,T}^k \quad (47)$$

$$\mathcal{D}\left([2\sigma(v_{2,i}^k)(1 - \sigma(v_{2,i}^k))]^\top\right)\mathcal{D}\left(\mathbf{M}^\top(\mathbf{M}X_{T-1}^k - Y)\right)\left(\mathbf{M}^\top(\mathbf{M}X_T^k - Y)\right),$$

$$= -\tfrac{\eta}{\beta^2}(\mathbf{M}X_T^k - Y)^\top \mathbf{M}\mathbf{B}_T^k\mathbf{M}^\top(\mathbf{M}X_T^k - Y),$$

where $\mathbf{B}_T^k$ is defined by:

$$\mathbf{B}_T^k$$

$$= \sum_{t=1}^T\left(\prod_{j=T}^{t+1}\mathbf{I} - \tfrac{1}{\beta}\mathcal{D}(P_j^k)\mathbf{M}^\top\mathbf{M}\right)\mathcal{D}\left(\mathbf{M}^\top(\mathbf{M}X_{t-1}^k - Y)\right)\mathcal{D}\left(P_t^k \odot (1 - P_t^k/2)\right)G_{L-1,t}^k{}^\top$$

$$G_{L-1,T}^k\mathcal{D}\left([2\sigma(v_{2,i}^k)(1 - \sigma(v_{2,i}^k))]^\top\right)\mathcal{D}\left(\mathbf{M}^\top(\mathbf{M}X_{T-1}^k - Y)\right).$$

We discuss the definite property of $\mathbf{B}_T^k$ case-by-case.

**Case 1:** $t = T$.  $\Pi_{j=T}^{T+1}\mathbf{I} - \frac{1}{\beta}\mathcal{D}(P_j)\mathbf{M}^\top\mathbf{M}$ degenerates to be 1. The Equation (47) becomes:

$$
\begin{aligned}
&[(Z - X_T^k)^\top\mathbf{M}^\top(\mathbf{M}X_T^k - Y)]_{\text{Part 1}}\\
&= -\tfrac{\eta}{\beta^2}(\mathbf{M}X_T^k - Y)^\top\mathbf{M}\\
&\qquad\mathcal{D}\big(\mathbf{M}^\top(\mathbf{M}X_{T-1}^k - Y)\big)\\
&\qquad\mathcal{D}\big(P_T^k\odot(1 - P_T^k/2)\big)\\
&\qquad G_{L-1,T}^k{}^\top G_{L-1,T}^k\\
&\qquad\mathcal{D}\big([2\sigma(v_{2,i}^k)(1 - \sigma(v_{2,i}^k))]^\top\big)\\
&\qquad\mathcal{D}\big(\mathbf{M}^\top(\mathbf{M}X_{T-1}^k - Y)\big)\mathbf{M}^\top(\mathbf{M}X_T^k - Y),
\end{aligned}
\tag{48}
$$

We first present the following corollary to show that there exists a negative upper bound of $[(Z - X_T^k)^\top\mathbf{M}^\top(\mathbf{M}X_T^k - Y)]_{\text{Part 1}}$:

**Corollary A.11.** *RHS of Equation* (48) $< 0$ *if* $\lambda_{\min}(G_{L-1,T}^k{}^\top G_{L-1,T}^k) > 0$.

*Proof.* Due to definition of eigenvalue and Cauchy-Schwarz inequality, we calculate:

$$
\begin{aligned}
&(\mathbf{M}X_T^k - Y)^\top\mathbf{M}\\
&\mathcal{D}\big(\mathbf{M}^\top(\mathbf{M}X_{T-1}^k - Y)\big)\\
&\mathcal{D}\big(P_T^k\odot(1 - P_T^k/2)\big)G_{L-1,T}^k{}^\top G_{L-1,T}^k\mathcal{D}\big([2\sigma(v_{2,i}^k)(1 - \sigma(v_{2,i}^k))]^\top\big)\\
&\mathcal{D}\big(\mathbf{M}^\top(\mathbf{M}X_{T-1}^k - Y)\big)\mathbf{M}^\top(\mathbf{M}X_T^k - Y),\\
&\geq\big(P_T^k\odot(1 - P_T^k/2)\big)_{\min}\big([2\sigma(v_{2,i}^k)(1 - \sigma(v_{2,i}^k))]^\top\big)_{\min}\\
&\qquad\lambda_{\min}(G_{L-1,T}^k{}^\top G_{L-1,T}^k)\lambda_{\min}(\mathbf{M}\mathbf{M}^\top)\|\mathbf{M}^\top(\mathbf{M}X_T^k - Y)\|_2^2,\\
&\overset{\text{\textcircled{1}}}{>}0,
\end{aligned}
$$

where \textcircled{1} is due to Sigmoid function is non-negative, $\lambda_{\min}(G_{L-1,T}^k{}^\top G_{L-1,T}^k) > 0$, and $\lambda_{\min}(\mathbf{M}\mathbf{M}^\top) > 0$ by definition. Thus, $(Z - X_T^k)^\top\mathbf{M}^\top(\mathbf{M}X_T^k - Y) < 0$ by nature. $()_{\min}$ means the minimal value among all entries. $\square$

To get an upper bound, we expect $G_{L-1,T}^k{}^\top G_{L-1,T}^k$ to be positive definition, in which we require $n_{L-1} \geq N$. Thus, we can easily get the upper bound from its minimal eigenvalue.

Based on Corollary A.11, we calculate the negative lower bound of Equation (48) by:

$$
\begin{aligned}
&(Z - X_T^k)^\top\mathbf{M}^\top(\mathbf{M}X_T^k - Y)\\
&\leq -\tfrac{\eta}{\beta^2}\big(P_T^k\odot(1 - P_T^k/2)\big)_{\min}\big([2\sigma(v_{2,i}^k)(1 - \sigma(v_{2,i}^k))]^\top\big)_{\min}\\
&\qquad\lambda_{\min}(G_{L-1,T}^k{}^\top G_{L-1,T}^k)\lambda_{\min}(\mathbf{M}\mathbf{M}^\top)\|\mathbf{M}^\top(\mathbf{M}X_T^k - Y)\|_2^2,
\end{aligned}
\tag{49}
$$

The remaining task is to calculate $\big(P_T^k\odot(1 - P_T^k/2)\big)_{\min}$ and $\big([2\sigma(v_{2,i}^k)(1 - \sigma(v_{2,i}^k))]^\top\big)_{\min}$. We achieve that by calculating the values on the boundary of closed sets.

First, denote $v_3^k := W_L^k G_{L-1,T}^k$, we represent $P_T^k\odot(1 - P_T^k/2)$ by:

$$
P_T^k\odot(1 - P_T^k/2) = 2\sigma(v_3^k)^\top\odot(1 - \sigma(v_3^k))^\top.
$$

Since the Sigmoid function is a coordinate-wise non-decreasing function, we can straightforwardly find $\big([2\sigma(v_{2,i}^k)(1 - \sigma(v_{2,i}^k))]^\top\big)_{\min}$ and $(2\sigma(v_3^k)^\top\odot(1 - \sigma(v_3^k))^\top)_{\min}$ by on the closed sets of $v_2^k$ and $v_3^k$, respectively, which is achieved by the following lemma.

**Lemma A.12.** *For some $b, B \in \mathbb{R}^{k2}$[2], $\forall v^k, b \leq v^k \leq B$, we calculate $(2\sigma(v^k)^\top \odot (1 - \sigma(v^k))^\top)_{\min}$ by:*

$$(2\sigma(v^k)^\top \odot (1 - \sigma(v^k))^\top)_{\min} = \begin{cases} \min\left(2\sigma(b)(1 - \sigma(b))^\top, 2\sigma(B)(1 - \sigma(B))^\top\right) & -b \neq B, \\ 2\sigma(B)(1 - \sigma(B)) & -b = B. \end{cases}$$

*Proof.* Since $\sigma(x) \in (0, 1) \forall x$, $\mathcal{D}(2\sigma(x) \odot (1 - \sigma(x)))$ is a quadratic function w.r.t. $x$. Since $\sigma(x) \in (0, 1) \forall x$, $\mathcal{D}(2\sigma(x) \odot (1 - \sigma(x))) > 0$. Since the coefficient before the $x^2$ term is negative, its lower bound is either the value on the boundary or 0.

Since $\sigma(b), \sigma(B) \in (0, 1)$, if $-b \neq B$, the lower bound is the smaller one, i.e., $\min(2\sigma(b) \odot (1 - \sigma(b)), 2\sigma(B) \odot (1 - \sigma(B)))$. Otherwise, since both $\sigma(x)$ and $\mathcal{D}(2\sigma(x) \odot (1 - \sigma(x)))$ are symmetric around $\frac{1}{2}$, we have $2\sigma(B) \odot (1 - \sigma(B)) = 2\sigma(b) \odot (1 - \sigma(b))$. $\square$

Further, we calculate the bounds of $v_2^k$ and $v_3^k$ and invoke Lemma A.12 to get $\left([2\sigma(v_{2,i}^k)(1 - \sigma(v_{2,i}^k))]^\top\right)_{\min}$ and $(2\sigma(v_3^k)^\top \odot (1 - \sigma(v_3^k))^\top)_{\min}$.

We present the following two lemmas to show the closed sets that $v_2^k$ and $v_3^k$ belong to.

**Lemma A.13.** *Denote $\ell \in [L]$, for some $\bar{\lambda}_\ell \in \mathbb{R}$, we assume $\|W_\ell^k\|_2 \leq \bar{\lambda}_\ell$. We define the following quantity:*

$$\delta_3 = \left((1 + \beta)\|X_0\|_2 + \left(2T - 1 + \tfrac{2T-2}{\beta}\right)\|\mathbf{M}^\top Y\|_2\right),$$

$$\Theta_L = \prod_{\ell=1}^L \bar{\lambda}_\ell.$$

*For $v_{2\,i}^k := \alpha_i(W_L^{k+1} G_{L-1,T}^k)_i + (1 - \alpha_i)(W_L^k G_{L-1,T}^k)_i, \alpha_i \in [0, 1]$, $v_2^k$ belongs to the following closed set:*

$$v_2^k \in [-\delta_3 \Theta_L, \delta_3 \Theta_L].$$

*Proof.* We calculate $v_2^k$'s upper bound by:

$$
\begin{aligned}
\|v_2^k\|_\infty &= \|\alpha \odot (W_L^{k+1} G_{L-1,T}^k) + (1 - \alpha) \odot (W_L^k G_{L-1,T}^k)\|_\infty, \\
&= \max_i \|\alpha_i(W_L^{k+1} G_{L-1,T}^k)_i + (1 - \alpha_i)(W_L^k G_{L-1,T}^k)_i\|_\infty, \\
&\overset{\textcircled{1}}{\leq} \max_i \alpha_i \|(W_L^{k+1} G_{L-1,T}^k)_i\|_\infty + (1 - \alpha_i)\|(W_L^k G_{L-1,T}^k)_i\|_\infty, \\
&\overset{\textcircled{2}}{\leq} \max_i \max(\|(W_L^{k+1} G_{L-1,T}^k)_i\|_\infty, \|(W_L^k G_{L-1,T}^k)_i\|_\infty), \\
&= \max(\max_i \|(W_L^{k+1} G_{L-1,T}^k)_i\|_\infty, \max_i \|(W_L^k G_{L-1,T}^k)_i\|_\infty), \\
&\leq \max(\|W_L^{k+1} G_{L-1,T}^k\|_\infty, \|W_L^k G_{L-1,T}^k\|_\infty),
\end{aligned}
\tag{50}
$$

where $\textcircled{1}$ is due to triangle inequality and $\textcircled{2}$ is due to $\alpha_i \in [0, 1]$ and upper bound of NN's inner output in Lemma A.5.

We calculate the bound of $\|W_L^{k+1} G_{L-1,T}^k\|_2$ by:

$$
\begin{aligned}
\|W_L^{k+1} G_{L-1,T}^k\|_\infty &\overset{\textcircled{1}}{\leq} \|W_L^{k+1}\|_2 \|G_{L-1,T}^k\|_2, \\
&\overset{\textcircled{2}}{\leq} \bar{\lambda}_L\left((1 + \beta)\|X_0\|_2 + \left(2T - 1 + \tfrac{2T-2}{\beta}\right)\|\mathbf{M}^\top Y\|_2\right)\prod_{\ell=1}^{L-1} \bar{\lambda}_\ell, \\
&= \underbrace{\left((1 + \beta)\|X_0\|_2 + \left(2T - 1 + \tfrac{2T-2}{\beta}\right)\|\mathbf{M}^\top Y\|_2\right)}_{\delta_3} \underbrace{\prod_{\ell=1}^L \bar{\lambda}_\ell}_{\Theta_L},
\end{aligned}
$$

where $\textcircled{1}$ is due to Cauchy-Schwarz inequality and $\textcircled{2}$ is due to definition and upper bound of NN's inner output in Lemma A.5. Similarly, we can get $\|W_L^{k+1} G_{L-1,T}^k\|_2 \leq \delta_3 \Theta_L$.

---

[2]$\mathbb{R}^k$ means the space at training iteration $k$.

Substituting back to Equation (50) yields:

$$\|v_2^k\|_\infty \le \delta_3 \Theta_L.$$

Thus, we have the following bound for vector $v_2^k$ by nature:

$$-\delta_3 \Theta_L \le v_2^k \le \delta_3 \Theta_L.$$

It is worth noting that the above lower bound is non-trivial since we cannot have $v_2^k \ge 0$, which can be easily violated by a little perturbation from gradient descent.

$\square$

**Lemma A.14.** *Denote $\ell \in [L]$, for some $\bar\lambda_\ell \in \mathbb{R}$, we assume $\|W_\ell^k\|_2 \le \bar\lambda_\ell$. We define the following quantity:*

$$\delta_3 = \big((1+\beta)\|X_0\|_2 + \big(2T - 1 + \tfrac{2T-2}{\beta}\big)\|\mathbf{M}^\top Y\|_2\big),$$

$$\Theta_L = \prod_{\ell=1}^L \bar\lambda_\ell.$$

*For $v_3^k := W_L^k G_{L-1,T}^k, \forall k$, $v_3^k$ belongs to the following closed set:*

$$v_3^k \in [-\delta_3 \Theta_L, \delta_3 \Theta_L].$$

*Proof.* We prove the lemma by a similar method. We calculate the bound of $\|W_L^k G_{L-1,T}^k\|_2$ by:

$$
\begin{aligned}
\|v_3^k\|_\infty &= \|W_L^k G_{L-1,T}^k\|_\infty \\
&\overset{\text{①}}{\le} \|W_L^k\|_2 \|G_{L-1,T}^k\|_2, \\
&\overset{\text{②}}{\le} \bar\lambda_L \big((1+\beta)\|X_0\|_2 + \big(2T-1+\tfrac{2T-2}{\beta}\big)\|\mathbf{M}^\top Y\|_2\big)\prod_{\ell=1}^{L-1}\bar\lambda_\ell, \\
&= \underbrace{\big((1+\beta)\|X_0\|_2 + \big(2T-1+\tfrac{2T-2}{\beta}\big)\|\mathbf{M}^\top Y\|_2\big)}_{\delta_3}\underbrace{\prod_{\ell=1}^L\bar\lambda_\ell}_{\Theta_L},
\end{aligned}
$$

where ① is due to Cauchy-Schwarz inequality and ② is due to definition and upper bound of NN's inner output in Lemma A.5.

We have the following bound for $v_3^k$ by nature:

$$-\delta_3 \Theta_L \le v_3^k \le \delta_3 \Theta_L.$$

$\square$

We calculate $\big([2\sigma(v_{2,i}^k)(1-\sigma(v_{2,i}^k))]^\top\big)_{\min}$ by substituting Lemma A.13 into Lemma A.12:

$$\big([2\sigma(v_{2,i}^k)(1-\sigma(v_{2,i}^k))]^\top\big)_{\min} = 2\sigma(\delta_3\Theta_L)(1-\sigma(\delta_3\Theta_L)).$$

Similarly, we get $\big(P_T^k \odot (1-P_T^k/2)\big)$ by substituting Lemma A.14 into Lemma A.12:

$$\big(P_T^k \odot (1-P_T^k/2)\big)_{\min} = 2\sigma(\delta_3\Theta_L)(1-\sigma(\delta_3\Theta_L)).$$

Substituting the above results into Equation (49) and Equation (48) yields:

$$
\begin{aligned}
&[(Z-X_T^k)^\top \mathbf{M}^\top(\mathbf{M}X_T^k - Y)]_{\text{Part 1}} \\
&\le -\tfrac{\eta}{\beta^2}\big(P_T^k \odot (1-P_T^k/2)\big)_{\min}\big([2\sigma(v_{2,i}^k)(1-\sigma(v_{2,i}^k))]^\top\big)_{\min} \\
&\quad \lambda_{\min}(G_{L-1,T}^k{}^\top G_{L-1,T}^k)\lambda_{\min}(\mathbf{M}\mathbf{M}^\top)\|\mathbf{M}^\top(\mathbf{M}X_T^k - Y)\|_2^2, \\
&\le -\tfrac{\eta}{\beta^2}4\sigma(\delta_3\Theta_L)^2(1-\sigma(\delta_3\Theta_L))^2\lambda_{\min}(G_{L-1,T}^k{}^\top G_{L-1,T}^k)\lambda_{\min}(\mathbf{M}\mathbf{M}^\top)\|\mathbf{M}^\top(\mathbf{M}X_T^k - Y)\|_2^2, \\
&\overset{\text{①}}{\le} -\eta 8\sigma(\delta_3\Theta_L)^2(1-\sigma(\delta_3\Theta_L))^2\tfrac{\beta_0^2}{\beta^2}\alpha_0^2\tfrac{1}{2}\|\mathbf{M}X_T^k - Y\|_2^2,
\end{aligned}
$$
(51)

where ① is from definition.

**Case 2:** $t < T$.  We derive the upper bound of above term by Cauchy-Schwarz inequality:

$$[(Z - X_T^k)^\top \mathbf{M}^\top (\mathbf{M} X_T^k - Y)]_{\text{Part 2}}$$

$$= -\tfrac{\eta}{\beta^2}(\mathbf{M} X_T^k - Y)^\top \mathbf{M}\Big(\sum_{t=1}^{T-1}\big(\prod_{j=T}^{t+1}\mathbf{I} - \tfrac{1}{\beta}\mathcal{D}(P_j^k)\mathbf{M}^\top\mathbf{M}\big)$$

$$\mathcal{D}\big(\mathbf{M}^\top(\mathbf{M} X_{t-1}^k - Y)\big)\mathcal{D}\big(P_t^k \odot (1 - P_t^k/2)\big){G_{L-1,t}^k}^\top G_{L-1,T}^k$$

$$\mathcal{D}\big([2\sigma(v_{2,i}^k)(1 - \sigma(v_{2,i}^k))]^\top\big)\mathcal{D}\big(\mathbf{M}^\top(\mathbf{M} X_{T-1}^k - Y)\big)\Big)\mathbf{M}^\top(\mathbf{M} X_T^k - Y),$$

$$\overset{①}{\leq}\tfrac{\eta}{\beta^2}\Big\|\sum_{t=1}^{T-1}\big(\prod_{j=T}^{t+1}\mathbf{I} - \tfrac{1}{\beta}\mathcal{D}(P_j^k)\mathbf{M}^\top\mathbf{M}\big)$$

$$\mathcal{D}\big(\mathbf{M}^\top(\mathbf{M} X_{t-1}^k - Y)\big)\mathcal{D}\big(P_t^k \odot (1 - P_t^k/2)\big){G_{L-1,t}^k}^\top G_{L-1,T}^k$$

$$\mathcal{D}\big([2\sigma(v_{2,i}^k)(1 - \sigma(v_{2,i}^k))]^\top\big)\mathcal{D}\big(\mathbf{M}^\top(\mathbf{M} X_{T-1}^k - Y)\big)\Big\|_2\|\mathbf{M}\mathbf{M}^\top\|_2\|\mathbf{M} X_T^k - Y\|_2^2,$$

$$\leq\tfrac{\eta}{\beta^2}\sum_{t=1}^{T-1}\Big\|\big(\prod_{j=T}^{t+1}\mathbf{I} - \tfrac{1}{\beta}\mathcal{D}(P_j^k)\mathbf{M}^\top\mathbf{M}\big)\Big\|_2$$

$$\|\mathcal{D}\big(P_t^k \odot (1 - P_t^k/2)\big)\|_2\|G_{L-1,t}^k\|_2\|G_{L-1,T}^k\|_2\|\mathcal{D}\big([2\sigma(v_{2,i}^k)(1 - \sigma(v_{2,i}^k))]^\top\big)\|_2$$

$$\|\mathcal{D}\big(\mathbf{M}^\top(\mathbf{M} X_{t-1}^k - Y)\big)\|_2\|\mathcal{D}\big(\mathbf{M}^\top(\mathbf{M} X_{T-1}^k - Y)\big)\|_2\|\mathbf{M}\mathbf{M}^\top\|_2\|\mathbf{M} X_T^k - Y\|_2^2,$$

$$\overset{②}{\leq}\tfrac{\eta}{\beta}\sum_{t=1}^{T-1}\|\mathcal{D}\big(P_t^k \odot (1 - P_t^k/2)\big)\|_2\|G_{L-1,t}^k\|_2\|G_{L-1,T}^k\|_2\|\mathcal{D}\big([2\sigma(v_{2,i}^k)(1 - \sigma(v_{2,i}^k))]^\top\big)\|_2$$

$$\|\mathcal{D}\big(\mathbf{M}^\top(\mathbf{M} X_{t-1}^k - Y)\big)\|_2\|\mathcal{D}\big(\mathbf{M}^\top(\mathbf{M} X_{T-1}^k - Y)\big)\|_2\|\mathbf{M} X_T^k - Y\|_2^2,$$

$$\overset{③}{\leq}\tfrac{\eta}{4\beta}\sum_{t=1}^{T-1}\|G_{L-1,t}^k\|_2\|G_{L-1,T}^k\|_2\|\mathcal{D}\big(\mathbf{M}^\top(\mathbf{M} X_{t-1}^k - Y)\big)\|_2\|\mathcal{D}\big(\mathbf{M}^\top(\mathbf{M} X_{T-1}^k - Y)\big)\|_2$$

$$\|\mathbf{M} X_T^k - Y\|_2^2,$$

$$\leq\tfrac{\eta}{4\beta}(\beta(\|X_0\|_2 + \tfrac{2T}{\beta}\|\mathbf{M}^\top Y\|_2) + \|\mathbf{M}^\top Y\|_2)\|G_{L-1,T}^k\|_2$$

$$\sum_{t=1}^{T-1}\|G_{L-1,t}^k\|_2(\beta(\|X_0\|_2 + \tfrac{2t}{\beta}\|\mathbf{M}^\top Y\|_2) + \|\mathbf{M}^\top Y\|_2)\|\mathbf{M} X_T^k - Y\|_2^2,$$

$$\leq\tfrac{\eta}{4\beta}(\beta(\|X_0\|_2 + \tfrac{2T-2}{\beta}\|\mathbf{M}^\top Y\|_2) + \|\mathbf{M}^\top Y\|_2)\big((1 + \beta)\|X_0\|_2 + \big(2T - 1 + \tfrac{2T-2}{\beta}\big)\|\mathbf{M}^\top Y\|_2\big)$$

$$\prod_{s=1}^{L-1}\bar{\lambda}_s\sum_{t=1}^{T-1}\big((1 + \beta)\|X_0\|_2 + \big(2t - 1 + \tfrac{2t-2}{\beta}\big)\|\mathbf{M}^\top Y\|_2\big)$$

$$\prod_{s=1}^{L-1}\bar{\lambda}_s(\beta(\|X_0\|_2 + \tfrac{2t-2}{\beta}\|\mathbf{M}^\top Y\|_2) + \|\mathbf{M}^\top Y\|_2)\|\mathbf{M} X_T^k - Y\|_2^2,$$

where ① is due to Cauchy-Schwarz inequality. It is worth noting that ① is non-trivial since $\mathbf{B}_{T-1}^k$ is non-necessarily to be positive definite. ② is due to upper bound of NN's output in Lemma A.1. ③ is based on the Sigmoid function is bounded.

Further, due to the definition of the quantities, we calculate:

$$[(Z - X_T^k)^\top \mathbf{M}^\top (\mathbf{M} X_T^k - Y)]_{\text{Part 2}}$$

$$\leq\tfrac{\eta\beta}{4}$$

$$\underbrace{\big((1 + \beta)\|X_0\|_2^2 + \big((4T - 3)(1 + \tfrac{1}{\beta}) + 1\big)\|X_0\|_2\|\mathbf{M}^\top Y\|_2 + \tfrac{(2T-1)(\beta(2T-1)+(2T-2))}{\beta^2}\|\mathbf{M}^\top Y\|_2^2\big)}_{\Lambda_T}$$

$$\sum_{t=1}^{T-1}$$

$$\underbrace{\big((1 + \beta)\|X_0\|_2^2 + \big((4t - 3)(1 + \tfrac{1}{\beta}) + 1\big)\|X_0\|_2\|\mathbf{M}^\top Y\|_2 + \tfrac{(2T-1)(\beta(2T-1)+(2T-2))}{\beta^2}\|\mathbf{M}^\top Y\|_2^2\big)}_{\Lambda_t}$$

$$\Theta_{L-1}^2\|\mathbf{M} X_T^k - Y\|_2^2,$$

$$=\tfrac{\eta\beta}{2}\Theta_{L-1}^2\Lambda_T\sum_{t=1}^{T-1}\Lambda_t\tfrac{1}{2}\|\mathbf{M} X_T^k - Y\|_2^2.$$

(52)

Combining the two parts in Equation (51) and Equation (52) yields:

$$(Z - X_T^k)^\top \mathbf{M}^\top (\mathbf{M} X_T^k - Y)$$

$$\leq\Big(\tfrac{\eta\beta}{2}\Theta_{L-1}^2\Lambda_T\sum_{t=1}^{T-1}\Lambda_t - \eta 8\sigma(\delta_3\Theta_L)^2(1 - \sigma(\delta_3\Theta_L))^2\tfrac{\beta_0^2}{\beta^2}\alpha_0^2\Big)\tfrac{1}{2}\|\mathbf{M} X_T^k - Y\|_2^2.$$

$\square$

Using quantities from Equation (12), substituting the upper bounds in Lemma A.8, Lemma A.9, and Lemma A.10 into Equation (39), we calculate:

$F([W]^{k+1})$

$=F([W]^k) + \frac{1}{2}\|\mathbf{M}X_T^{k+1} - \mathbf{M}X_T^k\|_2^2 + (\mathbf{M}X_T^{k+1} - \mathbf{M}X_T^k)^\top(\mathbf{M}X_T^k - Y),$

$\leq F([W]^k) + \frac{\beta^2\eta^2}{16}(\delta_1{}^T)^2\left(S_{\Lambda,T}\right)^2\left(\Theta_L^2\sum_{\ell=1}^L\bar{\lambda}_\ell^{-2}\right)^2\frac{1}{2}\|\mathbf{M}X_T^k - Y\|_2$

$\quad + \frac{\beta\eta}{2}(\Lambda_T + \delta_2)\Theta_L^2 S_{\bar{\lambda},L}S_{\Lambda,T}\frac{1}{2}\|\mathbf{M}X_T^k - Y\|_2^2$

$\quad + \left(-\eta 8\sigma(\delta_3\Theta_L)^2(1-\sigma(\delta_3\Theta_L))^2\frac{\beta_0^2}{\beta^2}\alpha_0^2 + \frac{\eta\beta}{2}\Theta_{L-1}^2\Lambda_T\sum_{t=1}^{T-1}\Lambda_t\right)\frac{1}{2}\|\mathbf{M}X_T^k - Y\|_2^2,$

$\overset{①}{=}F([W]^k) + \frac{\beta^2\eta^2}{16}(\delta_1{}^T)^2\left(S_{\Lambda,T}\right)^2\left(\Theta_L^2\sum_{\ell=1}^L\bar{\lambda}_\ell^{-2}\right)^2 F([W]^k)$

$\quad + \frac{\beta\eta}{2}(\Lambda_T + \delta_2)\Theta_L^2 S_{\bar{\lambda},L}S_{\Lambda,T}F([W]^k)$

$\quad + \left(-\eta 8\sigma(\delta_3\Theta_L)^2(1-\sigma(\delta_3\Theta_L))^2\frac{\beta_0^2}{\beta^2}\alpha_0^2 + \frac{\eta\beta}{2}\Theta_{L-1}^2\Lambda_T\sum_{t=1}^{T-1}\Lambda_t\right)F([W]^k),$

$=\left(1 + \frac{\eta^2\beta^2}{16}(\delta_1{}^T)^2\left(S_{\Lambda,T}\right)^2\left(\Theta_L^2\sum_{\ell=1}^L\bar{\lambda}_\ell^{-2}\right)^2 + \frac{\eta\beta}{2}(\Lambda_T + \delta_2)S_{\Lambda,T}\Theta_L^2 S_{\bar{\lambda},L}\right.$

$\quad \left. + \frac{\eta\beta}{2}\Theta_{L-1}^2\Lambda_T\sum_{t=1}^{T-1}\Lambda_t - \eta 8\sigma(\delta_3\Theta_L)^2(1-\sigma(\delta_3\Theta_L))^2\frac{\beta_0^2}{\beta^2}\alpha_0^2\right)F([W]^k),$

$\overset{②}{\leq}\left(1 + \eta\beta(\Lambda_T + \delta_2)S_{\Lambda,T}\Theta_L^2 S_{\bar{\lambda},L} + \frac{\eta\beta}{2}\Theta_{L-1}^2\Lambda_T\sum_{t=1}^{T-1}\Lambda_t - \eta 8\sigma(\delta_3\Theta_L)^2(1-\sigma(\delta_3\Theta_L))^2\frac{\beta_0^2}{\beta^2}\alpha_0^2\right)$

$\quad F([W]^k),$

$=\left(1 - \eta\left(8\sigma(\delta_3\Theta_L)^2(1-\sigma(\delta_3\Theta_L))^2\frac{\beta_0^2}{\beta^2}\alpha_0^2 - \beta(\Lambda_T + \delta_2)S_{\Lambda,T}\Theta_L^2 S_{\bar{\lambda},L} - \frac{\beta}{2}\Theta_{L-1}^2\Lambda_T\sum_{t=1}^{T-1}\Lambda_t\right)\right)$

$\quad F([W]^k),$

$\overset{③}{\leq}\left(1 - \eta\underbrace{4\sigma(\delta_3\Theta_L)^2(1-\sigma(\delta_3\Theta_L))^2\frac{\beta_0^2}{\beta^2}\alpha_0^2}_{4\eta\frac{\beta_0^2}{\beta^2}\delta_4}\right)F([W]^k),$

where ① is due to the definition of objective. ② is due to upper bound of learning rate in Equation (14a) and $\delta_1{}^T = \delta_2 + \sum_{j=1}^T\Lambda_j$ in definition. ③ is due to the lower bound of the least eigenvalue $\alpha_0$ in Equation (13b).

Due to learning rate's upper bound in Equation (14b), we know $0 < 1 - \eta 4\eta\frac{\beta_0^2}{\beta^2}\delta_4 < 1$, which yields the following linear rate by nature:

$$F([W]^k) \leq (1 - \eta 4\eta\frac{\beta_0^2}{\beta^2}\delta_4)^k F([W]^0).$$

$\square$

# B  Details for Initialization

## B.1  Preliminary

To begin with, we define the following quantities:

$\delta_5 = \sigma\left(\left(2T - 1 + \frac{2T-2}{\beta}\right)\|\mathbf{M}^\top Y\|_2\Theta_L\right)^{-2}\left(1 - \sigma\left(\left(2T - 1 + \frac{2T-2}{\beta}\right)\|\mathbf{M}^\top Y\|_2\Theta_L\right)\right)^{-2},$

$\delta_6 = \sigma_{\min}\left([\sum_{t=1}^{T-1}(\mathbf{I} - \frac{1}{\beta}\mathbf{M}^\top\mathbf{M})^{T-t}\mathbf{M}^\top Y | \mathbf{M}^\top(\mathbf{M}(\sum_{t=1}^{T-1}(\mathbf{I} - \frac{1}{\beta}\mathbf{M}^\top\mathbf{M})^{T-t}\mathbf{M}^\top Y) - Y)]\right),$

$\delta_7 = \sigma_{\min}(\sum_{t=1}^{T-1}(\mathbf{I} - \frac{1}{\beta}\mathbf{M}^\top\mathbf{M})^{T-t}).$

**Analysis for the numerical stability of $\delta_5$.** $\delta_5$ is a function with $\Lambda_t$, which is also enlarged w.r.t. $e^L$. In general, it is possible to push $\sigma\big(1 - \sigma\big(\big(2T - 1 + \frac{2T-2}{\beta}\big)\|\mathbf{M}^\top Y\|_2\Theta_L\big)\big)$ to zero and let RHS of above inequality to be $\infty$ when $e^L \to \infty$. As presented in the lemma, we claim that the required $e$ is not necessarily to be $\infty$. Thus, $\delta_5$ can be regarded as a $\mathcal{O}(e^{L-1}) \ll \infty$ constant. In the following proofs, we demonstrate that it holds since $e$ is finite.

We calculate the following exact formulations of the quantities defined in Theorem 4.3:

$$
\begin{aligned}
\Lambda_T =& (1+\beta)\|X_0\|_2^2 + \big((4T-3)(1+\tfrac{1}{\beta}) + 1\big)\|X_0\|_2\|\mathbf{M}^\top Y\|_2 \\
&+ \tfrac{(2T-1)(\beta(2T-1)+(2T-2))}{\beta^2}\|\mathbf{M}^\top Y\|_2^2, \\
=& \tfrac{4(\beta+1)}{\beta^2}\|\mathbf{M}^\top Y\|_2^2 T^2 + \Big(\tfrac{4(1+\beta)}{\beta}\|X_0\|_2\|\mathbf{M}^\top Y\|_2 - \tfrac{4\beta+6}{\beta^2}\|\mathbf{M}^\top Y\|_2^2\Big)T \\
&+ (1+\beta)\|X_0\|_2^2 - (2+\tfrac{3}{\beta})\|X_0\|_2\|\mathbf{M}^\top Y\|_2 + \tfrac{\beta+2}{\beta^2}\|\mathbf{M}^\top Y\|_2^2, \\
\overset{①}{=}& \tfrac{4(\beta+1)}{\beta^2}\|\mathbf{M}^\top Y\|_2^2 T^2 - \tfrac{4\beta+6}{\beta^2}\|\mathbf{M}^\top Y\|_2^2 T + \tfrac{\beta+2}{\beta^2}\|\mathbf{M}^\top Y\|_2^2,
\end{aligned}
\tag{53}
$$

where ① is due to $X_0 = 0$ and

$$
\begin{aligned}
\textstyle\sum_{i=1}^{T}\Lambda_i =& \textstyle\sum_{i=1}^{T}(1+\beta)\|X_0\|_2^2 + \big((4i-3)(1+\tfrac{1}{\beta}) + 1\big)\|X_0\|_2\|\mathbf{M}^\top Y\|_2 \\
&+ \tfrac{(2i-1)(\beta(2i-1)+(2i-2))}{\beta^2}\|\mathbf{M}^\top Y\|_2^2 \\
=& \tfrac{4(\beta+1)}{3\beta^2}\|\mathbf{M}^\top Y\|_2^2 T^3 + \Big(\tfrac{2(1+\beta)}{\beta}\|X_0\|_2\|\mathbf{M}^\top Y\|_2 - \tfrac{1}{\beta^2}\|\mathbf{M}^\top Y\|_2^2\Big)T^2 \\
&+ \Big((1+\beta)\|X_0\|_2^2 - \tfrac{1}{\beta}\|X_0\|_2\|\mathbf{M}^\top Y\|_2 - \tfrac{\beta+1}{3\beta^2}\|\mathbf{M}^\top Y\|_2^2\Big)T, \\
\overset{①}{=}& \tfrac{4(\beta+1)}{3\beta^2}\|\mathbf{M}^\top Y\|_2^2 T^3 - \tfrac{1}{\beta^2}\|\mathbf{M}^\top Y\|_2^2 T^2 - \tfrac{\beta+1}{3\beta^2}\|\mathbf{M}^\top Y\|_2^2 T,
\end{aligned}
\tag{54}
$$

where ① is due to $X_0 = 0$.

Then, we analyze the expansion of $\sigma_{\min}(G_{L-1,T}^0)$ w.r.t. $[W]_L = e[W]_L$. Due to the one line form equation of L2O model in Equation (22), we have $\sigma_{\min}(G_{L-1,T}^0)$ is calculated by:

$$
\sigma_{\min}(G_{L-1,T}^0) = \sigma_{\min}\big(\operatorname{ReLU}(\operatorname{ReLU}([X_{T-1}^0, \mathbf{M}^\top(\mathbf{M}X_{T-1}^0 - Y)]W_1^{0\top})\cdots W_{L-1}^{0}{}^\top)\big),
$$

where due to Equation (22), $X_{T-1}^0$ is given by:

$$
\begin{aligned}
X_{T-1}^0 =& \textstyle\prod_{t=T-1}^{1}(\mathbf{I} - \tfrac{1}{\beta}\mathcal{D}(P_t^0)\mathbf{M}^\top\mathbf{M})X_0 + \tfrac{1}{\beta}\sum_{t=1}^{T-1}\prod_{s=T-1}^{t+1}(\mathbf{I} - \tfrac{1}{\beta}\mathcal{D}(P_s^0)\mathbf{M}^\top\mathbf{M})\mathcal{D}(P_t^0)\mathbf{M}^\top Y, \\
\overset{①}{=}& (\mathbf{I} - \tfrac{1}{\beta}\mathbf{M}^\top\mathbf{M})^{T-1}X_0 + \tfrac{1}{\beta}\sum_{t=1}^{T-1}(\mathbf{I} - \tfrac{1}{\beta}\mathbf{M}^\top\mathbf{M})^{T-t}\mathbf{M}^\top Y, \\
\overset{②}{=}& \tfrac{1}{\beta}\sum_{t=1}^{T-1}(\mathbf{I} - \tfrac{1}{\beta}\mathbf{M}^\top\mathbf{M})^{T-t}\mathbf{M}^\top Y,
\end{aligned}
\tag{55}
$$

where ① is due to $P_t = \sigma(\mathbf{0}) = \mathbf{I}$ since $W_L = 0$. The result shows that $X_{T-1}^0$ is unrelated to $[W]_L$ with $W_L = 0$. ② is due to $X_0 = 0$.

Further, for $t \in [T]$, denote the angle between $X_{t-1}^0$ and $\mathbf{M}^\top(\mathbf{M}X_{t-1}^0 - Y)$ as $\theta_{t-1}$, we have $\sin(\theta_{t-1}) \in (0,1)$, setting $[W]_L = e[W]_L$, we calculate $\sigma_{\min}(\tilde{G}_{L-1,T}^0)$ by:

$$
\begin{aligned}
\sigma_{\min}(\tilde{G}_{L-1,T}^0) =& \sigma_{\min}\big(\operatorname{ReLU}(\operatorname{ReLU}([X_{T-1}^0, \mathbf{M}^\top(\mathbf{M}X_{T-1}^0 - Y)]eW_1^{0\top})\cdots eW_{L-1}^{0}{}^\top)\big), \\
\geq& \sigma_{\min}\big([X_{T-1}^0 | \mathbf{M}^\top(\mathbf{M}X_{T-1}^0 - Y)]\big)\textstyle\prod_{\ell=1}^{L-1}\sigma_{\min}(eW_\ell^0), \\
\geq& \tfrac{\|X_{T-1}^0\|_2\|\mathbf{M}^\top(\mathbf{M}X_{T-1}^0-Y)\|_2\sin(\theta_{T-1})}{\|X_{T-1}^0\|_2 + \|\mathbf{M}^\top(\mathbf{M}X_{T-1}^0-Y)\|_2}\textstyle\prod_{\ell=1}^{L-1}\sigma_{\min}(eW_\ell^0), \\
=& \tfrac{\sin(\theta_{T-1})}{\frac{1}{\|X_{T-1}^0\|_2} + \frac{1}{\|\mathbf{M}^\top(\mathbf{M}X_{T-1}^0-Y)\|_2}}\textstyle\prod_{\ell=1}^{L-1}\sigma_{\min}(eW_\ell^0), \\
\geq& \sin(\theta_{T-1})\textstyle\prod_{\ell=1}^{L-1}\sigma_{\min}(W_\ell^0)\Theta_L\|X_{T-1}^0\|_2.
\end{aligned}
\tag{56}
$$

Based on the definition of $X_{T-1}^0$ in Equation (55), we calculate following bound:

$$\sigma_{\min}(\tilde{G}_{L-1,T}^0) \geq \frac{\sin(\theta_{T-1})}{\beta} \|\sum_{t=1}^{T-1}(\mathbf{I} - \frac{1}{\beta}\mathbf{M}^\top\mathbf{M})^{T-t}\mathbf{M}^\top Y\|_2 \prod_{\ell=1}^{L-1}\sigma_{\min}(eW_\ell^0),$$
$$\geq \frac{\sin(\theta_{T-1})}{\beta} \underbrace{\sigma_{\min}(\sum_{t=1}^{T-1}(\mathbf{I} - \frac{1}{\beta}\mathbf{M}^\top\mathbf{M})^{T-t})}_{\delta_7} \|\mathbf{M}^\top Y\|_2 e^{L-1}\prod_{\ell=1}^{L-1}\sigma_{\min}(W_\ell^0),$$

(57)

where $X_{T-1}^0$ is a constant related to problem definition.

Substituting Equation (55), we calculate a tighter lower bound of $\|X_{T-1}^0\|_2$ by:

$$\|X_{T-1}^0\|_2 = \left\|\frac{1}{\beta}\sum_{t=1}^{T-1}\prod_{s=T-1}^{t+1}(\mathbf{I} - \frac{1}{\beta}\mathcal{D}(P_s^0)\mathbf{M}^\top\mathbf{M})\mathcal{D}(P_t^0)\mathbf{M}^\top Y\right\|_2,$$
$$\geq \frac{1}{\beta}\|\mathbf{M}^\top Y\|_2\sigma_{\min}\left(\sum_{t=1}^{T-1}\prod_{s=T-1}^{t+1}(\mathbf{I} - \frac{1}{\beta}\mathcal{D}(P_s^0)\mathbf{M}^\top\mathbf{M})\mathcal{D}(P_t^0)\right),$$
$$\overset{①}{\geq} \frac{1}{\beta}\|\mathbf{M}^\top Y\|_2\sum_{t=1}^{T-1}\sigma_{\min}\left(\prod_{s=T-1}^{t+1}(\mathbf{I} - \frac{1}{\beta}\mathcal{D}(P_s^0)\mathbf{M}^\top\mathbf{M})\right)\sigma_{\min}(\mathcal{D}(P_t^0)),$$
$$\geq \frac{1}{\beta}\|\mathbf{M}^\top Y\|_2\sum_{t=1}^{T-1}\left(\prod_{s=T-1}^{t+1}\sigma_{\min}(\mathbf{I} - \frac{1}{\beta}\mathcal{D}(P_s^0)\mathbf{M}^\top\mathbf{M})\right)\sigma_{\min}(\mathcal{D}(P_t^0)),$$

(58)

where ① is due to all matrices in the summation are positive semi-definite by definition.

We calculate lower bound for $\sigma_{\min}\left(\mathbf{I} - \frac{1}{\beta}\mathcal{D}(P_s^0)\mathbf{M}^\top\mathbf{M}\right)$ by:

$$\sigma_{\min}\left(\mathbf{I} - \frac{1}{\beta}\mathcal{D}(P_s^0)\mathbf{M}^\top\mathbf{M}\right) \geq 1 - \frac{1}{\beta}\sigma_{\max}\left(\mathcal{D}(2\sigma(eW_L^0\tilde{G}_{L-1,s}^0))\mathbf{M}^\top\mathbf{M}\right)$$
$$\geq 1 - 2\underbrace{\sigma(\delta_3\Theta_L)(1 - \sigma(\delta_3\Theta_L))}_{\delta_4}\sigma_{\max}(eW_L^0\tilde{G}_{L-1,s}^0),$$

(59)

It is easy to verify that the above equation equal to 1 when $e \to +\infty$ and it decreases with $e$. Also, a large $e$ ensures the RHS of above inequality to be positive.

Similarly, we calculate lower bound for $\sigma_{\min}(P_t^0)$ by:

$$\sigma_{\min}(\mathcal{D}(P_t^0)) \overset{①}{=} \min\left(2\sigma(eW_L^0\tilde{G}_{L-1,t}^0)\right),$$
$$\overset{②}{=} \min\left(\frac{\partial 2\sigma}{\partial v_4}(eW_L^0\tilde{G}_{L-1,t}^0)\right),$$
$$\overset{③}{\geq} 2\delta_4\sigma_{\min}\left(eW_L^0\tilde{G}_{L-1,t}^0\right),$$
$$\overset{④}{\geq} 2\delta_4 e\|W_L^0\|_2\sigma_{\min}(\tilde{G}_{L-1,t}^0),$$
$$\geq 2\Theta_L\delta_4\prod_{\ell=1}^{L}\|W_\ell^0\|_2\sigma_{\min}\left([X_{t-1}^0|\mathbf{M}^\top(\mathbf{M}X_{t-1}^0 - Y)]\right),$$
$$\overset{⑤}{\geq} 2\Theta_L\delta_4\prod_{\ell=1}^{L}\|W_\ell^0\|_2\sin(\theta_{T-1})\|X_{t-1}^0\|_2$$

(60)

where ① means we apply the expansion here. ② is due to Mean Value Theorem and $v_4$ denotes a inner point between 0 and $eW_L^0\tilde{G}_{L-1,T}^0$. ③ is due to Lemma A.12 and Lemma A.14. ④ is due to $W_L^0$ is a vector in definition. ⑤ is similar to the workflow in Equation (56).

Substituting Equation (59) and Equation (60) back into Equation (58) yields:

$$\|X_{t-1}^0\|_2 \geq \frac{1}{\beta}\|\mathbf{M}^\top Y\|_2\sum_{s=1}^{t-1}2\Theta_L\delta_4\prod_{\ell=1}^{L}\|W_\ell^0\|_2\sigma_{\min}\left([X_{s-1}^0|\mathbf{M}^\top(\mathbf{M}X_{s-1}^0 - Y)]\right),$$
$$\geq \frac{2}{\beta}\|\mathbf{M}^\top Y\|_2\Theta_L\sum_{s=1}^{t-1}\delta_4\prod_{\ell=1}^{L}\|W_\ell^0\|_2\sin(\theta_{s-1})\|X_{s-1}^0\|_2,$$

Similarly, we can get the following lower bound of $\|X_{t-1}^0\|_2$:

$$\|X_{t-1}^0\|_2 \geq \frac{2}{\beta}\|\mathbf{M}^\top Y\|_2\Theta_L\sum_{s=1}^{t-1}\delta_4\prod_{\ell=1}^{L}\|W_\ell^0\|_2\sin(\theta_{t-1})\|X_{s-1}^0\|_2,$$

Based on the above results, we calculate the $\boldsymbol{\Omega}$ of $\|X_{T-1}^0\|_2$ as in terms of $T$ and $\Theta_L$ as:

$$\|X_{T-1}^0\|_2 \geq \mathbf{\Omega}(\Theta_L \underbrace{\sum_{t=1}^{T-1} \Theta_L \sum_{s=1}^{t-1} \Theta_L \sum_{j=1}^{s-1} \cdots \sum_{j=1}^{2}}_{\text{T-2 terms}}) = \mathbf{\Omega}(\Theta_L^{T-2}).$$

Substituting back into Equation (56) yields:

$$\sigma_{\min}(\tilde{G}_{L-1,T}^0) = \mathbf{\Omega}(e^{L-1}e^{(T-2)(L-1)}) = \mathbf{\Omega}(e^{(T-1)(L-1)}). \tag{61}$$

## B.2 Proof of Lemma 5.1

*Proof.* Making up the lower bounding relationship with Equation (57) and Equation (62) yields:

$$e^{L-1}\|\mathbf{M}^\top Y\|_2 \delta_7 \prod_{\ell=1}^{L-1} \sigma_{\min}(W_\ell^0) \geq 8(1+\beta)(\|X_0\|_2 + \tfrac{2T-2}{\beta}\|\mathbf{M}^\top Y\|_2),$$
$$= \tfrac{8(1+\beta)}{\beta}(2T-2)\|\mathbf{M}^\top Y\|_2,$$

which yields:

$$e \geq \sqrt[L-1]{\tfrac{8(1+\beta)}{\beta}\delta_7^{-1}\sigma_{\min}(W_\ell^0)^{-1}(2T-2)}.$$

$\square$

## B.3 Proof of Lemma 5.4

We apply a similar workflow to prove Lemma 5.4.

*Proof.* With $X_0 = 0$, we find the upper bound of the RHS of Equation (13d) by substituting the quantity $\delta_5$:

$$\frac{(1+\beta)\beta^2\sqrt{\beta}}{2\beta_0^2}\delta_5\big(\sqrt{\beta}\|X_0\|_2 + (2T+1)\|Y\|_2\big)\zeta_2 S_{\Lambda,T}\Theta_{L-1}\Big(\sum_{\ell=1}^{L}\tfrac{\Theta_L}{\lambda_\ell^2}\Big)$$

$$\overset{\textcircled{1}}{=} \frac{(1+\beta)\beta^2\sqrt{\beta}}{2\beta_0^2}\delta_5\big(\sqrt{\beta}\|X_0\|_2 + (2T+1)\|Y\|_2\big)\zeta_2$$
$$\Big(\tfrac{4(\beta+1)}{3\beta^2}\|\mathbf{M}^\top Y\|_2^2 T^3 - \tfrac{1}{\beta^2}\|\mathbf{M}^\top Y\|_2^2 T^2 - \tfrac{\beta+1}{3\beta^2}\|\mathbf{M}^\top Y\|_2^2 T\Big)\Theta_{L-1}\Big(\sum_{\ell=1}^{L}\tfrac{\Theta_L}{\lambda_\ell^2}\Big),$$

$$\overset{\textcircled{2}}{=} \frac{(1+\beta)\beta\sqrt{\beta}}{2\beta_0^2}\delta_5\|Y\|_2\|\mathbf{M}^\top Y\|_2(2T-2)(2T+1) \tag{62}$$
$$\Big(\tfrac{4(\beta+1)}{3\beta^2}\|\mathbf{M}^\top Y\|_2^2 T^3 - \tfrac{1}{\beta^2}\|\mathbf{M}^\top Y\|_2^2 T^2 - \tfrac{\beta+1}{3\beta^2}\|\mathbf{M}^\top Y\|_2^2 T\Big)\Theta_{L-1}\Big(\sum_{\ell=1}^{L}\tfrac{\Theta_L}{\lambda_\ell^2}\Big),$$

$$\overset{\textcircled{3}}{\leq} \frac{(1+\beta)\sqrt{\beta}}{6\beta_0^2\beta}\delta_5\|Y\|_2\|\mathbf{M}^\top Y\|_2^3$$
$$\Big(16(\beta+1)T^5 - (8\beta+20)T^4 - 6(2\beta+1)T^3 + 2(\beta+4)T^2 + 2(\beta+1)T\Big)L\Theta_{L-1}^2,$$

where ① is due to Equation (54) and definition of quantity $\delta_1^{T-1}$ in Theorem 4.3. ② is due to $X_0 = 0$. ③ is due to $\bar{\lambda}_L = 1$ and $\bar{\lambda}_\ell > 1, \ell \in [L-1]$.

Making up the lower bounding relationship with Equation (57) and Equation (62) yields:

$$\Big(e^{L-1}\|\mathbf{M}^\top Y\|_2\delta_7\prod_{\ell=1}^{L-1}\sigma_{\min}(W_\ell^0)\Big)^3$$
$$\geq e^{2L-2}\frac{(1+\beta)\sqrt{\beta}}{6\beta_0^2\beta}\delta_5\|Y\|_2\|\mathbf{M}^\top Y\|_2^3 L\prod_{\ell=1}^{L-1}(\|W_\ell^0\|_2 + 1)^2 \tag{63}$$
$$\Big(16(\beta+1)T^5 - (8\beta+20)T^4 - 6(2\beta+1)T^3 + 2(\beta+4)T^2 + 2(\beta+1)T\Big),$$

which yields:

$$e \geq \sqrt[L-1]{C_{2,\delta_5}\Big(16(\beta+1)T^5 - (8\beta+20)T^4 - 6(2\beta+1)T^3 + 2(\beta+4)T^2 + 2(\beta+1)T\Big)}.$$

where $C_{2,\delta_5}$ denotes the $\frac{(1+\beta)\sqrt{\beta}}{6\beta_0^2\beta\delta_7^3}\delta_5\|Y\|_2 L\prod_{\ell=1}^{L-1}(\|W_\ell^0\|_2 + 1)^2\prod_{\ell=1}^{L-1}\sigma_{\min}(W_\ell^0)^{-3}$ term.

Similarly, the finite RHS of above inequality ensures $\delta_5 \ll \infty$.

$\square$

## B.4 Proof of Lemma 5.2

*Proof.* Using quantities from Equation (12), with $X_0 = 0$, we find the upper bound of the RHS of Equation (13b) by substituting the quantity $\delta_5$:

$$\frac{\beta^3}{4\beta_0^2}\delta_5\Big(-\tfrac{1}{2}\Theta_{L-1}^2\Lambda_T\big(\textstyle\sum_{t=1}^{T-1}\Lambda_t\big)+\Theta_L^2 S_{\bar{\lambda},L}(\Lambda_T+\delta_2)S_{\Lambda,T}\Big)$$

$$\overset{\text{①}}{=}\frac{\beta^3}{4\beta_0^2}\delta_5\Big(-\tfrac{1}{2}\Theta_{L-1}^2\Big(\tfrac{4(\beta+1)}{\beta^2}\|\mathbf{M}^\top Y\|_2^2 T^2-\tfrac{4\beta+6}{\beta^2}\|\mathbf{M}^\top Y\|_2^2 T+\tfrac{\beta+2}{\beta^2}\|\mathbf{M}^\top Y\|_2^2\Big)$$

$$\Big(\tfrac{4(\beta+1)}{3\beta^2}\|\mathbf{M}^\top Y\|_2^2(T-1)^3-\tfrac{1}{\beta^2}\|\mathbf{M}^\top Y\|_2^2(T-1)^2-\tfrac{\beta+1}{3\beta^2}\|\mathbf{M}^\top Y\|_2^2(T-1)\Big)$$

$$+\Theta_L^2 S_{\bar{\lambda},L}\Big(\Big(\tfrac{4(\beta+1)}{\beta^2}\|\mathbf{M}^\top Y\|_2^2 T^2-\tfrac{4\beta+6}{\beta^2}\|\mathbf{M}^\top Y\|_2^2 T+\tfrac{\beta+2}{\beta^2}\|\mathbf{M}^\top Y\|_2^2\Big)$$

$$+\textstyle\sum_{s=1}^{T-1}\Big(\prod_{j=s+1}^T\big(1+\tfrac{1+\beta}{2\beta}(2j-1)\Theta_L\|\mathbf{M}^\top Y\|_2\big)\Big)$$

$$\Big(\tfrac{4(\beta+1)}{\beta^2}\|\mathbf{M}^\top Y\|_2^2 s^2-\tfrac{4\beta+6}{\beta^2}\|\mathbf{M}^\top Y\|_2^2 s+\tfrac{\beta+2}{\beta^2}\|\mathbf{M}^\top Y\|_2^2\big)\Big)$$

$$\Big(\tfrac{4(\beta+1)}{3\beta^2}\|\mathbf{M}^\top Y\|_2^2 T^3-\tfrac{1}{\beta^2}\|\mathbf{M}^\top Y\|_2^2 T^2-\tfrac{\beta+1}{3\beta^2}\|\mathbf{M}^\top Y\|_2^2 T\Big)\Big),$$

$$\leq\mathcal{O}\big(e^{2L-2}T^5+e^{2L-4}T^5+e^{2L-4}T^6\textstyle\sum_{s=1}^{T-1}s^2\prod_{j=s+1}^T je^{L-1}\big),$$

$$=\mathcal{O}\big(e^{TL-T+2L-4}T^{3T+6}\big).$$

(64)

where ① is due to Equation (54) and definition of quantity $\delta_1^{T-1}$ in Theorem 4.3. ② is due to $X_0 = 0$. ③ is due to $\bar{\lambda}_L = 1$ and $\bar{\lambda}_\ell > 1, \ell \in [L-1]$.

Making up the lower bounding relationship with Equation (61) and Equation (62) yields:

$$(\mathbf{\Omega}(e^{(T-1)(L-1)}))^2 \geq \mathcal{O}(e^{TL-T+2L-4}T^{3T+6}),$$

which yields:

$$e = \mathbf{\Omega}\big(T^{\frac{3T+6}{TL-T-4L+6}}\big).$$

$\square$

## B.5 Proof of Lemma 5.3

*Proof.* Using quantities from Equation (12), with $X_0 = 0$, we find the upper bound of the RHS of Equation (13c) by substituting the quantity $\delta_5$:

$$\max_{\ell\in[L]}\frac{\Theta_L}{C_\ell\bar{\lambda}_\ell}\frac{\beta^2\sqrt{\beta}}{8\beta_0^2}$$

$$\underbrace{\sigma\big((2T-1+\tfrac{2T-2}{\beta})\|\mathbf{M}^\top Y\|_2\Theta_L\big)^{-2}\big(1-\sigma((2T-1+\tfrac{2T-2}{\beta})\|\mathbf{M}^\top Y\|_2\Theta_L)\big)^{-2}}_{\delta_5}$$

$$S_{\Lambda,T}(2T+1)\|Y\|_2,$$

$$\overset{\text{①}}{\leq}\frac{\beta^2\sqrt{\beta}}{8\beta_0^2}\delta_5 S_{\Lambda,T}(2T+1)\|Y\|_2\prod_{\ell=1}^{L-1}(\|W_\ell^0\|_2+1),$$

(65)

$$\overset{\text{②}}{=}\frac{\beta^2\sqrt{\beta}}{8\beta_0^2}\delta_5\big(\tfrac{4(\beta+1)}{3\beta^2}\|\mathbf{M}^\top Y\|_2^2 T^3-\tfrac{1}{\beta^2}\|\mathbf{M}^\top Y\|_2^2 T^2-\tfrac{\beta+1}{3\beta^2}\|\mathbf{M}^\top Y\|_2^2 T\big)$$

$$(2T+1)\|Y\|_2\prod_{\ell=1}^{L-1}(\|W_\ell^0\|_2+1),$$

$$=\frac{\beta^2\sqrt{\beta}}{8\beta_0^2}\delta_5\|Y\|_2\|\mathbf{M}^\top Y\|_2^2\big(\tfrac{8(\beta+1)}{3\beta^2}T^4+\big(\tfrac{4(\beta+1)}{3\beta^2}-\tfrac{2}{\beta^2}\big)T^3-\big(\tfrac{1}{\beta^2}+2\tfrac{\beta+1}{3\beta^2}\big)T^2-\tfrac{\beta+1}{3\beta^2}T\big)$$

$$\prod_{\ell=1}^{L-1}(\|W_\ell^0\|_2+1),$$

where ① is due to $\bar{\lambda}_\ell > 1, \ell \in [L-1]$ and $\bar{\lambda}_L = 1$. ② is due to Equation (54).

We analyze the two sides of the above inequality when $[W]_L = e[W]_L$ to demonstrate a sufficient lower bound of $e$ to ensure Equation (65) holds.

If $[W]_L = e[W]_L$, since $e \geq 1$, Equation (65) is upper-bounded by:

$$
\begin{aligned}
&\frac{\beta^2\sqrt{\beta}}{8\beta_0^2}\delta_5\|Y\|_2\|\mathbf{M}^\top Y\|_2^2\Big(\frac{8(\beta+1)}{3\beta^2}T^4 + \big(\frac{4(\beta+1)}{3\beta^2} - \frac{2}{\beta^2}\big)T^3 - \big(\frac{1}{\beta^2} + 2\frac{\beta+1}{3\beta^2}\big)T^2 - \frac{\beta+1}{3\beta^2}T\Big) \\
&\prod_{\ell=1}^{L-1}(e\|W_\ell^0\|_2 + e) \\
=& e^{L-1}\frac{\beta^2\sqrt{\beta}}{8\beta_0^2}\delta_5\|Y\|_2\|\mathbf{M}^\top Y\|_2^2 \\
&\Big(\frac{8(\beta+1)}{3\beta^2}T^4 + \big(\frac{4(\beta+1)}{3\beta^2} - \frac{2}{\beta^2}\big)T^3 - \big(\frac{1}{\beta^2} + 2\frac{\beta+1}{3\beta^2}\big)T^2 - \frac{\beta+1}{3\beta^2}T\Big)\prod_{\ell=1}^{L-1}(\|W_\ell^0\|_2 + 1).
\end{aligned}
\tag{66}
$$

If RHS (lower bound) of Equation (57) greater than the RHS (upper bound) of above result, lower bound condition for minimal singular value in Equation (65) sufficiently holds, which yields:

$$
\begin{aligned}
&\Big(e^{L-1}\|\mathbf{M}^\top Y\|_2\delta_7\prod_{\ell=1}^{L-1}\sigma_{\min}(W_\ell^0)\Big)^2 \\
\geq& e^{L-1}\frac{\beta^2\sqrt{\beta}}{8\beta_0^2\delta_6}\delta_5\|Y\|_2\|\mathbf{M}^\top Y\|_2^2\Big(\frac{8(\beta+1)}{3\beta^2}T^4 + \big(\frac{4(\beta+1)}{3\beta^2} - \frac{2}{\beta^2}\big)T^3 - \big(\frac{1}{\beta^2} + 2\frac{\beta+1}{3\beta^2}\big)T^2 - \frac{\beta+1}{3\beta^2}T\Big) \\
&\prod_{\ell=1}^{L-1}(\|W_\ell^0\|_2 + 1),
\end{aligned}
$$

which yields:

$$
e \geq \sqrt[L-1]{C_{1,\delta_5}\Big(\frac{8(\beta+1)}{3}T^4 + \big(\frac{4(\beta+1)}{3} - 2\big)T^3 - \big(1 + 2\frac{\beta+1}{3}\big)T^2 - \frac{\beta+1}{3}T\Big)},
$$

where $C_{1,\delta_5}$ denotes the $\frac{\sqrt{\beta}}{8\beta_0^2\delta_6\delta_7^2}\delta_5\|Y\|_2\prod_{\ell=1}^{L-1}(\|W_\ell^0\|_2 + 1)\prod_{\ell=1}^{L-1}\sigma_{\min}(W_\ell^0)^{-2}$ term, which is a "constant" w.r.t. $\delta_5$.

In the end, it is trivial to evaluate that the RHS of above $\delta_5$ is finite with such $e$. $\qquad\square$

## C   Additional Experimental Results

In this section, we present detailed experimental settings and corresponding results. We define problems at three distinct scales, as described in Appendix C.1. The smaller scale is utilized for ablation studies (Section 6.2), whereas the larger scales are adopted for training experiments (Section 6.1 and Appendix C.2) and inference experiments (Appendix C.4).

### C.1   Configurations for Different Experiments

Details of the three experimental configurations are presented in Table 1. **Scale 1** involves a DNN trained with input $X \in \mathbb{R}^{32\times32}$ and output $Y \in \mathbb{R}^{32\times25}$, featuring an $(L-1)$-th layer dimension of 1024. **Scale 2** utilizes input $X \in \mathbb{R}^{10\times512}$ and output $Y \in \mathbb{R}^{10\times400}$, with the $(L-1)$-th layer dimension established at 5120. **Scale 3** employs input $X \in \mathbb{R}^{2048\times512}$ and output $Y \in \mathbb{R}^{2048\times400}$. This configuration is designed as an under-parameterized system, with an $(L-1)$-th layer dimension of 5120, specifically to evaluate the robustness of our proposed L2O framework. The third model, although targeting the optimization problem with the same dimension, has a different number of training samples $N$. We design the scale to align with the training configurations of the baseline model LISTA-CPSS [7]. Moreover, due to the GPU memory limitation, we set a thin NN, whose convergence is not guaranteed by our proposed theorem. The related experimental result is used to further demonstrate our proposed framework in Section 3.

Table 1: Configurations with Different Scales

| Index | $d$ | $b$ | Dimension of $L-1$ **Layer's Output** | **Training Samples** |
|---|---|---|---|---|
| 1 | 32 | 25 | 1024 | 32 |
| 2 | 512 | 400 | 5120 | 10 |
| 3 | 512 | 400 | 20 | 2048 |

## C.2 Additional Training Performance Comparisons verses L2O Baselines

For these experiments, the **Scale 3** configuration is utilized. Both baseline state-of-the-art (SOTA) methods and our proposed L2O framework are trained for 2000 epochs using a learning rate of 0.001. However, the inherent model construction and training scheme of a prominent SOTA method, LISTA-CPSS [7], diverges considerably from the requirements of our problem. Direct application of its original settings to our scenario results in over-fitting and poor training convergence, indicating a lack of robustness for this specific application. The following discussion elaborates on these incompatibilities and the modifications undertaken.

The original LISTA-CPSS framework possesses two key characteristics pertinent to this discussion. First, regarding its model construction, LISTA-CPSS addresses inverse problems by formulating a learnable Least Absolute Shrinkage and Selection Operator (LASSO) problem, wherein it learns a scalar coefficient for the $L_1$ regularization term [7]. However, our objective in Equation (1) is quadratic. Second, its training protocol is supervised, utilizing an $L_2$ loss against pre-generated optimal solutions, and employs a layer-wise training scheme. In this scheme, one layer is progressively added to the set of trainable parameters per training iteration, and these parameters are updated using four back-propagation (BP) steps [7]. To adapt LISTA-CPSS for our purposes, we modify both its model architecture and original training scheme to enable unsupervised optimization of our loss function (defined in Equation (2)) and to better align with our established training configuration.

First, to demonstrate the challenges of applying LISTA-CPSS's original training paradigm to unsupervised quadratic objectives, we evaluate a minimally adapted version. This version is trained unsupervisedly by defining the loss as the objective function value from the final optimization step. Given our quadratic loss in Equation (2), any model components in LISTA-CPSS specifically designed for non-quadratic terms are not directly applicable. Moreover, a critical aspect of the publicly available LISTA-CPSS implementation is its initialization of the neural network (NN) with a fixed matrix $\mathbf{M}$. This initialization inherently restricts the trained model's utility to problems featuring this identical, predetermined $\mathbf{M}$.

We train this minimally adapted LISTA-CPSS variant for 50 epochs (corresponding to 20000 BPs due to its layer-wise updates) using the Adam optimizer[3] on a dataset of 2048 randomly generated samples. The loss function defined in Equation (2) is evaluated at an optimization step of $T = 100$. The experimental results, depicted in Figure 6, reveal that this configuration leads to severe over-fitting on the training samples. Specifically, Figure 6a illustrates the convergence of the objective function (at $T = 100$) as a function of the training iteration $k$. Concurrently, Figure 6b displays the mean objective value across 100 optimization steps during inference. These results indicate that while LISTA-CPSS achieves rapid convergence on the training data (which used a fixed $\mathbf{M}$), its performance degrades catastrophically (i.e., fails to generalize) when evaluated with a different matrix, $\mathbf{M}'$, during inference.

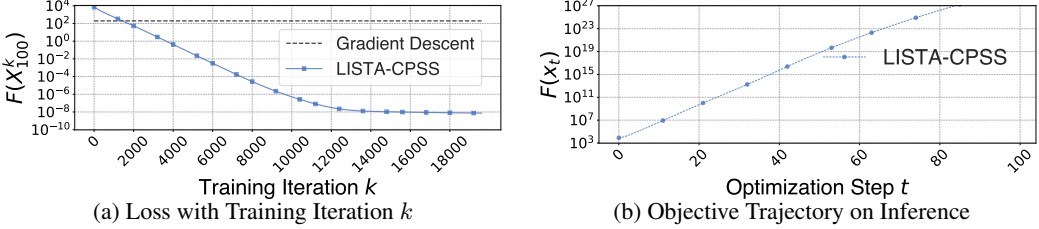

(a) Loss with Training Iteration $k$      (b) Objective Trajectory on Inference

Figure 6: Training Loss and Inference Trajectory of LISTA-CPSS [7] with Fixed $\mathbf{M}$

Informed by the above observation, a more robust approach is achieved through the random initialization of LISTA-CPSS. Specifically, weights are sampled from a standard Gaussian distribution and subsequently scaled by a factor of $\frac{1}{d \cdot b}$ to mitigate potential numerical overflow in cumulative products. The LISTA-CPSS model is then trained using this initialization strategy.

For our proposed L2O framework, the expansion coefficient $e$ is set to 100. As detailed in **Scale 3** in Table 1, we implement an under-parameterized system wherein the dimension of the $(L - 1)$-th layer

---

[3]Our preliminary experiments indicates that SGD fails to converge with LISTA-CPSS's original layer-wise training scheme.

is configured to 20. This implementation intentionally deviates from the theoretical requirements stipulated by our proposed theorems, which necessitate that the dimension of the $(L-1)$-th layer must be larger than the input dimension. This particular experiment is conducted to demonstrate the robustness of the proposed L2O framework, especially under such conditions that depart from our established theoretical framework.

The training losses of LISTA-CPSS and our proposed L2O framework are depicted in Figure 7, with the performance of non-learnable gradient descent (indicated by a horizontal line in the figure) serving as a baseline. Under scenarios with varied $\mathbf{M}$ configurations, LISTA-CPSS exhibits markedly slower convergence compared to both our proposed L2O framework and the gradient descent baseline. Moreover, the fast convergence observed for our L2O framework underscores the robustness and efficacy of its proposed initialization strategy, particularly when applied to under-parameterized models.

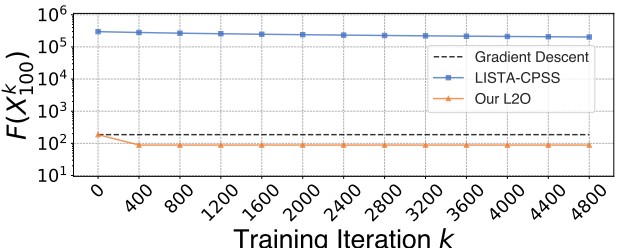

Figure 7: Training Losses with Varied $\mathbf{M}$

## C.3 Real-World Training Performance Comparisons

To empirically validate our proposed theorem, we perform an additional experiment comparing the training convergence of our L2O construction against standard Gradient Descent (GD). Utilizing a compact Convolutional Neural Network (CNN) on the MNIST dataset, our method achieved significantly faster convergence, thereby corroborating our theoretical findings.

We employ the **Scale 3** configuration (an under-parameterized setting from Table 1). The CNN architecture (Table 2) comprises two convolutional layers, two max-pooling layers, ReLU activation functions, and a final linear layer. The optimization objective is the total cross-entropy loss over 200 randomly selected MNIST samples. The learning rates for training our L2O model and the CNN were set to $10^{-6}$ and $10^{-2}$, respectively.

Table 2: Architecture of a Small CNN Model with MNIST Dataset

| Layer | Input Channel | Output Channel | Kernel Size | Input Size | Output Size |
|---|---|---|---|---|---|
| Convolution | 1 | 2 | 3 | $28 \times 28$ | $28 \times 28$ |
| Max Pooling | 2 | 2 | 2 | $28 \times 28$ | $14 \times 14$ |
| ReLU | 2 | 2 | N/A | $14 \times 14$ | $14 \times 14$ |
| Convolution | 2 | 3 | 3 | $14 \times 14$ | $14 \times 14$ |
| Max Pooling | 3 | 3 | 2 | $14 \times 14$ | $7 \times 7$ |
| ReLU | 3 | 3 | N/A | $7 \times 7$ | $7 \times 7$ |
| Linear | 147 | 10 | N/A | 1 | 1 |

To validate our framework, we conducted a comparative analysis of the CNN training loss on the MNIST dataset, contrasting our proposed L2O method with Gradient Descent (GD). The results are depicted in Figure 8a, which plots the training loss over 100 iterations. We evaluate two versions of our L2O optimizer, pre-trained for 100 and 200 epochs, respectively. In both scenarios, our L2O framework yields a substantially lower loss than the GD baseline, which corroborates the effectiveness of our approach for training DNN models.

Additionally, Figure 8b provides a quantitative comparison of the iteration cost for both methods. The proposed L2O framework converges to a more optimal (lower) loss value than GD in substantially fewer iterations, confirming its superior efficiency in training the CNN model.

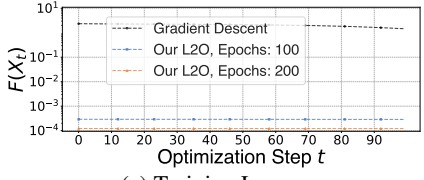

(a) Training Losses

| Method | Loss Value with Iterations |
|---|---|
| GD | 10,000 Iterations: 2.92e-04 |
| Our L2O, Epoch: 100 | 100 Iterations: 2.91e-04 |
| Our L2O, Epoch: 200 | 100 Iterations: 1.22e-04 |

(b) Final Loss Values of CNN on MNIST Dataset

Figure 8: Performance of Training CNN on MNIST Dataset

## C.4 Inference Experiment

Beyond analyzing training outcomes, we extend our evaluation to the robustness of the proposed L2O framework by assessing its performance in inference-stage optimization. This involves comparing the convergence characteristics of L2O against the Adam optimizer [10] and standard gradient descent (GD). It should be noted that while our theorems provide convergence guarantees for the training phase, such guarantees do not explicitly extend to this inference optimization context. For this empirical investigation, both our L2O framework and the Adam optimizer are executed across a range of hyperparameter settings for 3000 iterations (longer than 100 iterations in training), and their respective objective function trajectories are plotted as a function of the iteration count.

Adam utilizes momentum to accelerate gradient descent. In addition to the learning rate $\eta$, Adam employs two crucial hyperparameters, $\beta_1$ and $\beta_2$, which control the exponential moving averages of past gradients and their squared magnitudes, respectively. For the Adam optimizer in our experiments, we set the learning rate $\eta = \frac{1}{\beta}$ ($\beta$-smoothness of objective) and explored hyper-parameters $\beta_1 \in \{0.1, 0.3, \dots, 0.9\}$ and $\beta_2 \in \{0.95, 0.955, \dots, 1.0\}$.

Regarding our proposed L2O framework and consistent with the initialization strategy detailed in Section 5, we selected a large expansion coefficient $e = 100$ to enhance training stability. The L2O model is then trained with learning rates $\eta$ chosen from the set $\{10^{-3}, 10^{-4}, \dots, 10^{-7}\}$.

As illustrated in Figure 9, we present the objective trajectory over 3000 optimization steps, where each point is a mean value of 30 randomly generated problems' objectives. While the objective function initially exhibits rapid decay, the Adam optimizer fails to maintain this convergence, ultimately settling at sub-optimal values and not converging on average. In contrast, our proposed framework demonstrates superior performance compared to the Gradient Descent (GD) algorithm and exhibits robustness across various learning rates.

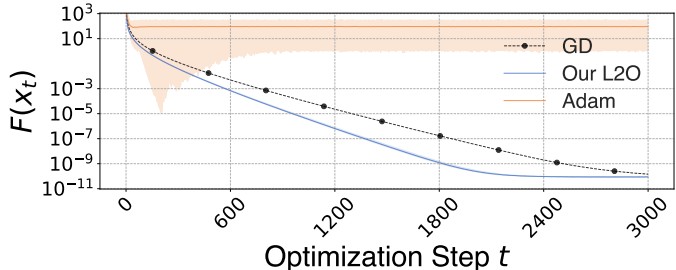

Figure 9: Inference Trajectory of Our Proposed L2O

## C.5 Corollary in Ablation Studies

**Corollary C.1** (LR's upper bound w.r.t. $e$)**.**

$$\eta = \mathcal{O}(e^{3-L}T^{-6}) \cap \mathcal{O}(e^{1-L}T^{-4}) \cap \mathcal{O}(e^{\frac{4}{3}(1-L)}T^{-\frac{10}{3}}) \cap \mathcal{O}(e^{-TL-2L+T+4}T^{-3T-6}) \cap \mathcal{O}(T^{-2}).$$

*Proof.* From Equation (14a), we calculate:

$$\eta$$

$$< \tfrac{8}{\beta}(\delta_2 + \Lambda_T)\Big(\delta_2 + S_{\Lambda,T}\Big)^{-1} S_{\Lambda,T}^{-2}\Theta_L^{-1}S_{\bar{\lambda},L}^{-1},$$

$$< \Bigg(\sum_{s=1}^{T-1}\Big(\prod_{j=s+1}^{T}\big(1 + \tfrac{1+\beta}{2\beta}(2j-1)\Theta_L\|\mathbf{M}^\top Y\|_2\big)\Big)$$

$$\Big(\tfrac{4(\beta+1)}{\beta^2}\|\mathbf{M}^\top Y\|_2^2 s^2 - \tfrac{4\beta+6}{\beta^2}\|\mathbf{M}^\top Y\|_2^2 s + \tfrac{\beta+2}{\beta^2}\|\mathbf{M}^\top Y\|_2^2\Big)$$

$$+ \Big(\tfrac{4(\beta+1)}{\beta^2}\|\mathbf{M}^\top Y\|_2^2 T^2 - \tfrac{4\beta+6}{\beta^2}\|\mathbf{M}^\top Y\|_2^2 T + \tfrac{\beta+2}{\beta^2}\|\mathbf{M}^\top Y\|_2^2\Big)\Bigg)$$

$$\Big(\sum_{s=1}^{T-1}\Big(\prod_{j=s+1}^{T}\big(1 + \tfrac{1+\beta}{2\beta}(2j-1)\Theta_L\|\mathbf{M}^\top Y\|_2\big)\Big)$$

$$\Big(\tfrac{4(\beta+1)}{\beta^2}\|\mathbf{M}^\top Y\|_2^2 s^2 - \tfrac{4\beta+6}{\beta^2}\|\mathbf{M}^\top Y\|_2^2 s + \tfrac{\beta+2}{\beta^2}\|\mathbf{M}^\top Y\|_2^2\Big)$$

$$+ \Big(\tfrac{4(\beta+1)}{3\beta^2}\|\mathbf{M}^\top Y\|_2^2 T^3 - \tfrac{1}{\beta^2}\|\mathbf{M}^\top Y\|_2^2 T^2 - \tfrac{\beta+1}{3\beta^2}\|\mathbf{M}^\top Y\|_2^2 T\Big)\Big)^{-1}$$

$$\Big(\tfrac{4(\beta+1)}{3\beta^2}\|\mathbf{M}^\top Y\|_2^2 T^3 - \tfrac{1}{\beta^2}\|\mathbf{M}^\top Y\|_2^2 T^2 - \tfrac{\beta+1}{3\beta^2}\|\mathbf{M}^\top Y\|_2^2 T\Big)^{-2}\Big(e^{L-1}\prod_{\ell=1}^{L-1}\bar{\lambda}_\ell\Big)^{-1}S_{\bar{\lambda},L}^{-1},$$

$$= \mathcal{O}(e^{3-L}T^{-6}).$$

From Equation (14b), due to the four lower bounds in Equation (13), we calculate following four upper bounds:

$$\eta$$

$$< \tfrac{1}{4}\tfrac{\beta^2}{\beta_0^2}\delta_4^{-2}\alpha_0^{-2},$$

$$\overset{66}{<}\tfrac{1}{4}\tfrac{\beta^2}{\beta_0^2}\delta_5\Bigg(e^{L-1}\tfrac{\beta^2\sqrt{\beta}}{8\beta_0^2}\delta_5\|Y\|_2\|\mathbf{M}^\top Y\|_2^2$$

$$\Big(\tfrac{8(\beta+1)}{3\beta^2}T^4 + \big(\tfrac{4(\beta+1)}{3\beta^2} - \tfrac{2}{\beta^2}\big)T^3 - \big(\tfrac{1}{\beta^2} + 2\tfrac{\beta+1}{3\beta^2}\big)T^2 - \tfrac{\beta+1}{3\beta^2}T\Big)\prod_{\ell=1}^{L-1}(\|W_\ell^0\|_2 + 1)\Bigg)^{-1},$$

$$= \mathcal{O}(e^{1-L}T^{-4}).$$

$$\eta$$

$$< \tfrac{1}{4}\tfrac{\beta^2}{\beta_0^2}\delta_4^{-2}\alpha_0^{-2},$$

$$\overset{\text{Equation (63)}}{<}\tfrac{1}{4}\tfrac{\beta^2}{\beta_0^2}\delta_5\Bigg(e^{2L-2}\tfrac{(1+\beta)\sqrt{\beta}}{6\beta_0^2\beta}\delta_5\|Y\|_2\|\mathbf{M}^\top Y\|_2^3$$

$$\Big(16(\beta+1)T^5 - (8\beta+20)T^4 - 6(2\beta+1)T^3 + 2(\beta+4)T^2 + 2(\beta+1)T\Big)$$

$$L\prod_{\ell=1}^{L-1}(\|W_\ell^0\|_2 + 1)^2\Bigg)^{-\tfrac{2}{3}}$$

$$= \mathcal{O}(e^{\tfrac{4}{3}(1-L)}T^{-\tfrac{10}{3}}).$$

$$\eta < \tfrac{1}{4}\tfrac{\beta^2}{\beta_0^2}\delta_4^{-2}\alpha_0^{-2} \overset{\text{Equation (64)}}{<} \tfrac{1}{4}\tfrac{\beta^2}{\beta_0^2}\delta_5\mathcal{O}((e^{TL-T+2L-4}T^{3T+6})^{-1}) = \mathcal{O}(e^{-TL-2L+T+4}T^{-3T-6}).$$

$$\eta < \tfrac{1}{4}\tfrac{\beta^2}{\beta_0^2}\delta_4^{-2}\alpha_0^{-2} \overset{\text{Equation (13a)}}{<} \tfrac{1}{4}\tfrac{\beta^2}{\beta_0^2}\delta_5\Big(8(1+\beta)(\|X_0\|_2 + \tfrac{2T-2}{\beta}\|\mathbf{M}^\top Y\|_2)\Big)^{-2} = \mathcal{O}(T^{-2}).$$

$\square$

## C.6 Additional Ablation Studies for Learning Rates

We present two additional ablation studies with $e$ of 25 and 100. Both use the configuration 1 in Table 1. The results are in Figure 10, which shows a deterministic relationship between LR and expansion coefficient. For $e = 25$ in Figure 10a, the $10^{-7}$ LR is too small and leads to worse optimality. The large LRs, i.e., $10^{-3}, 10^{-4}$, cause unstable convergence. Similarly, for $e = 100$ in Figure 10b, a proper LR is $10^{-4}$.

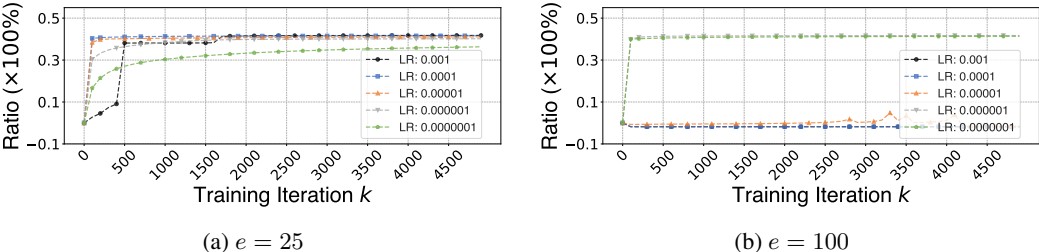

(a) $e = 25$          (b) $e = 100$

Figure 10: Additional Ablation Studies of Learning Rate with Different $e$.

## C.7 Additional Ablation Studies for Expansion Coefficient $e$ in Initialization

We present two additional ablation studies for $e$ with learning rates of $0.001$ and $0.00001$. Both use the configuration 1 in Table 1. The results are in Figure 11. For a large LR, a large $e$ may cause poor convergence due to Theorem 4.3. From Figure 11a, $e = 25$ is a proper setting for best convergence with $\eta = 0.001$. Similarly, for $\eta = 0.00001$, $e = 5$ is enough.

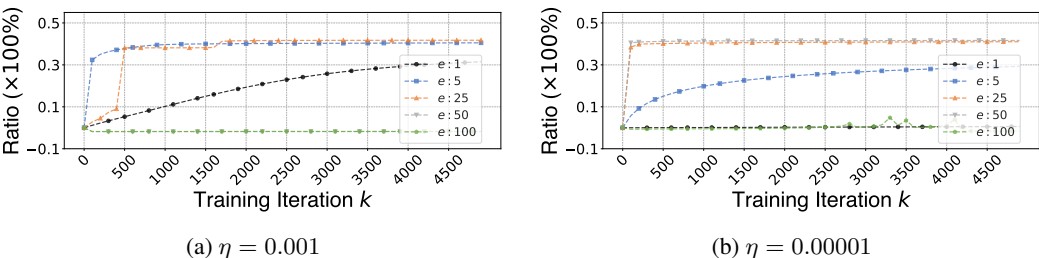

(a) $\eta = 0.001$          (b) $\eta = 0.00001$

Figure 11: Additional Ablation Studies of $e$ with Different Learning Rates.

# D Discussion

**Scope of Theoretical Guarantees.** Our theoretical analysis establishes convergence guarantees and demonstrates superior convergence rates specifically for *over-parameterized* Math-L2O systems compared to baseline optimization algorithms. While we acknowledge the empirical effectiveness of certain *under-parameterized* Math-L2O systems [23, 34], providing theoretical convergence proofs for them remains challenging due to the inherent non-convexity of the underlying neural network training. Alternative theoretical approaches, such as convex dualization [17, 18, 31], have been explored. However, these methods typically necessitate the inclusion of regularization terms within the loss function, which may deviate from the original optimization objective we aim to solve.

**Generalization to Other Objective Functions.** The central thesis of Section 3 is that learning can enhance algorithmic convergence. To substantiate this claim, we first require a convergence guarantee for the neural network training process—a well-known complex problem. We leverage Neural Tangent Kernel (NTK) theory, which typically analyzes convergence under an $L_2$-norm objective [16]. Despite generalizations of NTK to other loss functions [9, 40], we retain the $L_2$-norm for two reasons: (1) it permits the derivation of an explicit convergence rate, rather than a surrogate one [40], and (2) it aids in demonstrating a deterministic initialization strategy, which has practical implications for model and training design.

**Choice of Base Algorithm.** Our framework utilizes Gradient Descent (GD) as the core algorithm primarily because it admits a direct analytical formulation relating the initial point $X_0$ to the iterate $X_T$. This tractability is crucial for our analysis. In contrast, accelerated variants like Nesterov Accelerated Gradient Descent (NAG) [4] generally lack such closed-form expressions for $X_T$. This absence significantly complicates the derivation of the output bounds required to analyze the L2O system's dynamics and to prove convergence guarantees. Consequently, rigorously extending our current theoretical framework to momentum-based methods, despite attempts using inductive approaches, remains an open challenge.

We contend that a convergence proof for NAG can be constructed. Our central strategy involves bounding the L2O model's output to satisfy the convergence conditions of the backbone algorithm. This is analogous to our use of the $\beta$-smoothness property to derive the step size in Equation (3) and is a methodology applicable to any provably convergent algorithm. To this end, we aim to bound $X_T$ relative to $X_0$. The proof proceeds as follows: First, NAG is formulated as a linear dynamical system where a transition matrix maps $X_t$ to $X_{t+1}$. Second, we constrain the neural network outputs (i.e., momentum terms and step sizes) to ensure the transition matrix remains bounded over $T$ steps. Finally, by applying the Cauchy-Schwarz and Triangle inequalities to this stable system, a formal bound on $X_T$ is derived.

## E    Impact Statement

This paper presents work whose goal is to advance the field of Learning Theory and its combination with optimization. There are many potential societal consequences of our work, none of which we feel must be specifically highlighted here.

