# OpenReview forum: "Learning Provably Improves the Convergence of Gradient Descent"
_NeurIPS.cc/2025/Conference — NeurIPS 2025 poster_

### Official Review · Reviewer_VMJi · 2025-06-21

**Clarity:** 2
**Significance:** 3
**Originality:** 3
**Rating:** 4
**Confidence:** 2

**Summary:**

This paper addresses a significant and often overlooked issue in the Learning to Optimize (L2O) field: the formal proof of convergence for the training process of L2O models. The authors focus on the Math-L2O framework and leverage Neural Tangent Kernel (NTK) theory to prove that learning hyperparameters for Gradient Descent (GD) can provably enhance its convergence rate. A novel initialization strategy is also introduced to support the theoretical claims and ensure training stability.

**Questions:**

While the contributions are significant, the paper's clarity could be improved in several key sections. The following points should be addressed to strengthen the manuscript.

**Ethical Concerns:**

["NO or VERY MINOR ethics concerns only"]

**Final Justification:**

I am not very familiar with this field. If my understanding is correct, this work would be highly significant. However, the current writing requires more careful revision. Overall, the manuscript remains somewhat unrefined and requires further careful revision.

**Limitations:**

In section 3.1: the goal of this section is to analyze the limitations of existing frameworks, but the presentation could be more specific and balanced.

- In Lines 157-162, the descriptions of prior work use general terms like "stringent conditions" and "necessary conditions" without immediately specifying what those conditions are. It would be more impactful to state the specific limitations upfront before the empirical analysis.
- The section provides a detailed empirical analysis of the limitations of LISTA-CPSS (e.g., training instability and violation of sign consistency conditions). However, a similar analysis for the Math-L2O framework, which is also introduced as a target of analysis in this section, appears to be missing. The text should clarify the specific limitations of the original Math-L2O framework that this work aims to overcome.

In section 3.2: This section introduces the core conceptual framework of the paper, but its presentation is difficult to follow.
- Can you explain that why $X_{T}^0$ occurs two sides of equation (9)?
- In Line 202, the paper asserts that Equation (10) demonstrates a sub-linear convergence rate of at least $O(1/T^2)$. Can you illustrate it more details?

**Quality:**

3

**Strengths And Weaknesses:**

The primary strength of this work lies in its novelty and significance. Many L2O frameworks implicitly assume that the underlying neural network models are trainable and will converge during training.

---

> ### Author Rebuttal · Authors · 2025-07-29
>
> Thanks for your questions and suggestions.
>
> 1. (L1) Presentation of Section 3.1
>
>    1. (L1.1) Explanation for "stringent conditions" and "necessary conditions"
>
>       A: As in lines 171-181, page5, the stringent conditions of LISTA-CPSS are some necessary conditions for learnable parameters and variables to ensure convergence, which are our evaluation targets that are detailed in lines 171-181. The results are in Figure 3(a).
>
>       The necessary conditions (line 161) of Math-L2O do not prove sufficient conditions to guarantee that the proposed models can converge. It only assumes it converges, which is a necessary condition, to derive a reasonable architecture of neural network models.
>
>    2. (L1.2) Empirical results of Math-L2O
>
>       A: The evaluation results of Math-L2O are in Figure 3(b).
>
>       Due to the page limit, we put it along with the results of the LISTA-CPSS. The result shows the poor training convergence of Math-L2O in large steps $T$.
>
> 2. (L2)  Presentation of Section 3.2
>
>    1. (L2.1) $X_T^0$ in both size of inequality.
>
>       A: This is a typo. And there are two typos in inequalities (9) and (10). The right-hand side (RHS) of inequality (9) in line 197 should be $\frac{\beta}{T}\Vert X_0^0-X^*\Vert_2^2$, since the evaluating step is the subscript $_T$. The LHS represents the objective after $T$ steps. The RHS represents the objective of initial $X_0^0$.
>
>       And the complete inequality should be $F(X_T^0) \leq \frac{\beta}{T}\Vert X_0^0-X^*\Vert_2^2$, which illustrates a convergence rate within $T$ steps of Gradient Descent algorithm.
>
>       Moreover, the inequality (10) in line 200 should be $F(X_T^k) \leq r_k \frac{\beta}{T}\Vert X_0^k-X^*\Vert_2^2$.
>       This is due to $X_0$ being a given initial point, which is not violated by training iteration $k$.
>       In this inequality, the superscript is $^k$, which means that this is not related to training iteration.
>       The subscripts in the left-hand-side (LHS) are $_T$ and in the RHS are $_0$, which formulates the convergence rate of L2O model within $T$ steps.
>
>    2. (L2.2) Sub-linear convergence rate of $\mathcal{O}(1/T^2)$
>
>       A: This is a typo. The right version should be $O(1 / T)$, which is the convergence rate of vanilla gradient descent. We will fix it in final version. However, this is minor compared to our main contribution.

---

> > ### Comment · Reviewer_VMJi · 2025-08-01
> > **Reply to rebuttals**
> >
> > I basically agree with your rebuttal opinion. In view of the completeness of the paper, I will maintain the current score.

---

> > > ### Author Response · Authors · 2025-08-02
> > >
> > > Thanks for your comment.
> > >
> > > We think our paper is complete enough to demonstrate our claimed contributions. As in our title, we mainly focus on the training convergence demonstration for L2O. We take a step-by-step methodology to achieve it. We give a framework in Section 3 to show that learning can improve the convergence of an existing algorithm, where the prerequisite is the convergence of NN training within L2O. On the other hand, this framework ensures the convergence of L2O upon the convergence of an algorithm. Then, in Section 4, we prove the theorem to demonstrate the convergence of L2O training. Further, we apply the theorem and prove an initialization strategy to fulfill the conditions in the theorem.
> > >
> > > All the proofs and demonstrations are rigorously constructed in the Appendix. First, we present the detailed derivative of our L2O (Sections A1-A3). Then, we prove some tools in Section A4, such as the bound of L2O's outputs, semi-smoothness properties of L2O, and the bound of gradients. Moreover, following the methodology of Nguyen ICML 2021, we prove the convergence of NN training by induction in Section A5, where we split the objective function (by perfect square) into three parts and bound each part respectively. In the end, we merge the terms to formulate the convergence rate in Theorem 4.3.
> > >
> > > All the above descriptions are detailed in the Appendix and sketched in the main pages.
> > >
> > > For the generalization of our L2O, we think it is not within the scope of this paper. However, we do have some empirical results in Figure 8, where we use the L2O trained by 100 steps to optimize unseen optimization problems with 3000 steps. The results show fast and stable convergence.

---

### Official Review · Reviewer_u4ci · 2025-06-23

**Clarity:** 2
**Significance:** 3
**Originality:** 2
**Rating:** 4
**Confidence:** 3

**Summary:**

This work proves that the L2O methods probably improve the convergence of gradient descent.
The authors establish that L2O can improve GD convergence rates by a factor of $r_k$ and prove training convergence of the L2O network using NTK theory in the over-parameterized regime. To bridge theory and practice, they propose an initialization strategy that satisfies the theoretical assumptions. Experimental results demonstrate improved convergence stability of the trained optimizer and validate the effectiveness of the proposed initialization scheme.

**Questions:**

1. Regarding the improvement of L2O over GD:
   - What are the upper and lower bounds of this improvement?
   - Does this improvement stem primarily from learning optimal hyperparameters (as mentioned in line 205), or is there another explanation?
   - What is the worst-case/generalization performance of the trained L2O optimizer? Is it consistently superior to GD across all test instances with varying problem parameters?
2. Regarding neural network width requirements:
   - Section 4 requires NN width to be O(Nd), which is reflected in experiments (e.g., width 5120 in line 320). Do smaller widths (e.g., O(d) as suggested by universal approximation theory) perform adequately in practice?
   - Is there a direct performance comparison among the three neural network architectures in Table 1?

**Ethical Concerns:**

["NO or VERY MINOR ethics concerns only"]

**Final Justification:**

Initially, I scored 3, given the unclear improvement of L2O methods compared with gradient descent.
After the clarification by the authors, I raised my score to 4, but not higher due to:
- The convergence analysis for overparameterized L2O seems less critical than its generalization property.

**Limitations:**

See the weakness.

**Quality:**

3

**Strengths And Weaknesses:**

Strengths
- It provides solid theoretical analysis for L2O training convergence using NTK theory in the over-parameterized setting.

Weakness

- Convergence improvement claims require clarification (Section 3.2):
     - The relationship between the LHS of Eq. (8) and the RHS of Eq. (9) in line 198 is unclear. Does this indicate that an L2O algorithm's initial state matches non-learning GD algorithms? If so, does the Deterministic Initialization in Sec. 5 satisfy this requirement?
     - The author claims "Eq. (10) demonstrates a sub-linear convergence rate of at least $O(1/T^2)$", which is unclear since Eq. (10) shows a convergence rate of $O(r_k \beta / T)$ and $r_k$ is independent of $T$.
     - $r_k$ in Eq. 10 seems not to appear in Sec. 5 or Sec. 6.
     - A formal theorem with precise definitions and assumptions is needed to substantiate these claims.
- Novelty clarification needed:
    - The technical contribution when applying NTK theory to prove linear convergence of L2O training in Sec. 4, compared to the cited work (Nguyen 2021), should be stated more explicitly.

I will reconsider my evaluation if these concerns are addressed.

---

> ### Author Rebuttal · Authors · 2025-07-29
>
> Thanks for your questions and suggestions.
>
> Weaknesses:
>
> 1. Weakness 1: Presentation
>
>    1. (W1) Presentation of convergence improvement
>
>       1. (W1.1) Relationship between Eq. (8) and Eq. (9).
>
>          A: There are two typos in inequalities (9) and (10). The right-hand side (RHS) of inequality (9) in line 197 should be $\frac{\beta}{T}\Vert X_0^0-X^*\Vert_2^2$, since the evaluating step is the subscript $_T$. The LHS represents the objective after $T$ steps. The RHS represents the objective of initial $X_0^0$.
>
>          And the complete inequality should be $F\left(X_T^0\right) \leq \frac{\beta}{T}\Vert X_0^0-X^*\Vert _2^2$, which illustrates a convergence rate within $T$ steps of Gradient Descent (GD) algorithm.
>
>          Moreover, the inequality (10) in line 200 should be $F\left(X_T^k\right) \leq r_k \frac{\beta}{T}\Vert X_0^k-X^*\Vert _2^2$. This is due to $X_0$ is a given initial point, which is not violated by training iteration $k$. In this inequality, the superscript is $^k$, which means that this is not related to training iterations. The subscripts in the left-hand-side (LHS) are $_T$ and in the RHS is $_0$. which formulate the convergence rate of L2O model within $T$ steps.
>
>
>         2. (W1.2) Alignment between L2O algorithm's initial state and GD.
>
>             A: Yes.
>
>             Our proposed deterministic initialization method achieves the satisfaction of this condition by setting the last layer of neural network (NN) model to be all ones and eliminating bias term. This will give all zero output before activation layer. By a $2\times$ sigmoid activation function. The final output is all one. We manually produce a $1/\beta$ to achieve an identical step size with GD. These are detailed in Section 5.1.
>
>    2. (W2) Sub-linear convergence rate of $O\left(1 / T^2\right)$
>
>       A: This is a typo. The correct version is $O\left(1 / T\right)$. We will fix it in the final version.
>
>
>    3. (W3) Connection between Section 3 and Section 4
>
>       A: In Section 3, we use denotation $r_k$ as a general symbol to illustrate our main framework. In the first paragraph (lines 211-216) of Section 4, we use text descriptions to explain the relationship. In our Theorem 4.3 it is exactly $\left(1-4 \eta \frac{\beta_0^2}{\beta^2} \delta_4 \alpha_0^2\right)^k$. We will add this explanation of the correlation between $r_k$ and our demonstrated convergence rate in Theorem 4.3 in the final version.
>
>    4. (W4) Precise definitions and assumptions for lemmas and theorems
>
>       A: Thanks for your suggestion. As in Equation 11, to simplify the formulations in lemmas and theorems, we define several quantities. Each quantity represents an inner formulation in the demonstration of lemmas and theorems. The detailed step-by-step demonstrations are in the Appendix. In response 2 to Reviewer ooXd, we give an explanation and analysis of the magnitude for all quantities. We will add them to the paper in the final version.
>
> 2. Weakness 2: Novelty of NTK
>
>    A: Thanks for your suggestion. We are the first to utilize the new NTK framework in [2] to demonstrate an explicit convergence of RNN-like models.
>
>    In this work, we apply the NTK framework proposed in [2] to demonstrate the convergence rate of L2O. Our main contribution is to originally prove that training of L2O can yield a convergence-improved algorithm (L2O itself). However, the utilization of the NTK is not identical to that in [2]. The L2O framework (named as unrolling) recurrently uses one NN model to generate the step size of GD, which is demonstrated to be a more complicated case of convergence demonstration with NTK in the following citation. Also, we do not mainly claim the originality of utilizing the NTK framework in [2] for RNN-like models. However, to our knowledge, we are the first to achieve that.
>
>    [1] Allen-Zhu et al . On the convergence rate of training recurrent neural networks. NeurIPS 2019.
>
>    [2] Quynh Nguyen. On the Proof of Global Convergence of Gradient Descent for Deep ReLU Networks with Linear Widths. ICML, 2021.
>
>
> Questions:
>
> 1. (Q1) Improvement of L2O over GD
>
>    1. (Q1.1) Lower and upper bounds.
>
>       A: Lower bound is 0. Upper bound is 1.
>
>        From Theorem 4.3 and the above explanations. The improvement is determined by $r_k$, i.e., $r_k := \left(1-4 \eta \frac{\beta_0^2}{\beta^2} \delta_4 \alpha_0^2\right)^k$, which is a term less than one since $\delta_4=\sigma\left(\delta_3 \Theta_L\right)\left(1-\sigma\left(\delta_3 \Theta_L\right)\right) >0$ and $\alpha_0:=\sigma_{\min }(G_{L-1, T}^0) > 0$ ($G_{L-1, T}^0$ is a thin matrix). Trivially,$r_k$'s upper bound is one. From Equation (10), this implies that L2O is at least as fast as GD. However, we think that the investigation of the upper and lower bounds of the improvement is necessary to conduct new research work. Thanks for this question, which provides us with a new interesting question.
>
>    2. (Q1.2) Essential of improvement.
>
>        A: Yes.
>
>        As in Section 3, improvement of training is orthogonal to backbone algorithm steps. Intuitively, it can be somehow illustrated as finding a shortcut from offline training. After that, we will benefit in faster online inference.
>
>    3. (Q1.3) Generalization
>
>       A: Empirically, the inference performance in Figure 8 shows that it generalizes well.
>
>       The theoretical generalization analysis of solving optimization problems with L2O on test instances is a non-trivial research topic as well. This work does not include this part. Some existing works focus on this area. We list some citations in the following.
>
>       [1] Sucker, Michael, and Peter Ochs. A Generalization Result for Convergence in Learning-to-Optimize. ICML 2025.
>
>       [2] Qingyu Song et al. Towards Robust Learning to Optimize with Theoretical Guarantees. CVPR 2024.
>
> 2. (Q2) Neural network width requirements
>
>    1. (Q2.1) NN width requirement
>
>       A: Yes. Scale 3 in Table 1, page 40, is an under-parameterized case.
>
>       $O(d)$ represents an under-parameterized case, where the layer of NN is not wide. We do have some under-parameterized experiments in Appendix. For example, in Table 1 Appendix C.1 page 40, the Scale 3 is the case where the width of NN layer is only 20, which is extremely smaller than input dimension $10\times512$. The results (orange line) in Figure 7 show that the model still converges. For this, we would like to note that the proved bounds and conditions for training convergence are an upper bound. The tightness of the bound derived from NTK is a non-trivial research topic. In the future, we aim to improve the bound proposed in this paper.
>
>    2. (Q2.2) Performance comparison of scales in Table 1
>
>       A: They are not on the same scale.
>
>       The first two models with different scales are designed to solve problems that are not comparable. The third model, although targeting the optimization problem with the same dimension, has a different number of training samples, i.e., N. It is designed to align with the training configurations of the baseline model, LISTA-CPSS. Moreover, due to the GPU memory limitation, we set a thin NN, whose convergence is not guaranteed by our proposed theorem. This experimental result is used to further demonstrate our proposed framework in Section 3. Thus, we would like to note that the three scales are not comparable.

---

> ### Comment · Reviewer_u4ci · 2025-08-01
> **Response to Author Rebuttal**
>
> Thank you for your detailed response.
>
> Regarding the improvement of L2O over GD, I have some remaining concerns about the lower bound:
>
> - If we consider the limit case where training iterations $k\to \infty$, theoretically, the trained NN-optimizer could reach the optimum in a 1 gradient step. This would imply that the NN predicts $ g_W(x_0, \nabla F(x_0)) = \beta (X_0 - X^*) / \nabla F(x_0)$ according to Equation (3).
>   - How does this mechanism generalize to arbitrary initial points $X_0$ and problem parameters $M$ (even when limited to the training dataset)?
>   - Specifically, since $M$ does not appear to be explicitly treated as an input to the neural network, how does the network predict $M$-dependent optimal solutions in a single step?
>   - Alternatively, is this simply a case of the network memorizing the entire training dataset and exhibiting overfitting, as suggested in your response to Reviewer ooXd?

---

> > ### Author Response · Authors · 2025-08-01
> >
> > Thanks for your questions.
> >
> > Your summarization of one-step convergence is correct. It is achievable since our predicted steps ($P_t$) are coordinate-wise (as in Equation 4). However, in general, the overfitting problem is inherent to machine learning, which means that "zero-loss" training is achievable. But overfitting-avoidance is beyond the scope of this paper. We cannot avoid overfitting if we continuously train the L2O to achieve that one-step convergence. On the other hand, the conclusion from convergence rate is also consistent with overfitting.
> >
> > * First, if you assume that the L2O is overfitted, the answer is that it cannot generalize. However, if your question is about the generalization of our theorems, the answer is that our lemmas and theorems do not rely on $X_0$ and $M$. The evidence is that they are defined in the quantities of Equation 11.
> > * $M$ is implicitly included in the gradient term of input (Equation 4).
> > * Yes. However, this raises another interesting question: whether we can control the step $T$ to alleviate overfitting. Thanks for the insightful question. It provides us with a new idea: investigating the overfitting problem within our proposed framework. However, it will be included in our future work.

---

> > > ### Comment · Reviewer_u4ci · 2025-08-01
> > > **Response**
> > >
> > > Thank you for the detailed clarification.
> > >
> > > The authors claim that `if you assume that the L2O is overfitted, the answer is that it cannot generalize`:
> > > - Overfitting in over-parameterization settings is different: In the NTK regime, it has been shown that gradient descent can train an over-parameterized neural network to achieve good training and test accuracies, e.g., [1].
> > > - I am wondering, in your setting, why the fully-trained overparameterized NN can not generalize.
> > >
> > > [1] Allen-Zhu, Zeyuan, Yuanzhi Li, and Yingyu Liang. "Learning and generalization in overparameterized neural networks, going beyond two layers." Advances in neural information processing systems 32 (2019).

---

> > > > ### Author Response · Authors · 2025-08-02
> > > >
> > > > Thanks for your question.
> > > >
> > > > We do not claim that our L2O is overfitted. Our results in Figure 8 show that inference with more than 100 steps (step construction in training) has empirically demonstrated that our L2O, trained using our proposed theorems, generalizes very well.
> > > >
> > > > We need to admit that the referred generalization analysis method is not considered in this work. The technique takes another methodology, the Universal Approximation Theorem (UAT), which bounds and minimizes the generalization gap between expected risk (loss under a distribution) and empirical risk (loss from a sampled dataset). However, for our RNN-like L2O, the risk with the predicted $X\^*$ is really hard to formulate and construct a bound. We find a reference [1] that successfully bound the risk with NTK. However, it is different from our L2O in two aspects. First, the RNN considered is the Elman RNN with ReLU activation. Our L2O can be regarded as Elman RNN plus skip connection ($X_{t+1} = X_t + \dots$) with sigmoid activation, which is a more complex architecture. Second, it is constrained by classical NTK theorems that require an infinitely wide NN, which is not consistent with our $\mathcal{O}(Nd)$ width solution.
> > > >
> > > > However, we are not experts on the Universal Approximation Theorem. We would very much like to discuss any further questions!
> > > >
> > > > [1] Wang, Lifu, et al. On the provable generalization of recurrent neural networks. NeurIPS. 2021.

---

> > > > > ### Comment · Reviewer_u4ci · 2025-08-02
> > > > > **Response**
> > > > >
> > > > > Thanks for your detailed clarification. I will adjust my score accordingly.

---

> > > > > > ### Author Response · Authors · 2025-08-04
> > > > > >
> > > > > > We sincerely appreciate the time and effort you have dedicated to reviewing our manuscript. Your valuable comments and suggestions are instrumental in enhancing the quality of our work.

---

### Official Review · Reviewer_ooXd · 2025-06-30

**Clarity:** 1
**Significance:** 3
**Originality:** 4
**Rating:** 5
**Confidence:** 4

**Summary:**

This paper studies the convergence of learning an NN-based optimizer that is used to solve the quadratic optimization problem. In particular, the paper focus on Math-L2O, whose learned optimizer outputs a step size which is applied in the gradient descent step of the base optimization problem. By leveraging the NTK-based analysis, the paper derives a linear convergence rate of the optimizer training. The paper also provided experiments to validate the theoretical results.

**Questions:**

None.

**Ethical Concerns:**

["NO or VERY MINOR ethics concerns only"]

**Final Justification:**

It is seems that my previous major concern is due to the typo in the paper, which the author has promised to fix in the camera ready version.

**Limitations:**

Yes.

**Paper Formatting Concerns:**

None.

**Quality:**

3

**Strengths And Weaknesses:**

**Strengths**

1. The problem tackled by this paper is both significant and difficult. To further support the contribution of the paper, experimental results showing that the assumptions made in previous works are unrealistic are also included.

2. The paper deals with a complicated deep MLP optimizer in the Math-L2O setting. This is both technically difficult and general in terms of the result.

3. The convergence result proved by the paper is a clean linear convergence with no extra error terms.

**Weaknesses**

1. The main concern I have about the paper is that Theorem 4.3 shows that applying the output step size to the GD optimization problem with a fixed number of steps $T$, the final loss converges to zero as the training steps $k$ goes to infinity. This is particularly confusing as it seems that there is no requirement on $T$. In particular, this means that, as long as we do sufficient training to the optimizer model, its output step size can result in a convergence of the base optimization problem even with one step of gradient descent, which seems super intuitive. I suggest the author to provide a reasonable explanation of this surprising result.

2. The choice of the step size in (13a) and (13b) seems complicated. Yet, the paper does not provide a good interpretation of what these expressions means and how the step size scales intuitively. Without such interpretation, it is possible that the step size are super small and lead to very slow convergence.

3. Some of the notations are unclear. For instance,  $W\_{[L]}$ never appears on the right-hand side of Eq.(2), where the dependency should be through $X\_T$. In Eq.(6) and Eq.(7) I believe that parentheses are missing in the product terms, and similarly for the equation in Lemma 4.2, where the last parenthesis should include $||W\_{\ell}^{k+1} - W\_{\ell}^k||_2$. Moreover, It is a bit unclear what $F(W^k)$ really denotes, although it could be inferred from the text that it is $F(X\_T, W\_{[L]})$.

---

> ### Author Rebuttal · Authors · 2025-07-29
>
> Thanks for your questions and suggestions.
>
> Weakness:
>
> 1. (W1) Intuitive of our proposed framework
>
>    A: Yes. As demonstrated in our Section 3, the reader will find that the training iteration (denoted as $k$) is orthogonal to the convergence metric, i.e., step $T$, where the convergence of training performs an incremental convergence improvement on gradient descent (GD) by $r_k$. Our Theorem 4.3 (after a long journey) shows that we can achieve $r_k$ by proving a learning convergence from the NTK theory. Thus, it is natural to expect that we can utilize exhausted training to achieve extremely fast convergence and even reach the optimum by one step. This is known as the over-fitting problem in the machine learning community. Over-fitting will lead to poor robustness of inference. However, the generalization of solving optimization problems of L2O on test instances is a non-trivial research topic as well. This work does not include this part. Some existing works focus on this area. We list some citations as follows.
>
>    [1] Sucker, Michael, and Peter Ochs. A Generalization Result for Convergence in Learning-to-Optimize. ICML 2025.
>
>    [2] Qingyu Song et al. Towards Robust Learning to Optimize with Theoretical Guarantees. CVPR 2024.
>
>    Moreover, the training complexity for different steps is different. Training a L2O with a small $T$ is more complex than training a L2O with a large $T$. From the convergence rate of L2O in Equation (10), a small $T$, for example, 1, leads to a smaller right-hand-side (RHS) than a large $T$, 10. This requires a smaller $r_k$ for $T=1$ case. As in the convergence rate term from Theorem 4.2, we set a proper learning rate to ensure a consistent convergence rate for both cases. We need more training iterations for $T=1$ case. Moreover, if we would like to utilize fewer training iterations, from the following interpretation for the learning rate settings, we need to set a larger learning rate. However, this will lead to the fluctuation around the optimum for stochastic optimizers, for example, SGD, which requires more complicated training techniques to stabilize training.
>
>
> 2. (W2) Step size in Equations (13a) and (13b)
>
>    A: Equations (13a) and (13b) are based on the quantities defined in Equation (11). Each quantity represents an inner formulation in the demonstration of lemmas and theorems.
>
>    Due to the page limit, we put the main results on the main pages. We use these quantities to simplify the formulations. The detailed step-by-step demonstrations are in Appendix. First, we would like to introduce their magnitudes as follows. We will add this interpretation to the main page in the improved version, where we will have one bounce page. In the denotations, we use subscripts like $T$ and $L$ to emphasize that the defined constant terms are related to step $T$ and training iteration $k$, and use superscripts or subscripts like $j$ and $t$ to emphasize that the defined scalar value functions are related to step $t$ and $j$. We denote $\bar{\lambda}\_{\min}, \bar{\lambda}\_{\max} = \min\{\bar{\lambda}\_\ell \}, \max\{\bar{\lambda}\_\ell \},\ell \in [L]$, which are some constant upper bounds of the singular value of the NN layers (defined in Equation (11)).
>
>    * $\bar{\lambda}_{\ell}$:  A positive constant upper bound for each NN layer $\ell$. Demonstrated in the proof of Theorem 4.3.
>    * $\Theta\_{L}$: A positive constant w.r.t. $\bar{\lambda}\_{\ell}$. $\Theta\_{L}$ is lower and upper bounded by $\Omega(\bar{\lambda}\_{\min}^L)$ and $\mathcal{O}(\bar{\lambda}\_{\max}^L)$, respectively. Moreover, $\Theta\_{L}^{-1}$ is $\Omega(\bar{\lambda}\_{\max}^{-L})$ and $\mathcal{O}(\bar{\lambda}\_{\min}^{-L})$.
>    * $\Phi_j$: A scalar-valued function w.r.t. step $j$. $\Phi_j$ is $\mathcal{O}(j)$ and $\Omega(j)$.
>    * $\Lambda_{j}$: $\mathcal{O}(j^2)$ and $\Omega(j^2)$.
>    * $S\_{\Lambda,T}$ and $S\_{\bar{\lambda},L}$: $S\_{\Lambda,T}$ is $\mathcal{O}(T^3)$ and $\Omega(T^3)$. $S\_{\Lambda,T}^{-1}$ is $\mathcal{O}(T^{-3})$ and $\Omega(T^{-3})$. $S\_{\bar{\lambda},L}$ is $\Omega(L\bar{\lambda}\_{\max}^{-2})$ and $\mathcal{O}(L\bar{\lambda}\_{\min}^{-2})$. Moreover, $S\_{\bar{\lambda},L}^{-1}$ is $\Omega(L^{-1}\bar{\lambda}\_{\max}^{2})$ and $\mathcal{O}(L^{-1}\bar{\lambda}\_{\min}^{2})$.
>    * $\zeta_1$ and $\zeta_2$: $\zeta_1$ and $\zeta_2$ are both $\Omega(T)$ and $\mathcal{O}(T)$.
>    * $\delta\_1^t$: $\Omega(t\bar{\lambda}\_{min}^{Lt})$ and $\mathcal{O}(t\bar{\lambda}\_{max}^{Lt})$.
>    * $\delta\_2$: $\delta\_2$ is $\Omega(T\bar{\lambda}\_{min}^{LT})$ and $\mathcal{O}(T\bar{\lambda}\_{max}^{LT})$. $\delta\_2^{-1}$ is $\Omega(T\bar{\lambda}\_{max}^{-LT})$ and $\mathcal{O}(T\bar{\lambda}\_{min}^{-LT})$.
>    * $\delta_3$: Both $\Omega(T)$ and $\mathcal{O}(T)$.
>    * $\delta\_4$: $\delta\_4$ is  $\Omega(\exp(-T\bar{\lambda}\_{max}^L))$ and $\mathcal{O}(\exp(-T\bar{\lambda}\_{min}^L))$. Moreover, $\delta\_4^{-1}$ is $\mathcal{O}(\exp(T\bar{\lambda}\_{max}^L))$ and $\Omega(\exp(T\bar{\lambda}\_{min}^L))$.
>
>    Next, we analyze the magnitude of learning rate $\eta$, where we would like to highlight that it is acceptable by the NTK framework since training does not need large learning rates in NTK theory. Our learning rate settings are designed to be close to these results. The results also empirically demonstrate that convergence from NTK theory does not need a large learning rate.
>
>    1. Typo in Equation (13a): There are some typos in the subscricpts, the revised version is ...$ \big( \delta_2 + \Theta_L S_{\Lambda,T} S_{\bar{\lambda},L} \big)^{-1} S_{\Lambda,T}^{-2}$.
>
>    2. We calculate the scales of learning rate $\eta$ w.r.t., $\bar{\lambda}_{\max},T,L$, which are likely to scale with L2O configurations. $T$ is the GD steps and $L$ is NN layers. We can control these values by some specific initialization methods.
>
>    Equation (13a): We calculate the scale of $\eta$ by $\mathcal{O}(\frac{T\bar{\lambda}\_{max}^{LT} + T^2}{((T\bar{\lambda}\_{max}^{LT})+\bar{\lambda}\_{max}^{L}T^3L\bar{\lambda}\_{max}^{-2})T^6})$. The value is reaching one with the increase of $\bar{\lambda}\_{max}$. Learning rate will not be extremely small with some proper initialization method, such as our proposed method in Section 5.
>
>    Equation (13b): $\eta$ is correlated with lower bound of $\alpha_0$, which is the singular value of the layer before last layer (line 223, section 4.1). Moreover, in Eq. 12-(abcd), we demonstrate four lower bounds of $\alpha_0$. We then analyze the magnitude of $\eta$ in each case. We aim to formulate the upper bound of learning rate $\eta$.
>
>    * Inequality (12a) holds: $\eta$ is $\mathcal{O}(\exp(2T\bar{\lambda}\_{\max}^L)T^{-2})$.
>
>    * Inequality (12b) holds: $\mathcal{O}(\bar{\lambda}\_{\max}^{2-LT-2L}T^{-4}L^{-1})$.
>
>    * Inequality (12c) holds: $\mathcal{O}(\bar{\lambda}_{\max}^{-L}T^{-3})$.
>
>    * Inequality (12d) holds: $\mathcal{O}(\exp(T\bar{\lambda}\_{\max}^L)(\bar{\lambda}\_{\max}^{2L}T^2L)^{-\frac{2}{3}})$.
>
>    One will see that the increase of $\bar{\lambda}_{\max}$ leads to smaller learning rates. However, this will not lead to slow convergence. First, theoretically, a small learning rate does not contradict the NTK theory. The vanilla NTK theory [1] establishes a methodology that uses an infinitely wide NN with enormous neurons. Then, linear convergence can be obtained by finding an optimal solution within a small closed space around initialization [1]. Thus, a large learning rate is not necessary in NTK theory. Second, our empirical results show that the convergence performance of training is not sensitive to learning rates.** For example, in Figure (4a), learning rates from $10^{-3}$ to $10^{-7}$ have similar convergence speeds.
>
>    Moreover, the small learning rate setting is a compromise to avoid further increasing the width of NN. In some works [2] [3], although the infinite width requirement is eliminated, the polynomial width is still required, such as $N^3$ in [2] and [3], where $N$ is the number of samples. In this work, the number of samples is proportional to the dimensionality of the optimization problem in the coordinate-wise L2O (treat each dimension independently, defined in lines 118-122), where d dimensions feature are reshaped into the sample dimension, the input of NN is $Nd$ rows, defined in line 115. Hence, we take a different compromise by setting a smaller learning rate to avoid an extremely wide NN.
>
>    [1] Arthur J. et al. Neural tangent kernel: Convergence and generalization in neural networks. NeurIPS, 2018.
>
>    [2] Allen Z. et al. On the convergence rate of training recurrent neural networks. NeurIPS 2019
>
>    [3] Quynh Nguyen. On the Proof of Global Convergence of Gradient Descent for Deep ReLU Networks with Linear Widths. ICML, 2021
>
> 3. (W2) Unclear notations
>
>    A: Thank you for figuring out the typos. We will fix them in the final version.
>
>    $W_{[L]}$: We use it to represent the learnable parameters of a L2O model, where we eliminate all bias terms in every layer. The $X_T$ is generated by $W_{[L]}$ and a given $X_0$.
>
>    Parentheses: Due to character limit, we will fix it in the improved version.
>
>    Lemma 4.2: Fixed: ...$\Lambda\_{s} \Theta\_L\big(\sum\_{\ell=1}^{L} \bar{\lambda}\_{\ell}^{-1} \Vert W\_{\ell}^{k+1} - W\_{\ell}^{k} \Vert\_{2}\big).$ Proof is in line 613, Appendix.
>
>    $F(W^k)$: Sorry for the misleading. $W$ here is $W_{[L]}$ in Equation (2). $F(W^k)$ represents $F(X_T^k; W^k)$. Here, as in Equation (2), we use $;$ to represent that $X_T^k$ is generated by $W^k$ with given $X_0$ (aks, $X_0^k$ since it is not related to $k$). To make it clearer, we will define the objective in Equation (2) as $F(X)=\tfrac{1}{2}\Vert \mathbf{M} X-Y\Vert _2^2$, which only illustrates the target optimization problem and is not related to underlying methods.
>
>    For $W_{[L]}$, it is only used in problem definition in Equation (2), illustration of L2O's behavior in Equation (7). In other equations, $W$ is used. We will revise $W_{[L]}$ to be $W$.

---

> > ### Comment · Reviewer_ooXd · 2025-08-03
> >
> > Thank you for the detailed response, especially the analysis for the scale of the learning rate. I am still confused about W1 I raised, potentially due to the confusion about the dimensionality of the matrices. In particular, for Eq. (3), we know that $X\in\mathbb{R}^{Nd\times 1}$ and thus $\nabla F(X) \in \mathbb{R}^{Nd\times 1}$. In this case, what is the shape of $P_t$? If we can do a hadamard product between $P_t$ and $\nabla F(X)$, then $P_t$ should be $\mathbb{R}^{Nd\times 1}$. However, the paper also mentions that $P_t$ is a diagonal matrix, which requires it to be square. How should I understand this setup in terms of the shape of $P_t$?

---

> ### Author Response · Authors · 2025-08-04
>
> Thanks for your question. Sorry for the misleading description in line 123.
>
> $P_t$ is a vector. As defined in Equation (4), $P_t$ is the output of the NN by $2 \times $ sigmoid activation function. We will fix the typo by replacing "diagonal matrix" with "vector" in the related description in line 123.
>
> Throughout the duration of this study, the definition of $P_t$ was revised from the "diagonal matrix" to the vector form in Equation (4). Because we find that if we define $P_t$ to be a diagonal matrix, due to the chain rule, calculating the gradient of $P_t$ will yield matrix-by-matrix and vector-by-matrix derivatives. To simplify the formulation, we define $P_t$ as a vector. For example, in Equation (24) in line 521, page 15, Appendix, we need to calculate the derivative of NN's output ($P_t$) with respect to NN's parameters. In Equation (28) in line 529, page 15, Appendix, we need to calculate the derivative of L2O's output ($X_t$) to NN's output ($P_t$).
>
> We forgot to change the description in line 123.
>
> Thanks for raising this question for us. We will modify the description accordingly.

---

> > ### Comment · Reviewer_ooXd · 2025-08-04
> >
> > Thank you for the clarification. I think this paper makes a good contribution, so I am willing to raise the score.

---

### Official Review · Reviewer_yQfy · 2025-07-02

**Clarity:** 1
**Significance:** 2
**Originality:** 2
**Rating:** 4
**Confidence:** 3

**Summary:**

This paper provides a theoretical study of Learn to Optimize (L20) framework, where the optimization of the main model is performed using an additional model that generates per-coordinate stepsizes. This work provides convergence guarantees for such an optimization problem when the original problem that we want to solve using the main model is quadratic. The authors provide a linear convergence rate for the Math-L2O framework. The theoretical claims are supported by empirical ablations.

**Questions:**

- Can we use LSTM-type networks that suffer less from vanishing/exploding gradient problems instead of RNN-type updates?

- Can the theory be generalized to non-ReLU types of activations?

- Can the authors clarify why we obtain $O(1/T^2)$ rate (line 203)?

- Line 223 is confusing. Are $\alpha_0$, $\overline{\lambda}\_{\ell}$, $C\_{\ell}$ fixed? Can we always find such constants that (11) is verified? The next concern in this line is the assumption $\\{\\|W\_{\ell}^{k+1}\\|_2,\\|W\_{\ell}^{k}\\|_2\\}\le\overline{\lambda\_{\ell}}$. The proof of it is also unclear. Can the authors show how Weyl's inequality is used?

- The initialization (section 5) is unclear. The authors first set weight matrices to be zero, but then sample from a Gaussian distribution. Which one is used in the end?

- Typo: line 546, "eigenvalues" are missing

- The proof of Lemma A.2 is not full. How to obtain the final bound (31)?

- Where is the assumption $0 < P^k_t < 2$ verified which is used in some lemmas (e.g., Lemma A.1)?

- Can the authors provide more details on why the bound $\\|W\_j^{k+1}\\|\le\overline{\lambda}_j$ holds?

- Where do the authors use the NTK regime? Can the authors clarify this point, please? Is it used when assuming that $\sigma\_{\min}(G\_{L-1,T}^0) > 0$?

**Ethical Concerns:**

["NO or VERY MINOR ethics concerns only"]

**Final Justification:**

The authors addressed my main concerns during the rebuttals. I encourage them to incorporate their clarifications and modifications to the raised questions into the revised manuscript, such as:

- limited size of tested models and scalability of the approach;

- inaccuracies and typos in the claims and proofs;

- necessary NTK regime in the theoretical analysis and possible ways to avoid it.

**Limitations:**

- It seems that the proof techniques are extremely hard to generalize beyond the quadratic case.
- The authors discuss the limitations in Section D, mentioning that the explicit form for $X_T$ related to $X_0$ is crucial in the derivations. This is more complicated for other algorithms, e.g., Nesterov Accelerated Gradient.
- I believe it is hard to extend this approach to neural networks of real size since the input of the second NN is $[X_t,\nabla F(X_t)]$, which is a very high-dimensional nowadays.

**Quality:**

2

**Strengths And Weaknesses:**

***Strengths:***

- Convergence guarantees for a quadratic objective in NTK regime (which is somehow needed to ensure that PL condition holds and consequently leads to linear convergence).
- Experimental results demonstrate that Math-L2O framework might outperform classic algorithms like GD and Adam.
- The authors demonstrate that previous assumptions used in previous approaches, like LISTA-CPSS, are not verified during training (Figure 3)

***Weaknesses:***
- The paper is poorly written, and the technical details are hard to parse. Due to typos and/or inaccuracies in writing, overloaded notation it is hard to evaluate the theoretical results (e.g. $M_i, W_{[L]}$ are never defined in the main part; line 224 involves so many new definitions without any intuition why they are even needed for a presentation, how they are used, and so on). I also didn't find a specific place where some properties are explicitly used (the definition of NTK is also absent). Therefore, in my view, evaluation of the theoretical results requires a significant improvement in the presentation.

- Algorithms like Adam and SGD shine when training real networks of enormous sizes. Testing the performance of the proposed scheme against those algorithms on a toy quadratic problem is not a convincing result that can be used to claim "up to a 50% improvement". I acknowledge that deriving theoretical guarantees for Math-L2O seems to be a challenging task, but the evaluation has to be on at least MLP model on MNIST, which is a standard benchmark in DL (although small).

---

> ### Author Rebuttal · Authors · 2025-07-29
>
> Thanks for your questions and suggestions.
>
> Weakness (W) and Questions (Q):
>
> 1. Typos:
>
>    1. (W1) $\mathbf{M}_1$
>
>       A: Defined in line 467, page 13.
>
>    2. (W1) $W_{[L]}$
>
>       A: As in part 3 of the response to Reviewer ooXd, $W_{[L]}$ is only used in problem definition in Equation (2), illustration of L2O's behavior in Equation (7). In other equations, $W$ is used.
>
>    3. (Q3) $O(1 / T^2)$
>
>       A: Typo. Should be $O(1 / T)$.
>
> 2.  (Q7) Lemma A.2 Proof and (Q8) $0 < P_t^{k} < 2$
>
>    (Q7) A: $0 < P_t^{k} < 2$ is the configuration of activation by $2*sigmoid$.
>
>    (Q8) A: $\mathcal{D}$ is diagonalization operation on vector $P_t^{k}$, whose maximal eigenvalue is trivially bounded by $2$. We will add this to the appendix.
>
> 3. (Q4) Quantities in Line 223
>
>    A: They are symbols to simplify the formulations in lemmas and theorems.
>
>    First, the first two lines are formulations with pre-defined parameters. For example, $X_0$ is defined in Equation (3) and M and Y are defined in Equation 2. (A clearer definition is provided in part 3 of the response to Reviewer ooXd). Further, the remaining four lines include a composite formulation of first three lines and the parameters. Detailed analysis of these quantities is provided in part 2 of the response to Reviewer ooXd. They are used in formulations in Theorem 4.3 and step-by-step demonstrations in Appendix.
>
> 4. (Q4) Existence of $a_0, \bar{\lambda}\_\ell, C\_\ell$
>
>    A: They can be deterministically defined by our initialization methods and can be probably confined to a closed set by stochastic initialization methods.
>
>    Since $\bar{\lambda}\_{\ell} = \Vert W\_{\ell}^0\Vert_2 + C\_\ell$ and  $C\_\ell$ is a constant sequence (Section 4.1), if $\Vert W\_{\ell}^0\Vert\_2$ is known, $C\_\ell$ can be trivially constructed. Since $\alpha\_0 \coloneqq \sigma\_{min}(G^0\_{L-1,T})$, where $G^0\_{L-1,T}$ is generated by $W\_{\ell}^0$ and given input feature. It is bounded if $\Vert W\_{\ell}^0\Vert\_2$ is bounded as well. For a deterministic initialization, for example, our proposed method in Section 5, $W\_{\ell}^0$ is trivially deterministic. For a random initialization, for example, LeCun's initialization method introduced in Nguyen ICML 2021 [1], $W\_{\ell}^0$ is sampled from standard Gaussian distribution. It has a high probability of being bounded within the sphere around origin.
>
> 5. (Q4, Q9) Weyl's inequality and related bound proof
>
>    A: In lines 643-650, Appendix, we proved that $\Vert W_{\ell}^{k+1} - W_{\ell}^{0}\Vert_2 \leq C_\ell$. The Weyl's Inequality for singular values is $|\sigma_k(A)-\sigma_k(B)| \leq \sigma_1(A-B)$, where $\sigma_k$ represents $k$-th largest singular value. In our paper, we take $\sigma_k$, which is spectral norm in our formulation. Thus, we have $\Vert W_{\ell}^{k+1} \Vert_2 - \Vert W_{\ell}^{0}\Vert_2 \leq C_\ell \Vert W_{\ell}^{k+1} - W_{\ell}^{0}\Vert_2 \leq C_\ell$.
>
> 6. (W1, Q10) Illustration of NTK
>
>    A: Yes, NTK is implied by the non-singular. The kernel matrix is for the gradient of loss to learnable parameters. The main technique of NTK theory is the establishment of non-singularity of the kernel matrix by a wide-NN layer. This invokes the Polyak-Lojasiewicz condition (a more relaxed condition than strongly convex) for linear convergence. Due to the page limit, we eliminate the explicit formulation of kernel matrix. Based on Nguyen ICML 2021 [1], the non-singularity of kernel matrix is established by $\sigma_{\min}(G^0_{L-1,T}) > 0$. It is guaranteed by the condition in Theorem 4.3 and implemented by our initialization strategy in Section 5.
>
> 7. (Q5) Initialization strategy in Section 5
>
>    A: As in line 279, we only set learnable matrix of last layer to be all zeros. The former layers are randomly sampled. The reviewer may be confused by $\{W_1^0, \dots, W_{L-1}^0, W_L^0 = \mathbf{0}\}$, where $W_1^0, \dots, W_{L-1}^0$ represents no definition. We will make it clearer in the final version.
>
> 8. (Q1) & (Q2): Generalization problem: non-ReLU (Q1) and LSTM (Q2)
>
>    (Q1) A: As in line 572, we use 1-Lipchitz property of ReLU in the proof. Any activations with constant-Lipchitz can be applied.
>
>    (Q2) A: LSTM avoids gradient exploding or vanishing by a gate. However, the theoretical demonstration for LSTM is complex. To our knowledge, there is no LSTM convergence proof with NTK.
>
>
> 9. (W2) NN model on MNIST
>
>    A: We design an additional experiment to show the faster convergence of our proposed L2O than GD on a small CNN network for MNIST dataset in training (our paper mainly focuses on training). The constructions of experiment is as follows. The faster training convergence demonstrates the effectiveness of our proposed initialization and learning rate setting method.
>    * L2O: As in the paper, 3 layers, the width of wide layer: 20 (similar to Scale 3 in Table 1, page 40)
>    * CNN: 1537 learnable parameters
>       * 1st layer: Conv2d(in_channels=1,out_channels=2,kernel_size=3,stride=1,padding=1,dilation=1). Other:Default
>       * 2nd layer: Conv2d(in_channels=1,out_channels=2,kernel_size=3,stride=1,padding=1,dilation=1). Other:Default
>       * 3rd layer: Linear(in_features=3\*7\*7,out_features=10). Other:Default. 10 is for 10 classes of digitals in MNIST.
>    * Configuration of GD
>    * Step size: 0.01
>    * Steps $T$: 10000
>    * Configuration of L2O
>    * Step size: 0.01
>    * Steps $T$: 100 (Note: Our results show that 100 step is enough.)
>    * Number of samples: 200 from MNIST
>    * Loss: Cross-entropy of all samples
>    * Training Epochs: 200
>    * Optimizer: SGD
>    * Learning rate and initialization: Learning rate is $10^{-6}$. $e=100$. (Configurations in our paper)
>
>    We list the loss with step sizes. For GD, we list its loss every 1000 steps. For L2O, we select the first 10 steps. The results show that  L2O converges in one step. However, GD with a 0.01 step size does not converge with 10000 steps. The result demonstrates a great convergence outperformance of L2O over GD on non-convex cases.
>
> |  | 1 | 2  | 3 | 4| 5 | 6  | 7  | 8   | 9 | 10|
> | --- | ---| --- | --- | --- | --- | --- | --- | --- | --- | --- |
> | GD | 9.0698e-03 | 2.7549e-03 | 1.4932e-03 | 9.9039e-04 | 7.2780e-04 | 5.6902e-04 | 4.6368e-04 | 3.8922e-04 | 3.3406e-04 | 2.9170e-04 |
> | L2O 100 Epoch | 2.9170e-04 | 2.9166e-04 | 2.9163e-04 | 2.9159e-04 | 2.9155e-04 | 2.9151e-04 | 2.9148e-04 | 2.9144e-04 | 2.9140e-04 | 2.9136e-04 |
> | L2O 200 Epoch | 1.2220e-04 | 1.2220e-04 | 1.2219e-04 | 1.2218e-04 | 1.2217e-04 | 1.2216e-04 | 1.2215e-04 | 1.2215e-04 | 1.2214e-04 | 1.2213e-04 |
>
> 10. (Q6): Typo.
>
> Limitations:
>
> 1. Generalization
>
>    A: We primarily aim to demonstrate that learning can enhance the convergence of existing algorithms, which can be generalized to any unrolling L2O framework.
>
>    We need to establish NN training convergence. From our investigation, we find that NN training convergence demonstration is a really complex and non-trivial problem. We take the NTK theory to achieve that, which is proposed to demonstrate the convergence proof for typical supervised learning or inverse problems, where $L_2$-norm is typically used as objective. For the generalization of NTK theory for other optimization objectives, the existing works have generalized it to more objectives [1-3], for example, cross-entropy. However, we choose $L_2$-norm in this work since it is straightforward to derive an explicit convergence rate (most existing works do not prove convergence rate w.r.t the objective but a surrogate one [3]), which helps us to design a deterministic initialization strategy. This is more applicable to model and training settings.
>
>    [1] Arthur J. et al. Neural tangent kernel: Convergence and generalization in neural networks. NeurIPS, 2018.
>
>    [2] Chizat, L., & Bach, F. Implicit bias of gradient descent for wide two-layer neural networks trained with the logistic loss. COLT, 2020.
>
>    [3] Yu, Z., & Li, Y. Divergence of Empirical Neural Tangent Kernel in Classification Problems. ICLR, 2025.
>
>    [4] Quynh Nguyen. On the Proof of Global Convergence of Gradient Descent for Deep ReLU Networks with Linear Widths. ICML, 2021.
>
> 2. NAG
>
>    A: NAG is achievable by dynamic system formulation.
>
>    We chose the GD as the backbone algorithm to make it easier to follow. As discussed in Section 3, we are the first to demonstrate a sufficient condition for L2O's training convergence without any assumption on the sequences generated by L2O. From our knowledge, our result is the first general sufficient condition for training convergence. As our first work in the L2O convergence demonstration, we chose the GD as the backend algorithm to complete the training convergence demonstration. From our knowledge, there is no theoretical convergence demonstration for L2O with GD, either.
>
>    We think the convergence proof of NAG is still achievable. Our basic idea is that convergence can be established by properly setting a bound output of L2O to align the convergence of backbone algorithm. For example, in Equation (3), we use the \beta-smooth property to conduct step size. This methodology can be applied to any algorithm with a convergence guarantee. We aim to bound $X_T$ with $X_0$. First, we can formulate the NAG as a linear dynamic system from $X_0$. For some $t$, $X_{t+1}$ is transferred from $X_t$ by the left production of some transferring matrix. Then, we set a proper output of NN to generate momentum terms and step sizes to avoid the transfer matrix from exploding within T steps. Then, the bound of $X_T$ can be derived based on Cauchy-Schwarz and Triangle inequalities.
>
> 3. Input dimension
>
>    A: Our L2O model is coordinate wise, only 2 input features.
>
>    We follow the methodology of Math-L2O (citation 22) to define a L2O model. As in line 115, section 3.1, we set the L2O model to be coordinate-wise, achieved by reshaping the $N$ samples with $d$-size vectors of $X$ and $\nabla F(x)$ into the dimension of samples. Thus, the input features are $\in \mathbb{R}^{Nd\times 2}$.

---

> ### Comment · Reviewer_yQfy · 2025-08-03
> **Response to the rebuttals**
>
> I thank the authors for detailed replies. Most of my concerns were addressed, and I have only minor ones:
>
> - Can the authors discuss the scalability of L2O approach, i.e., what are the advantages and disadvantages of the approach with the size of the network.
>
> - Can the authors discuss the possible convergence in the finite-width regime? Is it necessary in practice to achieve a good performance of L2O framework?
>
> - Did the authors try to test L2O framework beyond simple models and datasets. This is especially important since training MLP on MNIST does not require lr adjustments during the training, and fixed stepsize works pretty well. How generalizable are those results to larger settings?

---

> ### Author Response · Authors · 2025-08-04
>
> Thanks for your comment and questions.
>
> 1. Q1 and Q2: Size of NN in L2O and finite-width regime.
>
>    1. Q1: Size of NN in L2O
>
>       L2O still requires learning. From the perspective of generalization, according to the VC dimension theory [1], we expect minimizing the number of learnable parameters to enhance its robustness in solving unseen optimization problems. However, it still requires enough learnable parameters to maintain its capability to learn.
>
>       From the perspective of theoretical training convergence, a wide NN is required to ensure the convergence. To our knowledge,  there is no convergence guarantee of a thin NN with fewer learnable parameters. We provide a simple, more in-depth analysis as follows.
>
>       **Wide-NN requirement.** The non-convexity is the nature of machine learning. For example, a simple MLP with ReLU as an inner activation function is non-convex. The optimization of NN is typically achieved by utilizing a simple first-order method to solve a non-convex problem. The convergence proof of such a process is extremely non-trivial since we can hardly make any reformulations of the variables (parameters of NN). Due to this difficulty, we cannot utilize many existing mathematical techniques, such as relaxation or primal-dual. However, the NTK theorem raises based on the Polyak-Łojasiewicz (PL) inequality, which derives a **linear** convergence rate from L-smoothness and $\frac{1}{2}\Vert \nabla f(x) \Vert^2 \geq \mu(f(x)-f^*)$ (PL inequality) [2]. The second term, PL inequality, requires that the gradient can be lower bounded by the objective and optimal value, which is a non-negative term. In the scenarios of NTK, the $\Vert \nabla f(x) \Vert^2$ (named NTKernal matrix) is calculated by $\lambda_\min (\nabla f(x) \nabla f(x)^T)$  (and further decoupled to layer-wise derivative [3]), where $\nabla f(x)$ is $N$-by-$w$ matrix, $w$ is the denotation for a layer's width, N is number of samples. The PL-inequality defines the NTK matrix to be non-singular, which requires at least one layer to be wide, such that there exists some $w \geq N$.
>
>    2. **Relaxation?** (Q2:  finite-width regime)
>
>       Since the convergence rate from NTK theory is linear but requires a strict condition of a wide NN, we can trivially expect some space for relaxation (a thin NN) and a corresponding slower convergence rate, for example, sub-linear. Empirically, it does since most applications of NN are not wide. Our result in Figure 8 is an example. However, to our knowledge,  there are no existing theoretical works to achieve that.
>
>
>
> 2. Q3: Larger experiments
>
>     For larger settings, we think it is not within the scope of this work. We would like to try to learn to optimize larger NNs in the future. However, it has gained a lot of attention and has several successful deployments in real-world scenarios. The related research topic is Meta-Learning. Nowadays, the most popular research direction of Meta-Learning is utilizing reinforcement learning to fine-tune LLMs.
>
>
>
> Citation:
>
> [1] Kearns, Michael J., and Umesh Vazirani. An introduction to computational learning theory. MIT press, 1994.
>
> [2] Karimi, Hamed, Julie Nutini, and Mark Schmidt. Linear convergence of gradient and proximal-gradient methods under the polyak-łojasiewicz condition. Joint European conference on machine learning and knowledge discovery in databases. Cham: Springer International Publishing, 2016.
>
> [3] Quynh Nguyen. On the Proof of Global Convergence of Gradient Descent for Deep ReLU Networks with Linear Widths. ICML, 2021

---

### Decision · Program_Chairs · 2025-09-17

**Decision:**

Accept (poster)

**Comment:**

**Summary.** This work presents a rigorous convergence analysis for the Learning to Optimize (L2O) framework. Specifically, the authors study the learning of coordinate-wise step sizes for Gradient Descent (GD) applied to the least-squares problem and demonstrate theoretical improvements over standard GD. The paper establishes a linear convergence rate for the Math-L2O framework, and the theoretical claims are well supported by accompanying numerical experiments.

**Strengths and weaknesses.** All reviewers recognize the non-triviality of the considered problem and acknowledge that the paper makes a meaningful contribution toward bridging the gap in the theoretical understanding of L2O methods. On the other hand, reviewers highlighted several weaknesses, including the limited scale of tested models, questions about scalability, and some inaccuracies and typographical errors in the claims and proofs. The authors acknowledged these issues and committed to addressing them in the final version.

**Final recommendation.** Given the importance of the problem, the novelty of the results, and the positive assessments from the reviewers, I recommend acceptance. In my view, the strengths of the paper outweigh its weaknesses. However, it is crucial that the authors carefully implement all the promised corrections and improvements in the camera-ready version.